# Functional phenotyping of genomic variants using joint multiomic single-cell DNA–RNA sequencing

Dominik Lindenhofer [1,2], Julia R. Bauman [3,16], John A. Hawkins [1,4,16], Donnacha Fitzgerald [1,5,6], Umut Yildiz [1], Haeyeon Jung [1], Anastasiia Korosteleva [1,7], Mikael Marttinen [8,9], Moritz Kueblbeck [1], Judith B. Zaugg [10,11], Kyung-Min Noh [1,12], Sascha Dietrich [5,6,13,14], Wolfgang Huber [1,6], Oliver Stegle [1,4] & Lars M. Steinmetz [1,2,3,15] ✉

Genetic variants (both coding and noncoding) can impact gene function and expression, driving disease mechanisms such as cancer progression. The systematic study of endogenous genetic variants is hindered by inefficient precision editing tools, combined with technical limitations in confidently linking genotypes to gene expression at single-cell resolution. We developed single-cell DNA–RNA sequencing (SDR-seq) to simultaneously profile up to 480 genomic DNA loci and genes in thousands of single cells, enabling accurate determination of coding and noncoding variant zygosity alongside associated gene expression changes. Using SDR-seq, we associate coding and noncoding variants with distinct gene expression in human induced pluripotent stem cells. Furthermore, we demonstrate that in primary B cell lymphoma samples, cells with a higher mutational burden exhibit elevated B cell receptor signaling and tumorigenic gene expression. SDR-seq provides a powerful platform to dissect regulatory mechanisms encoded by genetic variants, advancing our understanding of gene expression regulation and its implications for disease.

Genomic variation in both coding and noncoding regions of the genome drives human population differences and disease[1-3]. Over 90% of predicted genome-wide association study variants for common diseases are located in the noncoding genome, while their gene regulatory impact is challenging to assess. Genetic loss-of-function screening of coding genes and CRISPR interference (CRISPRi)/CRISPR activation screens in noncoding regions have provided valuable insights into disease mechanisms. However, they neglect precise genomic variation potentially masking more complex cellular disease phenotypes caused by individual variants[4-7]. Existing precision genome editing tools to introduce variants have limited efficiency and variable editing outcomes in mammalian cells[8-10]. This makes it difficult to use guide RNAs (gRNAs) as a proxy to annotate the variant perturbation in pooled screens. Although some droplet-based technologies enable assessment of variants within transcripts, they neglect the impact of noncoding variants, which constitute the vast majority of disease-associated variants[11]. Exogenous introduction of sequence variants, via episomal massively parallel reporter assays for noncoding variants or open reading frame expression for coding sequences, allows for high-throughput screening of variants for functional effects but lacks endogenous genomic position and sequence context[12-15]. These limitations hinder systematic studies of endogenous genetic variation and its impact on disease-relevant gene expression.

To confidently link precise genotypes to gene expression in their endogenous context, a combined single-cell genomic DNA (gDNA) and RNA assay is required to directly assess coding and noncoding variants and gene expression in the same cell. Current technologies that enable simultaneous high-sensitivity readout of both gDNA and RNA are well established and laborious with low throughput[16-25]. High-throughput droplet-based or split-pooling approaches can measure thousands

of cells simultaneously but lack combined high-sensitivity and tagmentation-independent readout of gDNA and RNA[26–29]. This results in sparse data with high allelic dropout (ADO) rates (>96%), making it impossible to correctly determine zygosity of variants on a single-cell level. Here, we developed targeted droplet-based single-cell DNA–RNA sequencing (SDR-seq), a scalable and sensitive method to screen genetic variation in high throughput, linking it to gene expression and distinct cellular states.

## Results

### Droplet-based SDR-seq

We developed SDR-seq to simultaneously measure RNA and gDNA targets in the same cell with high coverage across all cells. The assay combines in situ reverse transcription (RT) of fixed cells with a multiplexed PCR in droplets using Tapestri technology from Mission Bio (Fig. 1a). Cells are dissociated into a single-cell suspension, fixed and permeabilized. In situ RT is performed by using custom poly(dT) primers, adding a unique molecular identifier (UMI), a sample barcode (BC) and a capture sequence (CS) to cDNA molecules. Cells containing cDNA and gDNA are loaded onto the Tapestri machine. After generation of the first droplet, cells are lysed, treated with proteinase K and mixed with reverse primers for each intended gDNA or RNA target. During generation of the second droplet, forward primers with a CS overhang, PCR reagents and a barcoding bead containing distinct cell BC oligonucleotides with matching CS overhangs are introduced. A multiplexed PCR amplifies both gDNA and RNA targets within each droplet. Cell barcoding is achieved through the complementary CS overhangs on PCR amplicons and cell BC oligonucleotides. After multiplexed PCR, emulsions are broken, and sequencing-ready libraries are generated. Distinct overhangs on reverse primers containing either R2N (gDNA, Nextera R2) or R2 (RNA, TruSeq R2) allow for separation of next-generation sequencing (NGS) library generation for gDNA and RNA. This enables optimized sequencing of each library: (1) full-length to entirely cover variant information on gDNA targets along with the cell BC and (2) transcript and BC information (cell BC, sample BC and UMI) for RNA targets.

To test SDR-seq, we performed a proof-of-principle (POP) experiment amplifying a small number of gDNA (28) and RNA (30) targets in human induced pluripotent stem (iPS) cells (Fig. 1b). As fixation is critical for in situ RT, we tested two different fixatives, paraformaldehyde (PFA) and glyoxal. PFA is commonly used in in situ RT reactions but can impair gDNA and RNA quality as it cross-links nucleic acids[30]. Glyoxal does not cross-link nucleic acids and was expected to provide a more sensitive readout[31,32]. For simplicity, overhangs on reverse primers for gDNA and RNA were the same (R2N) in this experiment (Extended Data Fig. 1a). After filtering high-quality cells and removing doublets using distinct sets of sample BCs during in situ RT for each fixation condition, we obtained ~9,000 cells from a single SDR-seq run (Fig. 1c,d and Extended Data Fig. 1b–f). Cells were evenly distributed over the two fixation conditions, with over 95% of reads per cell mapping to the correct sample BC on average (Fig. 1e,f). For downstream analysis, contaminating reads were removed from each cell.

gDNA target coverage is expected to be uniform across cells as each cell contains the same gDNA input. We detected 23 of 28 gDNA targets (82%) with high coverage and in the vast majority of cells (Fig. 1g–i). Minimal differences in gDNA target detection and coverage were observed between PFA and glyoxal conditions (Extended Data Fig. 1g,i). RNA target coverage is expected to vary as they were chosen based on a range of expression levels. Indeed, individual RNA targets showed varying expression levels, with some only expressed in a subset of cells (Fig. 1j–l). RNA target detection and UMI coverage increased when using glyoxal compared to PFA (Extended Data Fig. 1h,j). Ubiquitously expressed housekeeping and iPS cell maintenance genes were detected in all cells, whereas other genes showed specific expression only in a subset of cells, consistent with published data (Extended

Data Fig. 2a–d)[33]. Comparing bulk RNA-seq data of human stem cells to pseudo-bulked SDR-seq gene expression showed comparable levels of expression for the vast majority of targets with high correlation (Fig. 1m,n). SDR-seq showed reduced gene expression variance and higher correlation between individually measured cells than iPS cell data from 10x Genomics and ParseBio, indicating greater measurement stability (Fig. 1o and Extended Data Fig. 2e).

To test for potential cross-contamination of gDNA and RNA between cells during in situ RT, we performed a species-mixing experiment using human WTC-11 iPS cells and mouse NIH-3T3 cells. Cells were processed either separately or as a mixed population during in situ RT (Extended Data Fig. 2f). This allowed us to distinguish contamination introduced during in situ RT from general ambient nucleic acids by comparing the mixed-species condition to the single-species controls. We obtained a total of 16,000 cells across the different in situ RT conditions with the vast majority of doublets effectively removed using the sample BC information introduced during in situ RT (Extended Data Fig. 2g,h). Cross-contamination of gDNA was minimal (<0.16% on average), with no difference between the mixed-species and single-species conditions (Extended Data Fig. 2i,k,l). RNA cross-contamination was low (0.8–1.6% on average), with increased levels in the mixed-species condition compared to in the single-species controls (Extended Data Fig. 2j,m,n). The majority of cross-contaminating RNA from ambient RNA could be removed using the sample BC information introduced during in situ RT (Extended Data Fig. 2m,n). These data indicate that overall levels of cross-contaminating nucleic acids are low in SDR-seq.

Together, these results demonstrate that SDR-seq enables highly sensitive detection of DNA and RNA targets across thousands of single cells in a single experiment, with the potential to link both modalities in a high-throughput manner.

### SDR-seq is scalable to hundreds of gDNA loci and genes

Next, we tested whether SDR-seq is scalable to detect hundreds of gDNA and RNA targets simultaneously. We designed an experiment using panels of 120, 240 and 480 targets, with equal numbers of gDNA and RNA targets in iPS cells (Fig. 2a). To enable cross-panel comparison, 60 gDNA and 30 RNA targets were shared between panels. To adjust for differences in sequencing depth, reads were subsampled for gDNA and RNA based on panel size to achieve equal average read coverage per cell for shared targets (Extended Data Fig. 3a–d). We confirmed that separately prepared NGS libraries for gDNA and RNA mapped with high specificity to their respective references (Extended Data Fig. 3e,f). Overall, 80% of all gDNA targets were detected with high confidence in more than 80% of cells across all panels, with only a minor decrease in detection for larger panel sizes (Extended Data Fig. 4a–c). Detection and coverage of shared gDNA targets were highly correlated between panels, indicating that gDNA target detection is largely independent of panel size (Fig. 2b,c). The minor decrease in detection for the larger panel sizes predominantly affected low-coverage targets (Extended Data Figs. 4d,e and 5a). Similarly, RNA target detection showed a minor decrease in larger panels compared to the 120 panel (Extended Data Fig. 4f–h). Detection and gene expression of shared RNA targets were highly correlated between panels (Fig. 2d,e and Extended Data Fig. 4i,j), indicating robust and sensitive detection independent of panel size. Variability was predominantly observed for lowly expressed genes (Extended Data Fig. 5b).

To assess whether chromosomal context influences gDNA detection using SDR-seq, we included target sites among the shared panels that were either overlapping expressed genes (OEGs) or not OEGs (NOEGs). Additionally, we tested for different chromatin marks and states (H3K3me3, H3K27ac and DNase sensitive), reflecting different genomic regulatory element types depending on their proximity to the transcription start site (TSS; Fig. 2f)[34]. We did not observe a strong impact on detection and coverage across panels based on OEG or NOEG

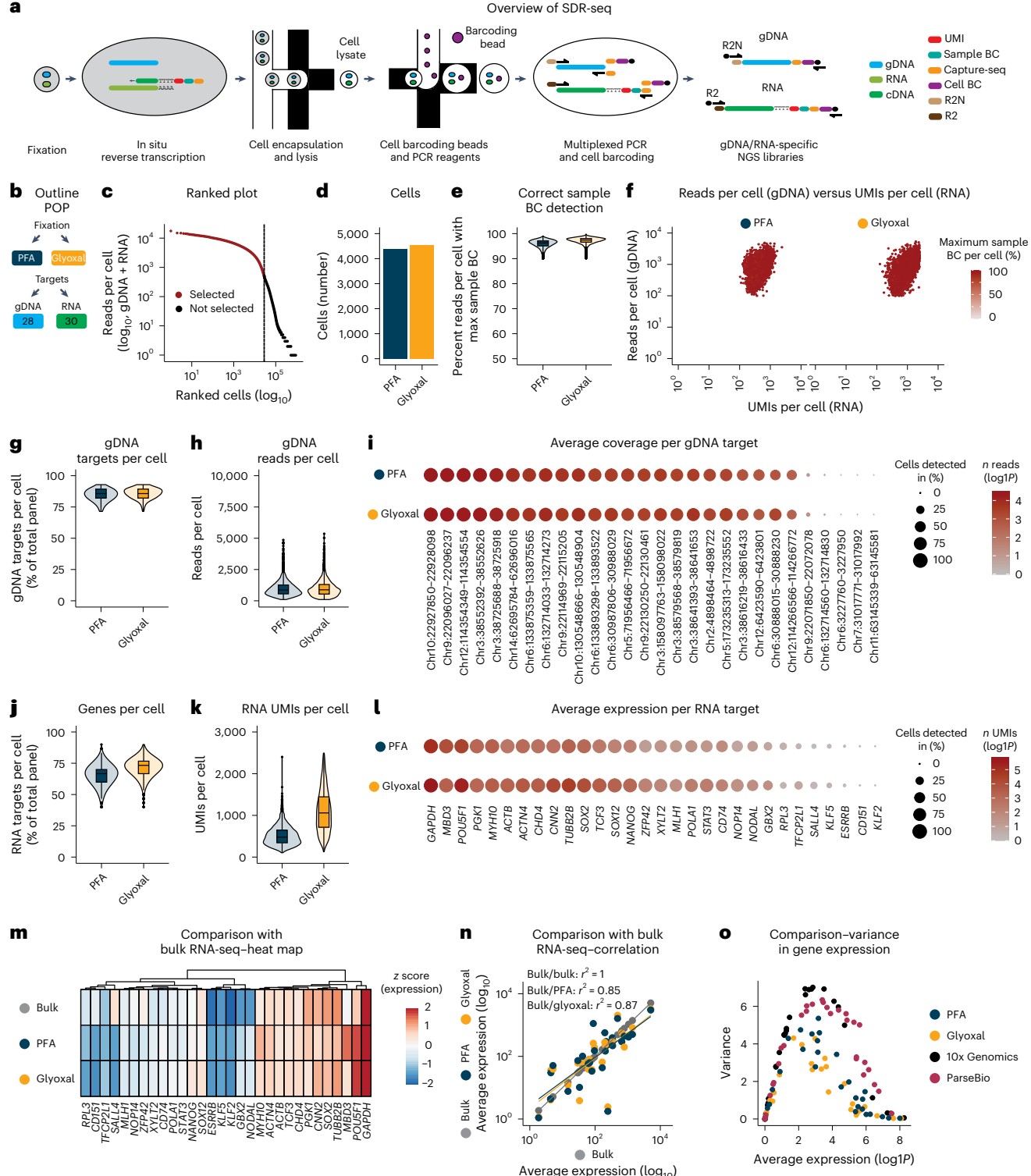

**Fig. 1 | SDR-seq links gDNA variants and gene expression in single cells.**
**a**, Overview of targeted SDR-seq. R2N (Nextera) or R2 (TruSeq) overhangs on reverse primers enable separate NGS library generation for gDNA and RNA. **b**, Outline of the POP experiment. The fixation conditions and number of gDNA/RNA targets are indicated. **c**, Knee plot of ranked cells by sequencing depth (gDNA + RNA). **d**, Number of cells found per fixation condition. **e**, Correct sample BC detection per cell. Read count for maximum sample BCs found was divided by the total amount of RNA reads per cell; $n = 4,391$ (PFA) and 4,553 (glyoxal) cells from one SDR-seq experiment. **f**, gDNA reads per cell versus RNA UMIs per cell per fixation condition. Color indicates the percentage of reads per cell with max sample BCs. **g,h**, Number of gDNA targets per cell (**g**) and gDNA reads per

cell (**h**); $n = 4,391$ (PFA) and 4,553 (glyoxal) cells from one SDR-seq experiment. **i**, Individual gDNA targets are shown per fixation condition. Size indicates the percentage of cells detected in. Color indicates read coverage; chr, chromosome. **j,k**, Number of genes per cell (**j**) and RNA UMIs per cell (**k**); $n = 4,391$ (PFA) and 4,553 (glyoxal) cells from one SDR-seq experiment. **l**, Individual genes are shown per fixation condition. Size indicates the percentage of cells detected in. Color indicates UMI coverage. **m**, Comparison of expressed genes to bulk RNA-seq data; $z$ score data are scaled by row. **n**, Pearson correlation of expressed genes to bulk RNA-seq data. **o**, Average expression and variance of genes assayed in the POP experiment using SDR-seq, 10x Genomics and ParseBio.

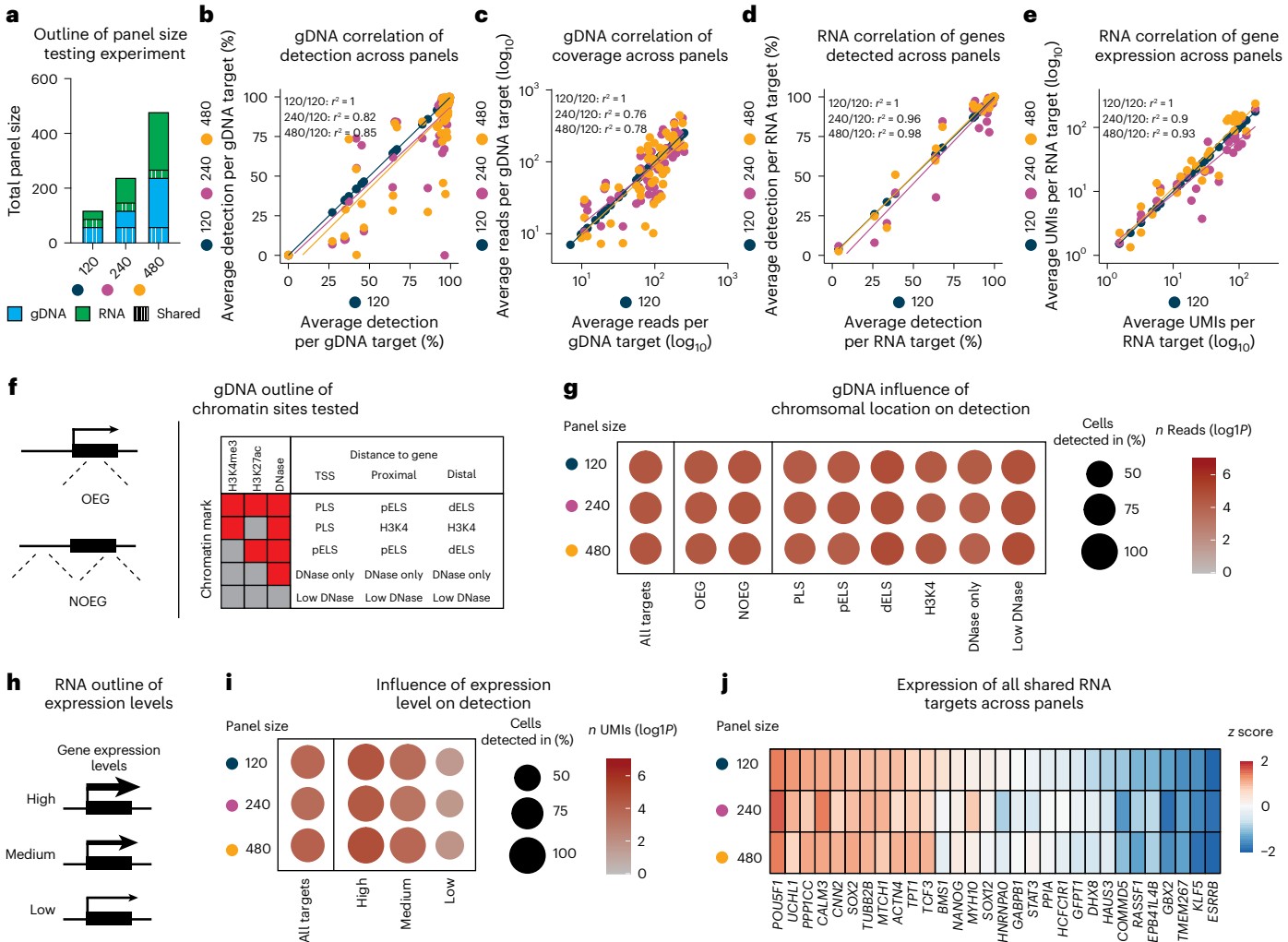

**Fig. 2 | SDR-seq scales to hundreds of targets simultaneously. a**, Outline of panel size testing experiments. gDNA and RNA targets are equal within panels; shared targets are indicated. **b,c**, Pearson correlation of detection (**b**) and coverage (**c**) of shared gDNA targets between panels. **d,e**, Pearson correlation of detection (**d**) and coverage (**e**) of shared genes between panels. **f**, Outline of chromatin sites tested. A combination of the chromatin marks detected and the relative distance to a gene define regulatory elements; PLS, promotor-like sequence; pELS, proximal enhancer-like sequence; dELS, distal enhancer-like sequence. **g**, Detection of different chromatin sites between panels. Size indicates the percentage of cells detected in. Color indicates read coverage. **h**, Outline of expression levels tested. **i**, Detection of genes with different expression levels between panels. Size indicates the percentage of cells detected in. Color indicates read coverage. **j**, Heat map of expression of all shared genes between panels; z score data are scaled by row.

location (Fig. 2g). Notably, no specific regulatory element type showed a systematic detection bias, and even sites with low DNase signal were confidently recovered.

Genes were selected based on a range of expression levels and grouped into high, medium and low expression (Fig. 2h). High- and medium-expression genes were detected in almost all cells, whereas low-expression genes showed reduced detection rates across panel sizes (Fig. 2i). This aligns with published data suggesting that some genes are not expressed in all iPS cells[33]. Overall expression levels of shared genes were highly similar across the different panel sizes tested (Fig. 2e,j).

To determine the ADO rate of SDR-seq, we selected gDNA amplicon loci containing heterozygous single-nucleotide polymorphisms based on bulk sequencing data. In amplicons detected in more than 80% of cells, heterozygous variants were correctly called in an average of 87–94% of cells (Extended Data Fig. 5c,d). The primary cause of ADO in larger panels was overall low detection rates of a gDNA target (Extended Data Fig. 5e). Noise levels of both miscalled variants and deletions or insertions were low (<0.15%) and showed comparable levels across panels (Extended Data Fig. 5f–h). The frequency of these

lowly abundant miscalled variants depends on the reference base, with PCR deamination byproducts likely being the most common. Variant allele frequencies (VAFs) of individual variants could distinguish true heterozygous alleles from variant noise (Extended Data Fig. 5i).

SDR-seq is thus scalable to assay hundreds of gDNA and RNA targets simultaneously with high reproducibility and sensitivity across different panel sizes, independently of chromatin state and expression level. This makes it a versatile tool to analyze variants at hundreds of loci in single cells, while simultaneously measuring cellular gene expression.

## SDR-seq confidently detects gene expression changes

Genomic variants can increase or decrease gene expression, but effect sizes are often small. Therefore, a sensitive readout of these gene expression changes is essential. We probed the ability of SDR-seq to detect strong and subtle gene expression changes across different perturbation systems designed to repress gene expression or introduce variants that alter expression levels.

To assess whether SDR-seq can detect strong gene expression changes, we designed a CRISPRi experiment composed of four gRNA

categories: (1) nontargeting control gRNAs (NTC), (2) gRNAs targeting expression quantitative trait loci (eQTLs), (3) gRNAs targeting the TSS of genes predicted to be affected by those eQTLs (CRISPRi controls) and (4) gRNAs targeting gene bodies to possibly introduce STOP codons through editing (STOP controls; Fig. 3a). CRISPRi-expressing human iPS cells infected with a lentiviral CROP-seq gRNA library were selected via fluorescence-activated cell sorting, followed by SDR-seq. The SDR-seq primer panel amplified the gDNA sites of the eQTLs together with associated transcripts, the viral CROP-seq transcript to assign cells to gRNAs and multiple housekeeping genes to normalize data. Cells were successfully assigned to gRNAs (75%), with an average of 30 cells per gRNA (Extended Data Fig. 6a,b). NTC gRNAs showed no significant effect on any of the genes measured, whereas most (95%) TSS-targeting CRISPRi control gRNAs caused a strong reduction in target gene expression (Fig. 3b). Seven eQTLs (24%) and three STOP control gRNAs (60%) significantly reduced target gene expression. Significantly scoring eQTL and STOP control gRNAs were located within a 2-kb window of the TSS, suggesting a direct inhibitory effect similar to CRISPRi control gRNAs (Extended Data Fig. 6c). This demonstrates that SDR-seq can confidently detect gene expression changes mediated by CRISPRi. Additionally, these data highlight the importance of directly assessing variants in proximity to the TSS to evaluate their impact on gene expression rather than approximating such effects with CRISPRi.

Next, we aimed to directly install eQTL variants and measure more subtle effects on gene expression (Fig. 3c). We generated two human iPS cell lines expressing a prime editing (PE) transgene, with or without the coexpression of a dominant-negative regulator of the mismatch repair pathway designed to enhance editing efficiency (Pemax or Pemax-MLH1dn)[10]. To validate the system, we used a fluorescent lentiviral reporter system that measures editing efficiency via reconstitution of a nonfunctional enhanced green fluorescent protein (eGFP; Extended Data Fig. 6d). Using a PE gRNA (pegRNA) that repairs eGFP, we observed ~50% editing efficiency, demonstrating the system's editing potential in human iPS cells (Extended Data Fig. 6e). We lipofected these PE iPS cells with a pegRNA library designed to introduce the same eQTLs tested in the CRISPRi screen, as well as STOP codons to assess nonsense-mediated decay. Following fluorescence-activated cell sorting enrichment of lipofected cells, we performed SDR-seq (Fig. 3c). Editing efficiency was limited in both PE cell lines, complicating the interpretation of many variants (Extended Data Fig. 7a,b). Despite this limitation, we performed differential gene expression testing for between called reference (REF), heterozygous (HET) and alternative (ALT) variant alleles (Extended Data Fig. 7c–e). Significant gene expression changes were only observed for the STOP controls (Fig. 3d). Depending on the position of the STOP codon within the transcript, effects of nonsense-mediated decay on transcript levels can vary[35]. For *SOX11*, we observed no changes, whereas STOP codons introduced in *ATF4* and *MYH10* resulted in significant reductions in gene expression (Fig. 3e).

In addition to installing eQTLs with PE, we tested the use of base editing (BE) in human iPS cells. We selected 56 high-likelihood eQTLs with a potential association for gene expression changes based on multiple studies, including noncoding variants that are located in open chromatin and editable with ABE8e or CBE base editors[36–38] (Fig. 3f). None of these variants have previously been experimentally validated in an endogenous context as causative for transcriptional regulation, to our knowledge. After introducing gRNA libraries into iPS cells, cells were selected, and SDR-seq was performed. We found several eQTL variants with a significant effect on target gene expression (Fig. 3g). Additionally, we measured the effect of non-BE-associated mutations using SDR-seq. Human iPS cells accumulate somatic mutations during cell culture, while they undergo constant competitive selection for variants that are advantageous in culture conditions[39]. We found a synonymous variant in the 3′ end of *POU5F1*, a gene encoding a critical pluripotency factor, which significantly altered gene expression in the

same direction as observed in prior eQTL studies[36] (Extended Data Fig. 8a). However, after assessing variants that may have accumulated during culturing along the entire amplicon, we found that certain combinations of variants showed different effects on *POU5F1* expression (Fig. 3h and Extended Data Fig. 8b). In particular a set of variants in the 3′ untranslated region was associated with significantly different transcript levels. The presence of these variants was confirmed by bulk amplicon sequencing of this locus (Extended Data Fig. 8c). This highlights the importance of directly assessing variants at the locus of interest to accurately resolve their impact on gene expression.

SDR-seq can confidently detect variants at the single-cell level and associate them with gene expression differences, demonstrating sensitivity even for subtle changes. This is the case even under conditions of limited editing efficiency in our experiments, which confound the interpretation of many tested eQTLs.

## B cell lymphoma variants linked to tumorigenic expression

Linking genetic variants to gene expression profiles is crucial for understanding cancer pathogenesis yet remains challenging in primary samples. B cell lymphomas are heterogenous cancers of the lymphatic system arising from distinct stages of B cell maturation. In this maturation process, naive B cells are stimulated to migrate through the dark zone (DZ) and light zone (LZ) of the germinal center, where they undergo somatic hypermutation and selection, followed by maturation into memory B cells and plasma cells[40–43]. Although the cell of origin is central to the classification of B cell lymphomas, it was recently shown that cancer cells retain their ability to differentiate. Thereby tumors acquire multiple maturation states from the same cell of origin while simultaneously undergoing clonal evolution through the accumulation of heterogenous genetic variants over time[44,45].

We used B cell lymphomas to investigate how genetic variation impacts gene expression and differentiation within tumors. We analyzed primary tumor samples from two individuals with follicular lymphoma and one individual with germinal center subtype diffuse large B cell lymphoma using SDR-seq (Fig. 4a). A targeted gDNA panel, based on variants from bulk DNA sequencing, was applied to profile 3,600 to 8,400 cells per sample. Clustering of cells showed distinct separation between B cells and non-B cells in both RNA- and variant-based analysis (Fig. 4b,c). Using a reference mapping approach based on mutual nearest neighbors and canonical correlation analysis, we mapped B cell maturation states from a dataset of nonmalignant reactive lymph nodes to tumor samples (Extended Data Fig. 9a,b)[44,46]. Immunoglobulin light chain restriction confirmed monoclonality and malignancy of tumor B cells (Extended Data Fig. 9c)[47]. Somatic HET or ALT variants detected in both malignant B cells and non-B cells suggested limited contributions to disease progression, whereas variants occurring exclusively in malignant B cells may be oncogenic (Fig. 4d). Variants found in bulk gDNA sequencing of the same samples could also be recovered using SDR-seq (Extended Data Fig. 9d). The three samples showed a number of distinct variants, while some predominately somatic variants were shared.

Next, we focused on a comparative analysis between DZ and LZ maturation states as most B cells belonged to these states (>80%). Clustering DZ and LZ cells based on variant information covered with our targeted gDNA panel revealed that two samples (FL2 (follicular lymphoma) and GCB1 (germinal center subtype diffuse large B cell lymphoma)) showed clonal structures (Fig. 4e). Genetic clones showed differences in proportions of the DZ and LZ states annotated by gene expression, indicating that clonal evolution and differentiation are predominantly separate processes. Our data suggest that genetic clones with different variant composition continue differentiating after they arise and can have an impact on differentiation rates.

Differential abundance testing showed that *BCL2* variants, a gene encoding an antiapoptotic factor frequently overexpressed in B cell lymphomas and central to B cell maturation, were enriched in the LZ

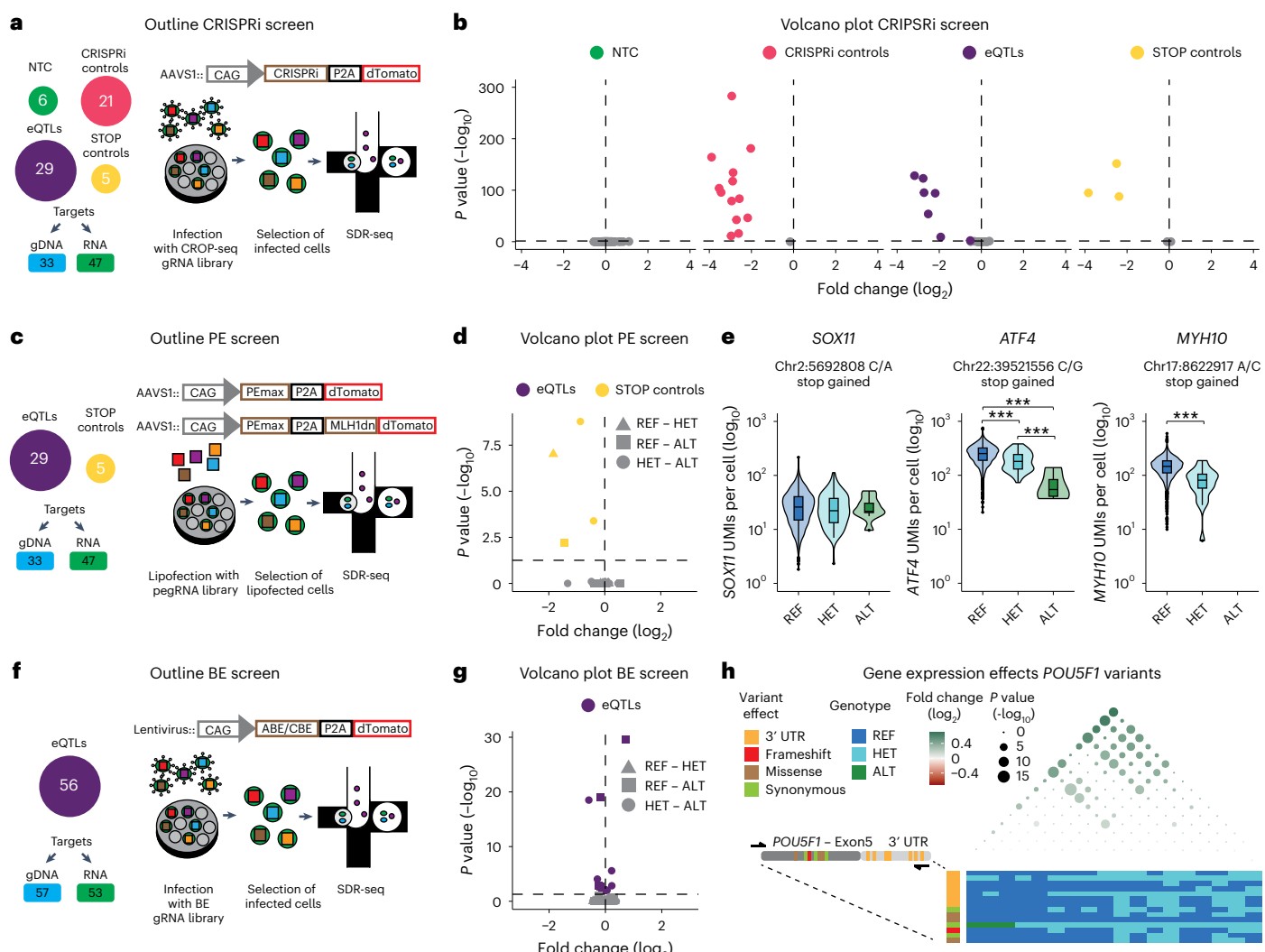

**Fig. 3 | SDR-seq is sensitive to detect gene expression changes and link them to variants. a**, Outline of the CRISPRi screen. **b**, Volcano plot for the CRISPRi screen with different gRNA classes indicating fold change and *P* value calculated using MAST with a Benjamini–Hochberg correction for multiple testing. Significant hits (*P* < 0.05) are colored. For NTCs, all genes measured are shown. For other gRNA classes, only the intended target for each gRNA is shown. **c**, Outline of the PE screen. **d**, Volcano plot for the PE screen with different gRNA classes indicating fold change and *P* value calculated using MAST with a Benjamini–Hochberg correction for multiple testing. Significant hits (*P* < 0.05) are colored. Comparisons between the different alleles are shown as shapes; REF, reference allele; HET, heterozygous allele; ALT, alternative allele. **e**, Alleles and gene expression for *SOX11*, *ATF4* and *MYH10* STOP controls are shown; \*\**P* < 10⁻³ and \*\*\**P* < 10⁻⁴ calculated using MAST with a Benjamini–Hochberg correction

for multiple testing; *n* = 4,152 (*SOX11*: REF), 117 (*SOX11*: HET), 9 (*SOX11*: ALT), 4,916 (*ATF4*: REF), 53 (*ATF4*: HET), 4 (*ATF4*: ALT), 4,925 (*MYH10*: REF) and 18 cells (*MYH10*: HET) from one SDR-seq experiment; $P = 3.05 \times 10^{-4}$ (*ATF4*: REF–HET), $5.38 \times 10^{-8}$ (*ATF4*: REF–ALT), $4.95 \times 10^{-3}$ (*ATF4*: HET–ALT) and $7.90 \times 10^{-10}$ (*MYH10*: REF–HET). **f**, Outline of the BE screen. **g**, Volcano plot for different gRNA classes indicating fold change and *P* value calculated using MAST with a Benjamini–Hochberg correction for multiple testing. Significant hits (*P* < 0.05) are colored. Comparisons between the different alleles are shown as shapes. **h**, Variants in the *POU5F1* locus and their impact on gene expression are shown; UTR, untranslated region. The impact of each variant is color coded. REF, HET and ALT alleles are shown for each genotype. Fold change between the combination of variants is indicated in color (green), and *P* value (−log₁₀) is indicated as size calculated using MAST with a Benjamini–Hochberg correction for multiple testing.

---

compared to DZ (Fig. 4f)[48]. Variants were also enriched in many immunoglobulin variable region genes, which are targeted during somatic hypermutation. LZ cells predominantly exhibited an increase in ALT or REF variant alleles compared to DZ cells (Fig. 4f). Next, we tested if frequent variants impact gene expression in cells belonging to either the LZ or DZ state. We subset cells within each state into variant containing or not containing and performed differential gene expression testing. This revealed a number of genes involved in B cell receptor signaling and tumorigenesis, frequently affected in both DZ and LZ states, with increased participant-specific expression levels predominantly in the LZ compared to in the DZ (Fig. 4g,h and Extended Data Fig. 9e). Elevated B cell receptor signaling is associated with repressing apoptosis in B cell lymphomas[49,50]. Cells with higher mutational burden, characterized by

frequent HET and ALT variants, showed elevated levels of B cell receptor signaling compared to cells with lower mutational burden (Fig. 4i). LZ cells in the geminal center can revert to the DZ following unsuccessful binding to antigens from antigen-presenting cells and thereby undergo multiple rounds of somatic hypermutations[51]. Our data suggest that cells with a high mutational burden may have undergone more rounds of somatic hypermutation and have increased B cell receptor signaling and tumorigenic gene expression patterns to evade apoptosis induced by unsuccessful antigen binding in the LZ. This is in line with the distinct enrichment of variants in the LZ compared to in the DZ that we observe.

Using SDR-seq, we profiled variants and gene expression simultaneously in primary tumor samples, linking cell states to mutational burden. We could distinguish variants present in malignant B cells and

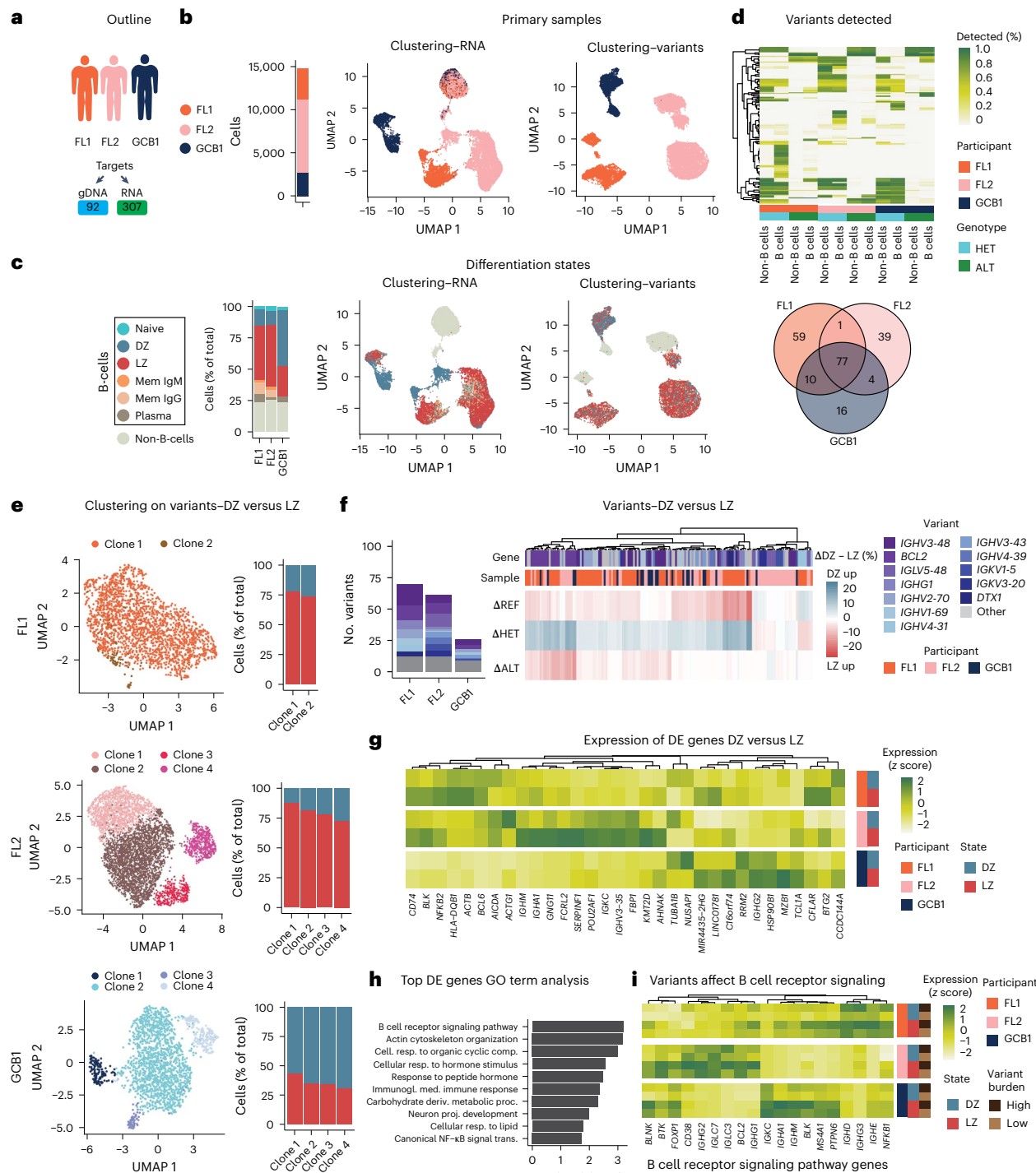

**Fig. 4 | SDR-seq to profile primary samples from individuals with B cell lymphoma. a**, Outline of the experiment. Primary samples and target panels are indicated. **b**, Uniform manifold approximation and projection (UMAP) highlighting the different samples clustered by either gene expression (RNA) or variants (gDNA). The numbers of cells for each sample are indicated as a bar graph. **c**, UMAP highlighting the maturation states clustered by either gene expression (RNA) or variants (gDNA). The numbers of cells within a maturation state are indicated as a bar graph (percentage of total); Mem, memory B cells. **d**, Variants detected in the experiment. Color indicates the percentage within B cells and non-B cells for each variant. Samples and HET/ALT alleles are indicated by color. Venn diagram showing the overlap of variants that occur with more than 5% frequency in each sample. **e**, Subset cells for DZ and LZ maturation states clustered by variants (gDNA) with clones indicated by color for each sample. The numbers of cells within DZ or LZ maturation states are indicated as a bar graph (percentage of total). **f**, Differentially abundant variants between DZ and LZ

states ($P < 0.05$, $\chi^2$ test with a Benjamini–Hochberg correction). Summed counts of genes that the variants map to are shown in a bar graph. ΔDZ – LZ (percentage of the respective allele in DZ minus LZ) is shown in a heat map. Genes that the variants map to and patient of origin are indicated by color. **g**, Gene expression of the most frequently differentially expressed genes across samples in LZ and DZ states. Color indicates expression (z score, data are scaled by column), primary samples and maturation state. **h**, Gene Ontology (GO) term analysis of the most frequently differentially expressed genes. P values were computed using a Fisher's exact test with the weight01 algorithm (topGO), correcting for GO hierarchy structure. cell., cellular; comp., compounds; DE, differentially expressed; proc., process; resp., response; trans., transduction. **i**, Genes involved in B cell receptor signaling in cells with high (top 20%) and low (bottom 20%) variant burden in DZ and LZ states. Color indicates expression (z score, data are scaled by column), primary samples, genes, maturation state and variant burden.

non-B cells, perform clustering analysis based on variants identifying clonal structures, test for enrichment of variants in maturation states and assess their impact on gene expression. This revealed elevated tumorigenic and antiapoptotic signaling in cells with higher mutational burden.

## Discussion

Here, we developed SDR-seq to directly measure gene expression combined with coding and noncoding variants in single cells with high throughput and sensitivity. This method uses targeted primer panels for droplet-based multiplexed PCR to assess both gDNA and RNA in the same cells. Importantly, SDR-seq enables variant detection in noncoding regions of the genome, where the vast majority of disease-associated variants are located[1–3,11]. The targeted approach of SDR-seq facilitates high coverage of gDNA and RNA targets, allowing for confident detection of genomic variants and their zygosity, sensitive gene expression readout and reduced sequencing costs. This contrasts with existing split-pooling or droplet-based approaches, which rely on tagmentation of nucleosome-depleted chromatin and require whole-genome sequencing of each cell, resulting in sparse data and difficulties in correctly determining variant zygosity[26–29]. For these methods, ADO rates are high (>96%), whereas SDR-seq enables accurate detection of ~90% of alleles at the single-cell level. ADO levels of SDR-seq are comparable to targeted single-cell DNA sequencing using Tapestri (ADO < 10%)[52]. Lower-throughput single-cell methods that are plate based rely on tagmentation, amplification via multiple displacement or primary template-directed amplification (PTA) for gDNA readouts while enabling a whole-genome sequencing readout[16–25]. Although tagmentation- and multiple displacement amplification-based technologies also have high ADO rates, PTA achieves a high recovery rate for correctly determined alleles (>90%) when sequencing libraries are at saturation[20,23,24,53]. SDR-seq achieves a drastic ~100-fold increase in cell throughput compared to PTA-based single-cell DNA and RNA-sequencing technologies, while reducing total genome coverage due to its targeted approach[24].

Our results demonstrate that SDR-seq can assay hundreds of gDNA loci and genes simultaneously with high reproducibility and sensitivity across different panel sizes, covering up to 42.8 kb of gDNA per cell. Variants could be detected independent of chromatin context across hundreds of gDNA loci in the same cell. Distinct RNA targets can be picked and adjusted according to experimental needs. The scalability and sensitivity of SDR-seq make it a versatile tool for studying a wide range of coding and noncoding genetic variants and their effects on gene expression across diverse cell types. We can detect variants at a frequency of around 0.15% depending on the type and length of the variant. In both human iPS cells and primary human samples, we link variants to distinct gene expression patterns and can sensitively detect subtle gene expression changes. Advances in PE and pegRNA prediction tools might overcome limitations that we observed in this study constraining the interpretation of several infrequently edited eQTLs. In B cell lymphoma samples, SDR-seq enabled the identification of tumor-specific variants and their associated gene expression profiles, highlighting its potential for studying intratumor heterogeneity and cancer evolution. We could associate cells with higher mutational burden to elevated B cell receptor signaling and tumorigenic gene expression in primary B cell lymphoma samples.

In future applications, SDR-seq could be combined with other readouts, including a targeted protein readout or DNA methylation, to provide a more holistic view of cellular regulation[54,55]. Targeting the mitochondrial genome with SDR-seq could enable clonal tracing of cell populations based on mitochondrial somatic variants[56,57]. Enhanced gene expression readouts might enable measurement of larger RNA panels or a whole-transcriptome readout in parallel to a highly sensitive targeted gDNA readout for multiple loci. Although our attempts for a combined whole-transcriptome readout by using template switch oligonucleotides during the in situ RT reaction were unsuccessful, other experimental approaches might be successful, thereby broadening the scope of potential applications.

SDR-seq offers a powerful, scalable and sensitive approach to link genomic variants to gene expression in single cells, and this method is flexible to assay both genetically engineered cell lines and primary tissue samples. With the vast majority of predicted variants for common diseases located in the noncoding genome, SDR-seq enables the study of these variants systematically at scale[1–3]. In combination with gene editing tools, it holds great potential to decipher the regulatory mechanisms that underlie endogenous variants, complementing other high-throughput approaches that assay the gene expression-to-variant link of endogenous loci or via barcoding approaches[12–15,58,59]. This method advances our ability to study gene expression regulation and its implications for disease, providing insights that could drive the development of therapeutic strategies and enhance our understanding of complex genetic disorders.

## Online content

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

[1]Genome Biology Unit, European Molecular Biology Laboratory (EMBL), Heidelberg, Germany. [2]DZHK (German Centre for Cardiovascular Research), Partner Site Heidelberg/Mannheim, Heidelberg, Germany. [3]Department of Genetics, Stanford University School of Medicine, Stanford, CA, USA. [4]Division of Computational Genomics and Systems Genetics, German Cancer Research Center (DKFZ), Heidelberg, Germany. [5]Department of Hematology, Oncology and Rheumatology, Heidelberg University Hospital, Heidelberg, Germany. [6]Molecular Medicine Partnership Unit (MMPU), European Molecular Biology Laboratory (EMBL), Heidelberg, Germany. [7]Faculty of Biosciences, Heidelberg University, Heidelberg, Germany. [8]Molecular Systems Biology Unit, European Molecular Biology Laboratory (EMBL), Heidelberg, Germany. [9]Faculty of Medicine and Health Technology, Tampere University, Tampere, Finland. [10]Structural and Computational Biology Unit, European Molecular Biology Laboratory (EMBL), Heidelberg, Germany. [11]Department of Biomedicine, University of Basel, University Hospital Basel, Basel, Switzerland. [12]Department of Biomedicine, Aarhus University, Aarhus, Denmark. [13]Department of Oncology, Hematology and Clinical Immunology, Medical Faculty of Heinrich-Heine-Universität, Düsseldorf University Hospital, Düsseldorf, Germany. [14]Center for Integrated Oncology Aachen–Bonn–Cologne–Düsseldorf (CIO ABCD), Düsseldorf, Germany. [15]Stanford Genome Technology Center, Palo Alto, CA, USA. [16]These authors contributed equally: Julia R. Bauman, John A. Hawkins. ✉e-mail: lars.steinmetz@stanford.edu

## Methods

### SDR-seq protocol

A detailed protocol for SDR-seq is published on protocols.oi at https://doi.org/10.17504/protocols.io.6qpvr9q43vmk/v1.

### Cell culture

WTC-11 iPS cells (Coriell Institute for Medical Research, GM25256) were verified to display a normal karyotype, were contamination free and were regularly tested for mycoplasma. Cells were cultured in Essential 8 medium (E8; Thermo Fisher Scientific, A1517001) on Vitronectin XF-coated (StemCell Technologies, 07180) tissue culture plates. Cells were maintained at 37 °C and 5% $CO_2$. iPS cells were split using Accutase (StemCell Technologies, 07922) and E8 supplemented with 10 μM Y-27632 dihydrochloride (RI; Tocris, 1254). After single-cell dissociation, 1 volume of E8 + RI was added, and cells were spun at 200$g$ (5 min), resuspended and plated in E8 + RI. Mouse NIH-3T3 (DSMZ, ACC 59) cells were cultured in DMEM (Gibco, 11965092) supplemented with 10% fetal bovine serum (FBS), 100 U ml$^{-1}$ penicillin/streptomycin and 1× nonessential amino acids on gelatin-coated tissue culture plates at 37 °C with 5% $CO_2$.

### SDR-seq in human iPS cells

WTC-11 iPS cells were dissociated into single cells using Accutase, filtered through a 40-μm cell strainer and counted. For all experiments performed in human iPS cells, $1.5 \times 10^6$ cells were used as input for fixation. This was the minimum number of cells that were used as input for fixation in any experiment that involved human iPS cells.

For glyoxal fixation, cells were resuspended in 200 μl of glyoxal fixation solution (3% glyoxal, 20% ethanol and 0.75% acetic acid (glacial), pH 4.0) and incubated for 7 min at room temperature. One milliliter of ice-cold wash buffer 1 (1× PBS with 2% bovine serum albumin (BSA), 1 mM DTT and 0.5 U μl$^{-1}$ RNasin Plus ribonuclease inhibitor; Promega, N2615) was added, and cells were spun at 500$g$ for 3 min at 4 °C. The supernatant was carefully removed, and the wash step was repeated with wash buffer 1 for a total of two washes. Cells were resuspended in 175 μl of ice-cold permeabilization buffer (10 mM Tris-HCl (pH 7.5), 10 mM NaCl, 3 mM $MgCl_2$, 0.1% Tween 20, 0.2 U μl$^{-1}$ RNasin Plus ribonuclease inhibitor, 1 mM DTT, 2% BSA, 0.1% IGEPAL CA-630 and 0.01% digitonin) and incubated for 4 min on ice. One milliliter of ice-cold wash buffer 2 (10 mM Tris (pH 7.5), 10 mM NaCl, 3 mM $MgCl_2$, 0.1% Tween 20, 0.2 U μl$^{-1}$ RNasin Plus ribonuclease inhibitor, 1 mM DTT and 2% BSA) was added, and the tube was gently inverted four to six times. Cells were spun at 500$g$ for 5 min at 4 °C, resuspended in ice-cold resuspension buffer (1× PBS, 2% BSA, 1 mM DTT and 0.2 U μl$^{-1}$ RNasin Plus ribonuclease inhibitor), filtered through a 40-μm strainer, counted and diluted to $1.4 \times 10^6$ cells per ml.

PFA fixation was performed as described elsewhere with adaptations[60]. In short, cells were resuspended in 1 ml of 1× PBS with 0.2 U μl$^{-1}$ RNasin Plus ribonuclease inhibitor, 3 ml of 1.3% PFA solution (in 1× PBS) was added, and cells were fixed for 10 min on ice. One hundred and sixty microliters of permeabilization buffer (5% Triton X-100 with 0.2 U μl$^{-1}$ RNasin Plus ribonuclease inhibitor) was added, and the tube was gently inverted four to six times and incubated for 3 min on ice. Cells were spun at 500$g$ for 3 min at 4 °C, the supernatant was carefully removed, and cells were resuspended in 500 μl of 1× PBS with 0.2 U μl$^{-1}$ RNasin Plus ribonuclease inhibitor. Ice-cold 100 mM Tris-HCl at pH 8.0 (500 μl) was added and mixed by inverting the tube. Then, 20 μl of permeabilization buffer was added and mixed by inverting the tube four to six times. Cells were spun at 500$g$ for 3 min at 4 °C, the supernatant was removed, resuspended in 300 μl of 0.5× PBS with 0.2 U μl$^{-1}$ RNasin Plus ribonuclease inhibitor, filtered through a 40-μm strainer, counted and diluted to $1.4 \times 10^6$ cells per ml.

Cell loss during fixation ranged between 10 and 30%. This was achieved by performing spins in swinging-bucket rotors and using 15-ml polypropylene centrifuge tubes during the entire process.

RT master mix consisting of a final concentration of 1× RT buffer, 0.25 U μl$^{-1}$ Enzymatics RNase Inhibitor (Biozym, 180520), 0.2 U μl$^{-1}$ RNasin Plus ribonuclease inhibitor, 500 mM dNTPs and 20 U μl$^{-1}$ Maxima H Minus Reverse Transcriptase (Thermo Fisher, EP0752) was prepared on ice in 8 μl for a total reaction volume of 20 μl. Four microliters of RT oligonucleotides (12.5 μM) was combined in each 96-well plate with 8 μl of RT master mix (Supplementary Tables 1 and 2). Eight microliters of fixed and permeabilized cells (10,000 cells) was added to each well, yielding a total reaction volume of 20 μl. We used a total of 48 RT reactions, yielding 480,000 cells, and recommend this number as it provides enough surplus to be in the range of the optimal cell concentration needed for the Tapestri microfluidic device from Mission Bio (105,000–200,000 cells). RT was performed in a thermocycler using the following program: 10 min at 50 °C and three cycles of 2 s at 8 °C, 45 s at 15 °C, 45 s at 20 °C, 30 s at 30 °C, 2 min at 42 °C and 3 min at 50 °C, followed by 5 min at 50 °C. All RT reactions were pooled into a 15-ml conical tube containing 10 ml of 1× PBS with 1% BSA, and cells were spun at 500$g$ for 5 min.

Samples were processed using a Tapestri microfluidic device from Mission Bio (version 2, MB51-0007, MB51-0010 and MB51-0009) according to the manufacturer's protocol, with modifications. The in situ RT-processed cell pellet was resuspended in cell buffer from Mission Bio, and cells were counted and diluted to the appropriate concentration of 4,000–8,000 cells per μl. Custom primers were used in the multiplexed droplet PCR amplification step. RNA primers were designed using the TAP-seq primer prediction tool with a targeted optimal melting temperature of 60 °C (minimum 58 °C and maximum 62 °C) and a product size range from 150 to 300 bp using published single-cell RNA-sequencing iPS cell data for primer prediction (https://www.ebi.ac.uk/biostudies/arrayexpress with E-MTAB-6687)[4,33]. gDNA primers were designed using the Tapestri Designer (https://designer.missionbio.com). Primers were not validated before use; a dropout of around 10–20% is expected for gDNA primers.

Especially for gDNA primers a dropout of custom primers is to be expected. An overview of version 1 and version 2 primer sequences with corresponding overhangs can be found in Supplementary Tables 1 and 2. Version 1 gDNA and RNA primers both had CS and R2N overhangs (only used in the POP experiment). Version 2 gDNA primers had CS and R2N overhangs, whereas RNA primers had CS and R2 overhangs. Detailed information on sample multiplexing using RT primers can be found in Supplementary Table 3. Forward and reverse stock primers had concentrations of 20 μM and 120 μM for both versions, respectively. Both reverse and forward primer mixes contained equimolar amounts of gDNA and RNA targeting primers. For version 1, final sequencing libraries were generated according to the Mission Bio user guide. For version 2, RNA and gDNA sequencing libraries were generated separately using the corresponding library amplification primers (Supplementary Table 2).

### SDR-seq for species cell mixing experiments using human iPS cells and mouse NIH-3T3 cells

Human iPS cells (WTC-11) and mouse fibroblasts (NIH-3T3) were fixed as described above with glyoxal. Cells of each species origin were either used individually or mixed during subsequent in situ RT using a distinct sample BC-introducing RT primer per well (48 in total). Human and mouse genes to profile by SDR-seq were selected to display a range of expression. RNA targeted primers were designed as described above using public data (https://www.10xgenomics.com/datasets/500-1-1-mixture-of-human-hek-293-t-and-mouse-nih-3-t-3-cells-3-lt-v-3-1-chromium-x-3-1-low-6-1-0) from 10x Genomics for NIH-3T3 primer design. Genomic sites were randomly selected, and gDNA targeted primers were designed as described above. Samples were processed using a Tapestri microfluidic device from Mission Bio (version 3, MB03-0091, MB03-0092 and MB03-0093) with modifications as described above.

## SDR-seq in primary B cell lymphoma samples

The study (S-254/2016) was approved by the University of Heidelberg's Ethics Committee. Informed consent from every participant was gathered beforehand. Lymph node samples were processed and frozen following previously described methods[61,62]. Frozen samples were thawed, added to 10 ml of RPMI (Gibco, 11875093) supplemented with 10% FBS and 0.5 mM EDTA and spun at 400*g* for 5 min. Cells were resuspended in 10 ml of 1× DPBS supplemented with 5% FBS, filtered through a 70-μm strainer and spun at 400*g* for 5 min. Cells were resuspended in 100 μl of bead solution from a Dead Cell Removal kit (Miltenyi Biotec, 130-090-101) and incubated for 15 min in the dark. Binding buffer was prepared according to the manufacturer's protocol, and the LS column (Miltenyi Biotec, 130-042-401) was washed with 500 μl of binding buffer. Cells were applied to the column and washed four times with binding buffer while collecting the flow through. Cells were spun at 400*g* for 5 min and resuspended in 1 ml of binding buffer. We proceeded with glyoxal fixation and SDR-seq as described above. For primary cells, input can be a limiting factor to perform SDR-seq. As described above, we recommend 48 wells with 10,000 cells each for the in situ RT reaction yielding 480,000 cells, enough surplus to be in the optimal range for the Mission Bio microfluidic device (105,000–200,000 cells). If there are not enough cells in an individual primary sample, they can be multiplexed on the same Tapestri run by using distinct sample BCs during the in situ RT reaction to achieve the optimal cell concentration needed as input for the Tapestri device. To fill up an entire lane of a Tapestri run, the minimum number of cells that was used in this study for a primary sample was 380,000 cells as input for the glyoxal fixation and 350,000 cells as input for the in situ RT. This represents the lowest number used in this manuscript and yielded around 8,400 cells for this particular run.

The gDNA panel was constructed for regions with >20% VAF detected in the selected tumor samples from the targeted DNA-sequencing data, which were sampled previously in Fitzgerald et al.[44]. Genes for a targeted expression readout of the profiled B cell lymphoma samples were chosen based on both the literature and CITE-seq data from the same samples[44]. These included genes from maturation markers found in the literature and variable features, differentially expressed genes and housekeeping genes based on single-cell RNA-sequencing data (Supplementary Table 4). Primers were designed as described above.

## SDR-seq data analysis

For each SDR-seq dataset generated, we first performed custom BC identification and error correction, mapped reads to custom reference sequences and built read and deduplicated UMI matrices. This was performed with a software package we named SDRranger (https://github.com/hawkjo/SDRranger; https://doi.org/10.5281/zenodo.14762618 (ref. 63)).

The full BC structure for the RNA targeted libraries is of the format cell BC1 (variable-length linker (14–17 bp)), cell BC2 (constant length linker (15 bp)) and sample BC (UMI). The gDNA libraries are the same but lack the sample BC and UMI. To identify these, we first aligned each read to all possible linker backbone sequences to account for the variable-length linker sequences. We discarded alignments with length-normalized alignment scores more than two standard deviations below average, measured from the first 10,000 reads. We then performed error correction on BC1 and BC2 to unique corrected BCs with a Levenshtein distance of 0 or 1. Due to the adjacent UMI, the sample BC does not have an identifiable end point in the case of insertions and deletions, so we corrected sample BCs with free divergence of 0 or 1 and with no other BC with free divergence only 1 higher[64].

Following BC identification, reads were mapped to custom alignment references built for each gDNA and RNA library. For gDNA, the chromosomal locations of the amplicons were used to extract reference sequences. For RNA, the site of the primer binding until the end of the poly(A) tail was used to extract reference sequences. Reference sequences were extracted from GRCh38.p14. Custom fasta and .gtf files

were generated and used to build references using the genomeGenerate function of STAR (v2.7.11a). Separate gDNA or RNA-sequencing libraries were aligned to the corresponding reference, except for checking the specificity of the sequencing libraries.

For the POP experiment, reads were separated into gDNA and RNA reads before BC identification by a separate mapping step to the corresponding references. Final bam files are produced, which contain tags with cell BC and sample BC sequences for each read, both before and after error correction.

Matrices were then constructed by tallying reads by cell BC and sample BC versus gene or gDNA amplicon. To construct the UMI matrices, UMIs were deduplicated by adapting the directional network deduplication method described in the UMItools package[65]. For all reads from a given cell and given gene or amplicon, a connectivity graph of all observed UMIs is constructed. Each node is a unique UMI sequence and read count of that sequence, and directed edges are added between nodes A and B if the two UMIs have a free divergence of 1 to allow for indels, but only if $n_A \geq 2n_B - 1$ reads, where $n_A$ and $n_B$ are the respective numbers of reads. This is based on the observation that each additional error to a UMI sequence should reduce the frequency of observing that sequence. Furthermore, only one incoming 'parent' edge is allowed per node to avoid artifactual connections through singletons. The final number of UMIs is the number of connected components of the graph at the end of this process. This is repeated for each cell and sample BC for each gene or amplicon to build the full matrix.

The resulting cell–gene (UMIs–RNA) and cell–amplicon (reads–gDNA) matrices were analyzed in R using the Seurat R package (v5.0.3). First, a general threshold per cell was set on reads per cell (RNA + gDNA) based on rank–rank plots of reads per cell ranked by size to determine an initial set of cells to include in the analysis. Then, multiple metrics were used to filter for high-quality cells. Only one distinct set of sample BCs is expected to be found per cell; therefore, this can be used to effectively remove doublets in the dataset. Contaminating reads that did not belong to the maximum sample BC found per cell were removed. RNA count matrixes were processed (log-normalized, scaled). A principal component analysis (PCA) was performed on all genes measured for RNA matrices. A probabilistic PCA was performed on all variants measured for VAF matrices. This was followed by subsequent UMAP embedding. For clustering, the shared nearest neighbor graph was calculated and used as input for Louvain clustering.

Each cell that was defined as high quality was then used to call variants using the GATK HaplotypeCaller (v4.2.3.0)[66]. Individual bam files were generated using the cell BC of the high-quality cells using the package sinto (v0.10.0). Each individual cell bam file was modified to contain the cell BC in the read name and indexed using samtools (v1.17), and the MAPQ scores were set from 255 (STAR output) to 60 and to be compatible as an input in the GATK HaplotypeCaller. GATK HaplotypeCaller was run using no maximum read threshold per cell and using a diploidy of two, and resulting vcf files were merged to yield a matrix of cells to variants compatible as input for Seurat. Low-frequency variants (<0.1% for editing and <0.3% for primary human samples) were removed, and remaining variants were input into the Ensembl Variant Effect Predictor for functional annotation[67]. This functional annotation was added as metadata. Genotypes of the GATK HaplotypeCaller were added as an assay to the previous Seurat object, while the remaining output was added as metadata. Wild-type alleles were included based on the read depth for a given amplicon per cell. Both wild-type alleles and variant alleles were excluded from subsequent analysis if the read depth was low (<10 reads) or the genotype quality score of the GATK HaplotypeCaller was low (<30 GQ).

For comparison of variance in the proof-of-concept experiment, published 10x Genomics and ParseBio data (https://www.parsebiosciences.com/customer-datasets/multi-omics-approach-for-near-full-length-human-ipsc-transcriptomes-in-cardiomyocyte-models/#download) were used[33].

## Cloning, molecular biology and generation of transgenic iPS cells

For constitutive CRISPRi and PEmax and PEmax-MLHdn1 cell lines, the corresponding transgene was inserted into the *AAVS1* locus in WTC-11 iPS cells as previously described using specific TALENs[5,68]. The *AAVS1* targeting vector containing the homology arms, the CAGG promotor and a WPRE was a kind gift from J. A. Knoblich (Institute of Molecular Biotechnology of the Austrian Academy of Science, Vienna BioCenter). For the CRISPRi plasmid, pHR-UCOE-SFFV-dCas9-mCherry-ZIM3-KRAB (Addgene, 154473) was modified to pHR-UCOE-SFFV-dCas9-mCherry-KRAB-MECP2 with DNA fragments ordered from Twist Bioscience containing KRAB–MECP2. dCas9, KRAB–MECP2 and dTomato were amplified and cloned into the *AAVS1* targeting vector described above with NEBuilder HiFi DNA Assembly Master Mix (New England Biolabs, M5520). For the PEmax and PEmax-MLHdn1 plasmids, the CRISPRi plasmid was used as a backbone while inserting the PEmax or PEmax-P2A-MLHdn1 (Addgene, 174828) sequence with NEBuilder HiFi DNA Assembly Master Mix. To generate transgenic iPS cells expressing the CRISPRi, PEmax or PEmax-MLHdn1 transgene from the *AAVS1* locus, WTC-11 iPS cells were electroporated with the corresponding homology plasmid (3 μg per electroporation) and two TALEN plasmids (0.75 μg per electroporation each) targeting the *AAVS1* locus (Addgene, 52341 and 52342). iPS cells were dissociated into a single-cell suspension and counted, and $1 \times 10^6$ cells were electroporated using the CB-150 program of the 4D-Nucleofector System and the P3 Primary Cell 4D-Nucleofector X Kit L (Lonza, V4XP-3024), according to the manufacturer's protocol and plated in E8 + RI. Cells were sorted 7–10 days after electroporation for dTomato using a BD Fortessa instrument running Diva (V9.0.1) sofware, plated at low density in E8 + RI, grown to colonies, picked and genotyped (Supplementary Table 5). Positively genotyped clones were checked for homogenous dTomato signal and validated for activity in a corresponding assay, and three clones of each cell line were subjected to a genotyping array screening using an Infinium Global Screening Array-24 kit (Illumina, 20030770) to check for chromosomal rearrangements in iPS cell clones. Only clones that showed no or minor differences to the WTC-11 wild-type parental cell line were used in this study.

## gRNA/pegRNA design and library cloning

All gRNA and pegRNA libraries were cloned in pools. eQTLs were selected based on high confidence from published data, and both lowly expressed genes (counts per million (CPM) > 150) and essential genes in iPS cells were removed also based on published data[5,36,37,69]. eQTLs for the base editor screen were further filtered by overlap with ATAC-seq peaks, expression (transcripts per million > 10) and compatibility with transversion by adenine or cytosine base editors (A > G, C > T, G > A, T > C)[38]. Sites to introduce STOP codons were chosen manually in the selected genes. pegRNAs were designed using Prime-Design (https://drugthatgene.pinellolab.partners.org), and linkers to separate the pegRNA from the tevopreQ1 3' stabilizing sequence were designed using pegLIT (https://peglit.liugroup.us)[70,71]. For the CRISPRi experiment, the spacer sequences of the above pegRNAs were used for eQTLs and STOP controls, whereas gRNAs targeting the TSS for genes predicted to be affected by eQTLs were designed using CRISPick (https://portals.broadinstitute.org/gppx/crispick/public), and NTCs were chosen from the GeCKO-v2 library[72–74]. BE gRNAs were designed and selected based on highest predicted editing efficiency using the BE-Hive tool[75]. The pegRNA screening vector was a kind gift from J. A. Knoblich. This vector was modified to remove the ERT2-Cre-ERT2 sequence and the gRNA scaffold and include a 3' stabilizing tevopreQ1 after the insertion site for pegRNAs via BbsI Golden Gate cloning. The BE vectors were all-in-one cytosine or adenine base editor + guide expression constructs (Addgene, 158581 and 179097). The gRNA screening vector was a modified CROP-seq vector (Addgene, 86708) to also express eGFP and include a distinct gRNA CS in the scaffold of the gRNA[76]. Oligonucleotide pools for pegRNA and gRNA libraries were checked for the presence of BbsI and Esp3I sites within the spacer/RT/PBS sites and ordered from IDT as oPools. pegRNA oligonucleotides included a spacer sequence, PBS and RT with overhangs for amplification that included BbsI sequences compatible with Golden Gate cloning. Spacer and PBS/RT sequences were separated by a constant sequence containing two Esp3I sites for a second round of Golden Gate cloning to introduce the pegRNA scaffold. gRNA oligonucleotides consisted of spacer sequences and overhangs for amplification that included BbsI sequences compatible with Golden Gate cloning. Oligonucleotides were amplified (eight cycles) with compatible primers. The purified PCR product was cloned into the respective pegRNA or gRNA screening vector described above using BbsI and Golden Gate cloning. Electrocompetent bacteria (Lucigen, 60242-1) were electroporated (10 μF, 600 Ω, 1,800 V, E = 184 V cm⁻¹) with purified ligation product and grown in a pool for 10 h at 30 °C before extracting plasmid DNA. For pegRNA and base editor guide libraries, a scaffold sequence was ordered with overhangs that included Esp3I overhangs (IDT), amplified with complementary primers (eight cycles), purified and cloned as described above using Esp3I Golden Gate cloning. Sequence overviews for cloning of the respective gRNA/pegRNA libraries can be found in Supplementary Table 5.

## Virus production, infection of human iPS cells and lipofection of human iPS cells

Lentiviruses were produced in HEK293T (ATCC, CRL-3216) cells grown in DMEM supplemented with 10% FBS, 1× GlutaMAX (Gibco, 35050061), 100 U ml⁻¹ penicillin–streptomycin (Gibco, 15140122) and 1× MEM nonessential amino acids (Gibco, 11140050) and coated using VSV-G. The day before transfection, HEK293T cells were plated at 80% confluency, plasmids were lipofected using Lipofectamine 3000 Transfection reagent (L3000001), and the cells were split 1:10 5 h after lipofection. The supernatant was collected 3 days after lipofection, cell debris was pelleted at 200*g* for 5 min at 4 °C, the remaining supernatant was spun at 28,000*g* for 5 h, and the virus pellet was resuspended in the appropriate volume of E8 + RI. On the day of infection, human iPS cells were split 1:2.5 2 h before infection using Accutase. Infections were performed overnight in E8 + RI. Medium was replaced the next day with E8. For some editing experiments, constructs were only expressed transiently in human iPS cells. For this, transfection was performed using Lipofectamine Stem Transfection Reagent (STEM00003) according to the manufacturer's protocol.

## Target selection and subsampling for panel size testing

Public variant information data for the WTC-11 cell line was downloaded from University of California, Santa Cruz (https://s3-us-west-2.amazonaws.com/downloads.allencell.org/genome-sequence/AH77TTBBXX_DS-229105_GCCAAT_recalibrated.vcf.gz). Variants were filtered using bcftools for heterozygous variants and quality (GT = 'het', filter = 'PASS', format/DP > 70, format DP < 150, QUAL > 1,000, INFO/MQ > 59.8) and subset to contain single-nucleotide polymorphisms, insertions/deletions or multinucleotide polymorphisms. Candidate *cis*-regulatory elements (cCREs) for five human iPS cell lines (H1, H7, H9, iPS DF 6.9 and iPS DF 19.11) were obtained from SCREEN (https://screen.encodeproject.org), and corresponding regulatory elements were subset from these[34]. Genomic regions were defined as OEGs or NOEGs if the gene overlapping that genomic region was expressed in bulk RNA-seq data (>10 CPM)[5]. These OEG or NOEG regions were then overlapped with the filtered WTC .vcf file to select for regions containing high-quality variant information. One hundred and twenty regions were randomly subsampled for each cCRE within the OEG and NOEG classes, and primers were designed as described above. Each cCRE with OEG and NOEG was equally represented in both the total and shared panels. Genes were subset into highly (>400 CPM), medium (<400, >40) and lowly (<40, >4)

expressed gene groups. Primers were designed as described above. After determining high-quality cells in all panels, as described above, they were subset from the bam file, and reads per cell for gDNA and RNA were scaled according to panel size in a way so that the average number of reads per cell for shared gDNA and RNA targets was the same. Variants were called as described above.

## Maturation state assignment in primary tumor samples and immunoglobulin light chain restriction analysis

B cell maturation states were mapped to each tumor sample from a published reactive lymph node single-cell RNA-sequencing dataset through shared gene expression features using previously described methods[44,46]. Gene expression was used to determine immunoglobulin light chain restriction. Log-normalized counts (without batch effect correction to prevent bias introduced by sample integration) were used to find transfer anchors and project samples on the reference PCA (50 dimensions) and UMAP (2 dimensions) reductions. The expression of genes encoding immunoglobulin-κ (*IGKC*) and immunoglobulin-λ (*IGLC1–IGLC7*) light chain was used to determine cell malignancy through light chain restriction[47].

## Data analysis for primary B cell lymphoma samples

Separate NGS sequencing libraries for gDNA and RNA were analyzed with SDRranger, and variants were called as described above. Low-frequency variants (<5% or less than 30 heterozygous/homozygous variants) were excluded from the analysis. GO term analysis was performed using the R package topGO (v2.54.0), the weigth01 algorithm and a Fisher's exact test. Enrichment for biological processes was computed for the top 21 differentially expressed genes in the LZ versus DZ across all samples versus all genes measured.

## Statistics and reproducibility

No data were excluded from analysis, and cutoffs for defining high-quality cells in SDR-seq were set as described above. Differential gene expression testing in the single-cell data was performed using MAST and by subsetting cells in the respective genotype within a given cell or perturbation state[77]. Differential abundance testing of variants between maturation states in primary B cell lymphoma samples was performed using $\chi^2$ testing, followed by adjusting *P* values with the Benjamini–Hochberg method. All box plots shown in this study show the center line as the median, box limits indicate 25th and 75th percentiles, and whiskers indicate 1.5× the interquartile range; all outliers are displayed.

## Reporting summary

Further information on research design is available in the Nature Portfolio Reporting Summary linked to this article.

## Data availability

Sequencing data and processed data for nonprimary human samples are available on Gene Expression Omnibus under accession number GSE268646. Sequencing data and processed data for primary human samples are available on the European Genome–Phenome Archive under study number EGAS50000000374 and dataset ID EGAD50000000551. The dataset on European Genome–Phenome Archive is read-only under ega-archive.org/datasets/EGAD50000000551. Access to the data will be granted for appropriate use in research and will be governed by the provisions laid out in the terms contained in the Data Access Agreement. Source data are provided with this paper.

## Code availability

All relevant code will be deposited on GitHub upon publication. Code containing SDRranger to generate count/read matrices from RNA or gDNA raw sequencing data is available under https://github.com/hawkjo/SDRranger. Code for TAP-seq prediction, generation of custom STAR references and processing of the data is available under https://github.com/DLindenhofer/SDR-seq.

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

## Acknowledgements

We thank all members of the laboratory of L.M.S. for feedback and technical expertise. We thank the Genomics Core Facility (GeneCore) at EMBL for consultation and sequencing. We thank J.-P. Mallm and the Single-Cell Open Lab (scOpenLab) at DKFZ for expertise and access to the Tapestri instrument from Mission

Bio. Support by the DKFZ scOpenLab is gratefully acknowledged. We thank the Flow Cytometry Core Facility at EMBL for support and flow cytometry services. D.L. received funding from an HFSP postdoctoral fellowship (LT0023/2022-L). J.R.B. was supported by an NSF graduate research fellowship (DGE-1656518) and a Stanford Graduate Fellowship (Smith Fellowship). J.A.H. was supported by the European Research Council (Synergy Grant DECODE under grant agreement 810296) and a fellowship through state funds approved by the State Parliament of Baden-Württemberg for the Innovation Campus Health + Life Science Alliance Heidelberg Mannheim. D.F. received funding from a grant from the German Federal Ministry of Education and Research (SIMONA, 031L0263A). M.K. received funding from an EMBL-GSK Collaboration Fund. U.Y. was supported by the EMBL predoctoral fund. L.S.M. was supported by the Dieter Schwarz Foundation Endowed Professorship and the National Human Genome Research Institute, National Institutes of Health (R01HG011664). Work in the laboratory of L.M.S. at EMBL was supported by a grant from the Else Kröner Fresenius Stiftung (2021_EKS.:61).

## Author contributions

D.L. and L.M.S. designed the study, analyzed the data and wrote the manuscript with input from all authors. D.L., J.R.B., D.F., H.J., A.K. and M.K. performed experiments and analyzed the data. J.A.H. developed the SDRranger pipeline for processing SDR-seq data. D.L. and D.F. designed, performed and interpreted the study of B-cell lymphoma patient samples. U.Y. and M.M. provided protocols for fixation compatible with in situ RT. J.B.Z., K.-M.N., O.S., S.D., W.H. and L.M.S. acquired funding, contributed to data interpretation and provided scientific guidance and supervision.

## Funding

## Competing interests

L.M.S. is cofounder and shareholder of Sophia Genetics and founder and board member of LevitasBio and Recombia Biosciences. L.M.S. and D.L. have submitted a patent application on 'High-throughput multiomic readout of RNA and genomic DNA within single cells' (PCT/US2024/029950). D.F. is a cofounder and board member of Anthropocene Bio. The remaining authors declare no competing interests.

## Additional information

**Extended data** is available for this paper at https://doi.org/10.1038/s41592-025-02805-0.

**Correspondence and requests for materials** should be addressed to Lars M. Steinmetz.

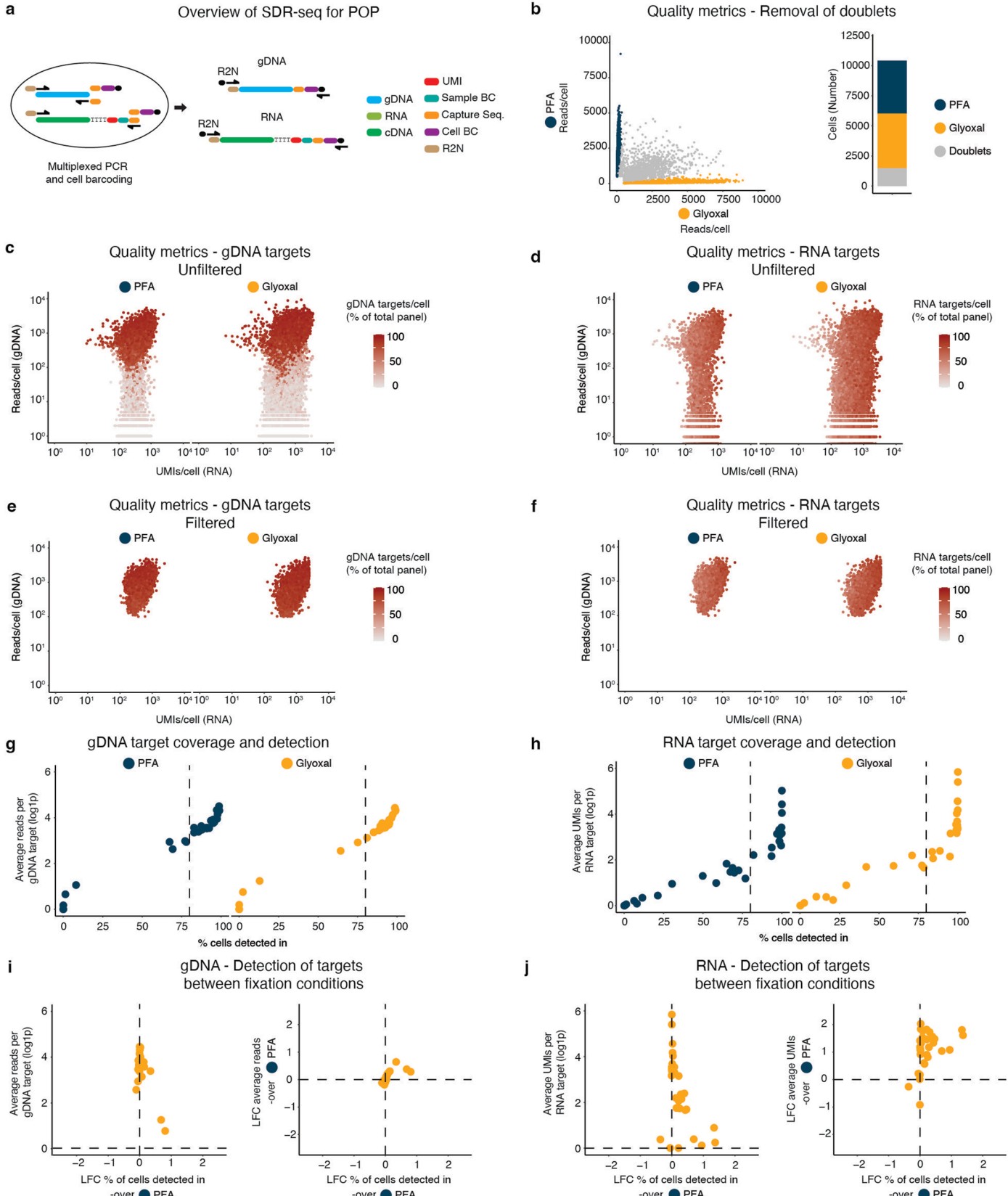

**Extended Data Fig. 1 | Quality controls and comparison of PFA and glyoxal fixation conditions. a**, Overview of targeted scDNA-scRNA-seq (SDR-seq) method for proof-of-principle (POP) experiment. R2N (Nextera) reverse primer overhangs were used for both gDNA and RNA. **b**, Quality control plots to remove doublets showing RNA reads per cell for each set of sample barcodes used to discriminate PFA and glyoxal fixation conditions. Doublets are indicated in grey.

Cell numbers are indicated on the right. **c–f**, Quality control plots before (**c**, **d**) and after (**e**, **f**) filtering for low quality cells. Color indicates fraction of detected gDNA or RNA targets respectively. **g**, **h**, Coverage and detection of each gDNA (**g**) or RNA (**h**) target. Dashed line indicates 80% detection. **i**, **j**, Comparison of coverage and detection between PFA and glyoxal for each gDNA (**i**) and RNA (**j**) target. LFC, log fold change (log2).

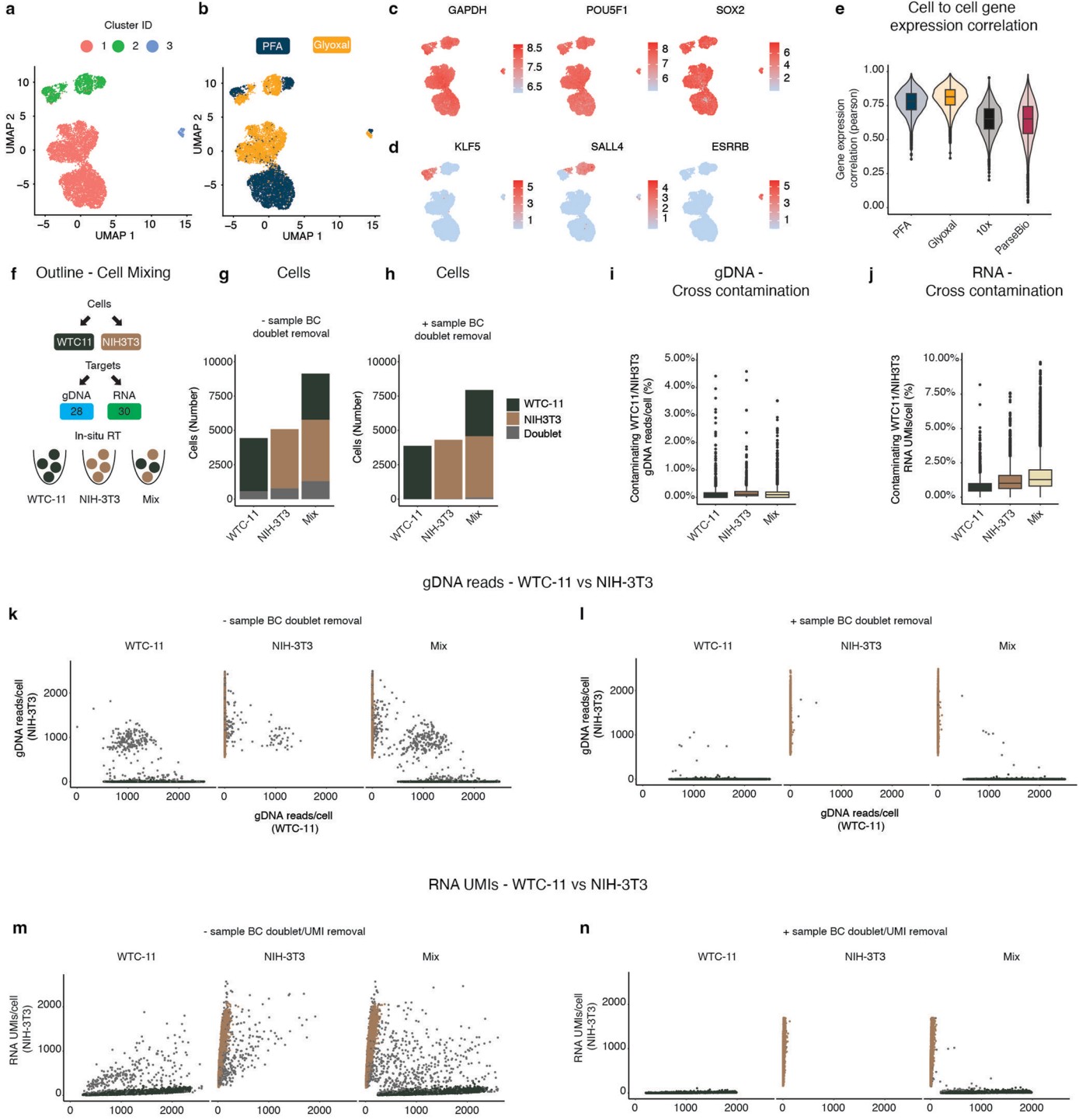

**Extended Data Fig. 2 | UMAPs and clustering of POP data. a**, UMAP clustering of POP SDR-seq data based on gene expression. **b**, Color coding of UMAP by fixation condition. **c**, UMAP plots of ubiquitously expressed genes *GAPDH*, *POU5F1* and *SOX2*. Color indicates normalized expression (log1p). **d**, UMAP plots of cluster-specifically expressed genes *KLF5*, *SALL4* and *ESRRB*. Color indicates normalized expression (log1p). **e**, Gene expression correlation (Pearson) between 100 subsampled cells comparing them individually against each other for SDR-seq, 10x Genomics and ParseBio. n = 4950 comparisons for SDR-seq, 10x Genomics and ParseBio. **f**, Outline of cell mixing experiment using either human (WTC-11)

or mouse (NIH-3T3) cells. During in situ RT cells were either used in individual wells or mixed together. **g**, **h**, Number of cells found for each species per in situ RT condition before (**g**) and after (**h**) doublet removal using sample BC information during analysis. **i**, **j**, Cross contamination of gDNA (**i**) and RNA (**j**) molecules between species for each in situ RT condition. n = 3865 cells (WTC-11), 4308 cells (NIH-3T3) and 7930 cells (Mix) from 1 independent SDR-seq experiment. **k**–**n**, Quality control plots to show either gDNA reads/cell (**k**, **l**) or RNA UMIs/cell (**m**, **n**) for each in situ RT condition before and after doublet removal and ambient RNA/UMI removal using sample BC information.

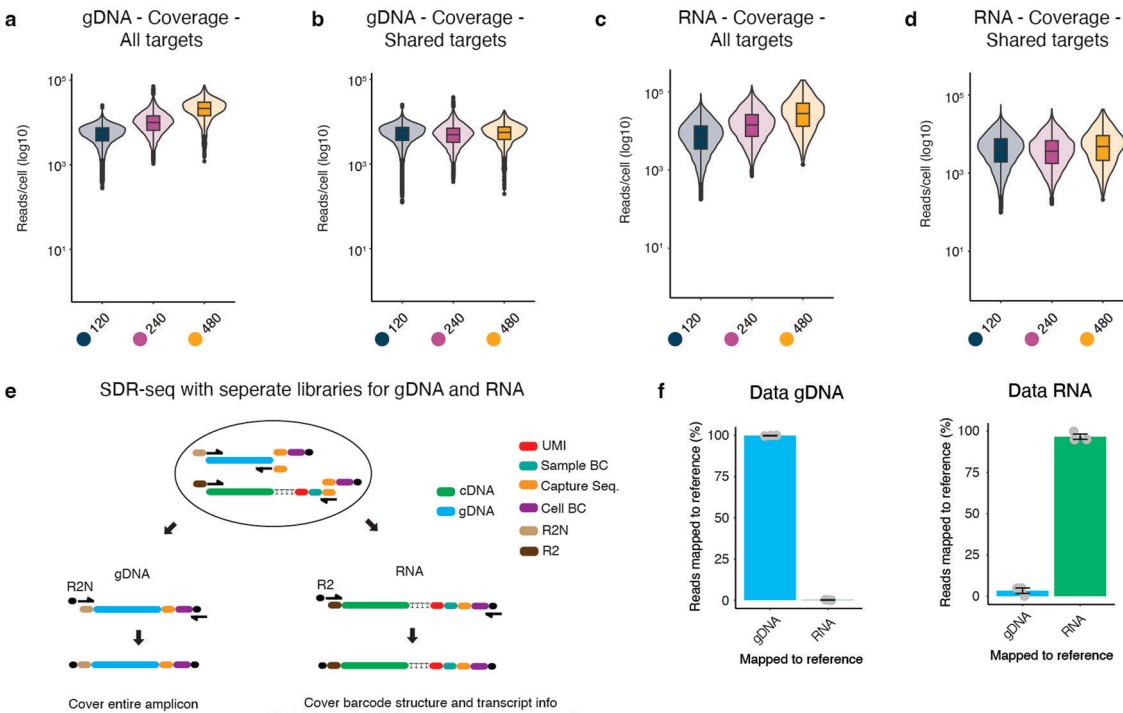

**Extended Data Fig. 3 | SDR-seq with separate library generation for RNA and gDNA targets. a–d**, Subsampled reads/cell for all gDNA (**a**) or RNA (**c**) targets or shared gDNA (**b**) or RNA (**d**) targets (**d**). n = 9680 cells (120 panel), 6610 cells (240 panel) and 804 cells (480 panel) from 1 independent SDR-seq experiment for each panel size testing. **e**, Overview of SDR-seq with separate library generation for RNA and gDNA. Distinct R2 (RNA) or R2N (gDNA) overhangs for each library. Sequencing ready libraries can be generated using specific library primers binding to R2 or R2N, respectively. **f**, Specificity of gDNA and RNA NGS libraries. Data from gDNA or RNA libraries was mapped to either gDNA or RNA references. Data points represent means ± SEM. n = 3 independent SDR-seq experiments.

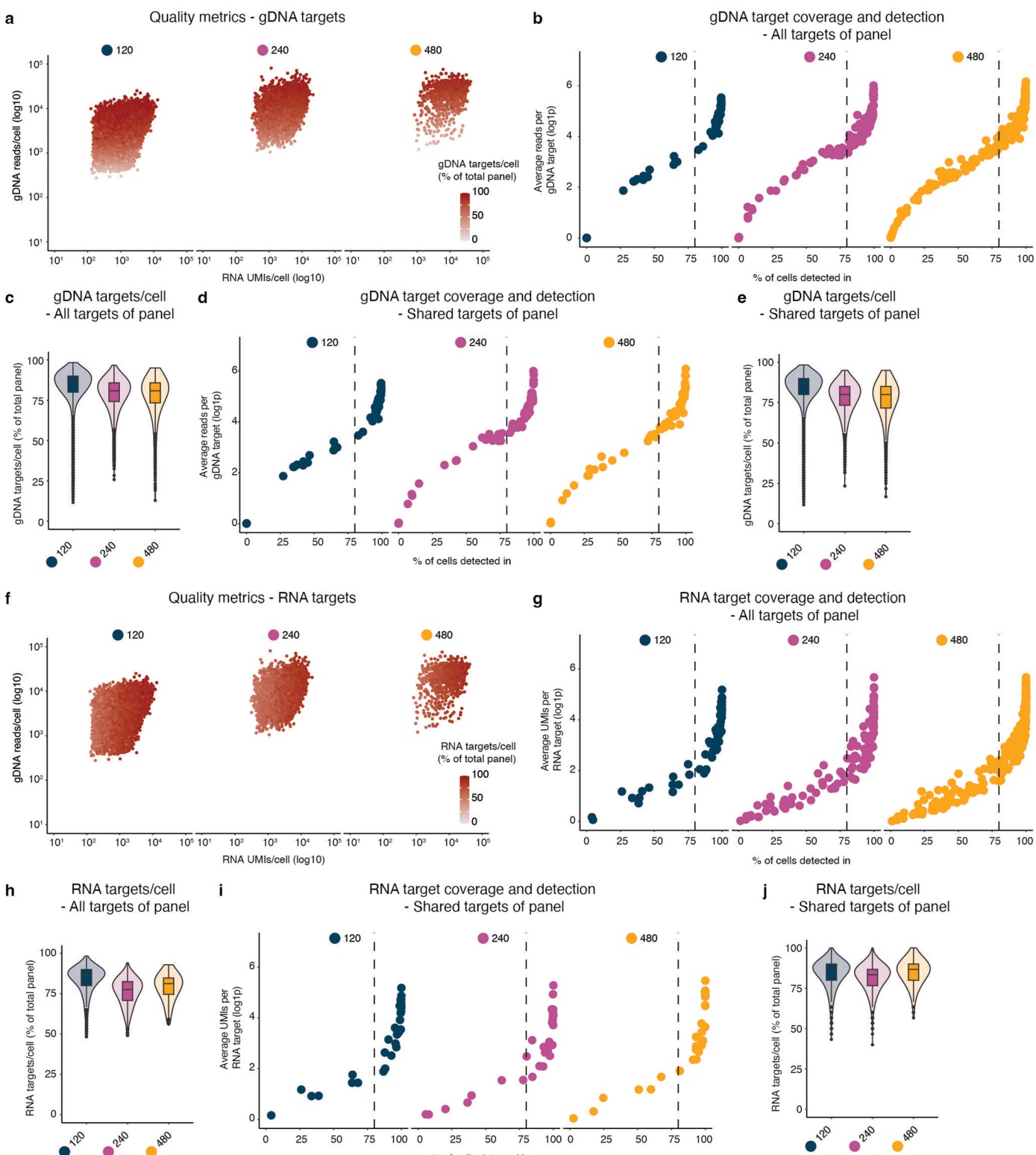

**Extended Data Fig. 4 | Metrics for target detection and coverage across differently sized target panels. a**, Quality metrics of panel size experiment. Color indicates fraction of gDNA targets/cell recovered. **b**, gDNA coverage and detection for all targets across panels tested. Dashed line indicates 80% detection. **c**, gDNA target detection per cell for all targets across panels tested. n = 9680 cells (120 panel), 6610 cells (240 panel) and 804 cells (480 panel) from 1 independent SDR-seq experiment for each panel size testing. **d**, gDNA coverage and detection for shared targets across panels tested. Dashed line indicates 80% detection. **e**, gDNA target detection per cell for shared targets across panels tested. n = 9680 cells (120 panel), 6610 cells (240 panel) and 804 cells (480 panel) from 1 independent SDR-seq experiment for each panel size testing. **f**, Quality metrics of panel size experiment. Color indicates fraction of RNA targets/cell recovered. **g**, RNA coverage and detection for all targets across panels tested. Dashed line indicates 80% detection. **h**, RNA target detection per cell for all targets across panels tested. n = 9680 cells (120 panel), 6610 cells (240 panel) and 804 cells (480 panel) from 1 independent SDR-seq experiment for each panel size testing. **i**, RNA coverage and detection for shared targets across panels tested. Dashed line indicates 80% detection. **j**, RNA target detection per cell for shared targets across panels tested. n = 9680 cells (120 panel), 6610 cells (240 panel) and 804 cells (480 panel) from 1 independent SDR-seq experiment for each panel size testing.

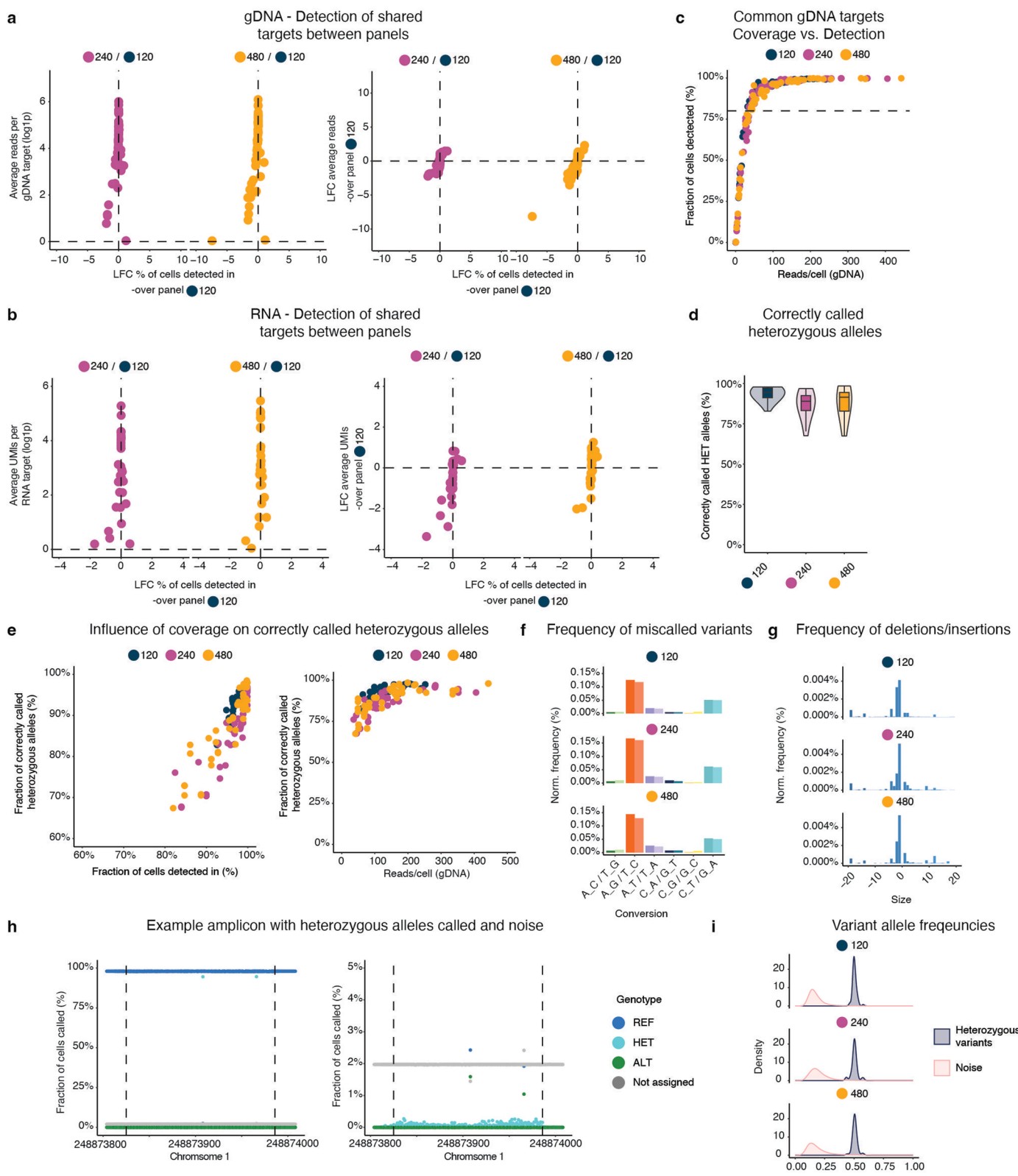

**Extended Data Fig. 5 | Comparison of target detection and coverage across differently sized target panels. a**, **b**, Comparison of coverage and detection for shared targets across panels tested for each gDNA (**a**) and RNA (**b**) target. LFC, log fold change (log2). **c**, Coverage and detection for shared targets across panels. Dashed line indicates 80% of cells detected in, which is the threshold used for categorizing highly covered amplicons that were used to determine allelic dropout (ADO). **d**, Percentage of correctly called heterozygous alleles across

panels on shared targets. n = 58 variants from 1 independent SDR-seq experiment for each panel size testing. **e**, Influence of detection and gDNA coverage on correctly called heterozygous alleles. **f**, **g**, Normalized frequency of miscalled variants across panels on shared targets for conversions (**f**) or deletions/insertions (**g**). **h**, Example gDNA target amplicon with indicated frequencies of variants. Dashed lines indicate primer binding sites. **i**, Density of variant allele frequencies for heterozygous variants and noise.

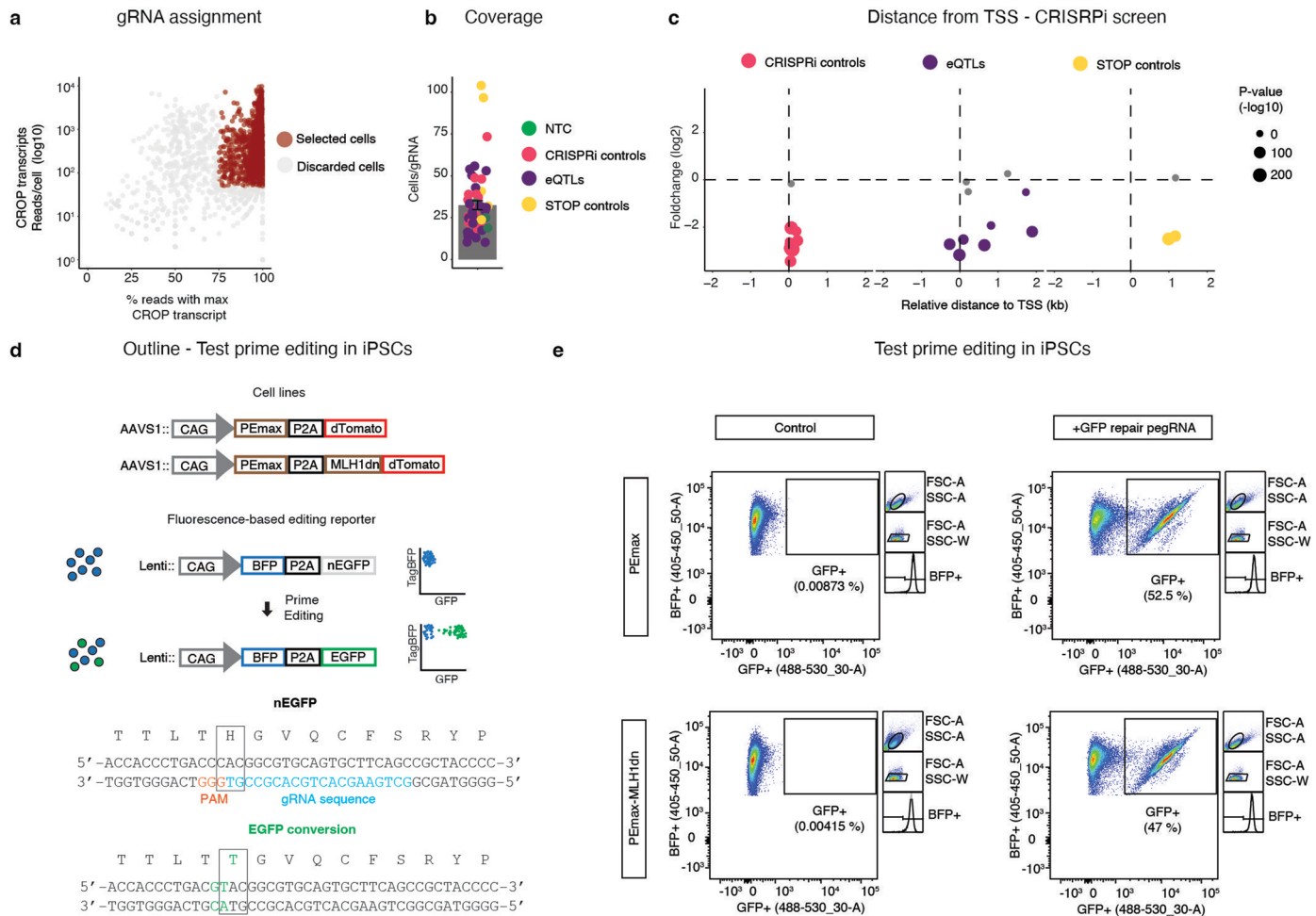

**Extended Data Fig. 6 | Quality metrics for CRISPRi screen and functional testing of PE iPSCs. a**, Overview of gRNA assignment for CRISPRi screen. **b**, Coverage for each gRNA in CRISPRi screen. Data points represent means ± SEM. n = individual gRNAs from 1 SDR-seq experiment. **c**, Relative distance of gRNA binding site to transcription start site (TSS). Positive values are after TSS (within transcript), negative values before transcript. Size indicates *P*-value calculated using MAST with Benjamini-Hochberg correction for multiple testing. Significant hits (*P*-value < 0.05) are colored. **d**, Outline of testing for PE iPSCs. Editing can be measured by repairing a non-functional EGFP that is integrated via a lentivirus. **e**, Flow cytometry indicating editing in PE iPSCs.

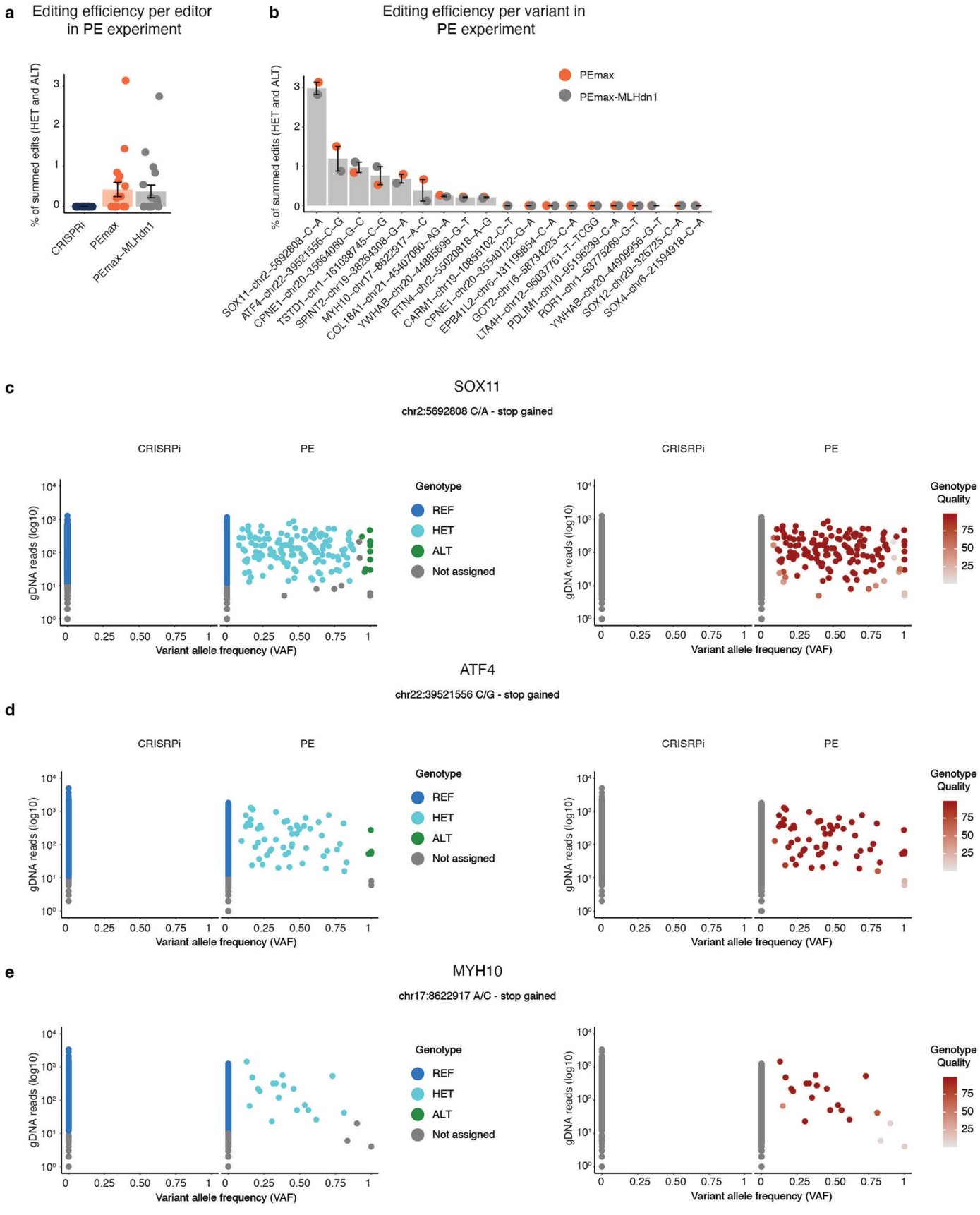

**Extended Data Fig. 7 | Editing efficiency and genotyping in PE screen. a**, Editing efficiency in PE screen. CRISPRi is indicated as a control. Data points represent means ± SEM. n = 19 variants from 1 CRISPRi SDR-seq experiment and 1 PEmax/ PEmax-MLHdn1 SDR-seq experiment. **b**, Editing efficiency for each locus that was assessed. Loci that had somatic alleles in the PE iPCSs that were either HET or ALT are not shown. Color indicates PE cell lines. Data points represent means ± SEM. n = 19 variants from 1 PEmax/PEmax-MLHdn1 SDR-seq experiment. **c**–**e**, Called genotypes and genotype quality vs. variant allele frequency (VAF) for *SOX11* (**c**), *ATF4* (**d**) and *MYH10* (**e**).

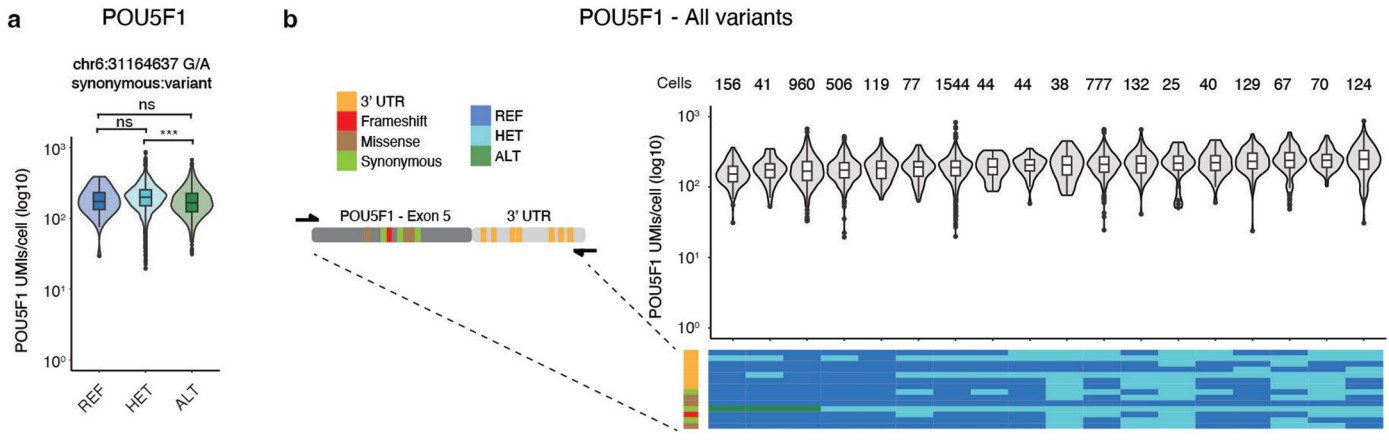

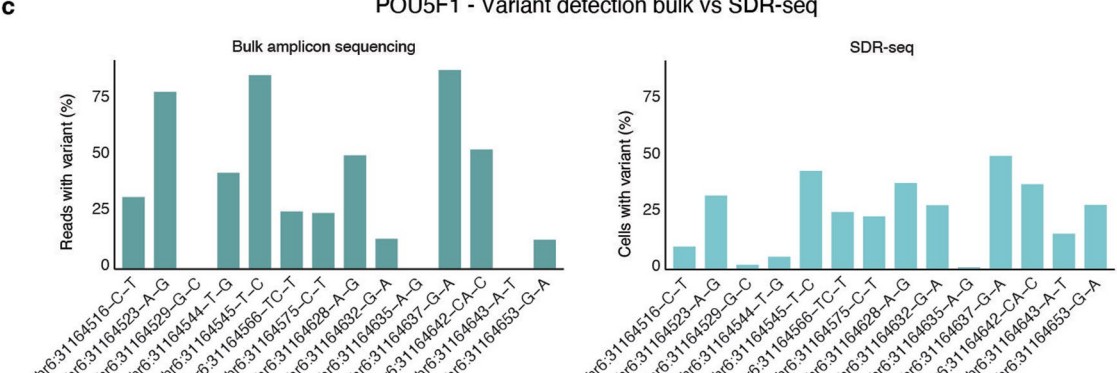

**Extended Data Fig. 8 | *POU5F1* locus in BE screen. a,** Intended edit to be introduced by base editing gRNA is shown at the *POU5F1* locus and its impact on gene expression. ***$P < 10^{-4}$, MAST with Benjamini-Hochberg correction. n = 46 cells (*POU5F1*:REF), 4172 cells (*POU5F1*:HET) and 1163 cells (*POU5F1*:ALT) from 1 SDR-seq experiment. *P*-value = 5.78x10$^{-20}$ (*POU5F1*: HET-ALT), **b,** All measured variants in combination along the measured gDNA site shown for *POU5F1* with their impact on gene expression. Number of cells are indicated on top. **c,** Comparison of variants at *POU5F1* locus identified with SDR-seq to bulk amplicon sequencing.

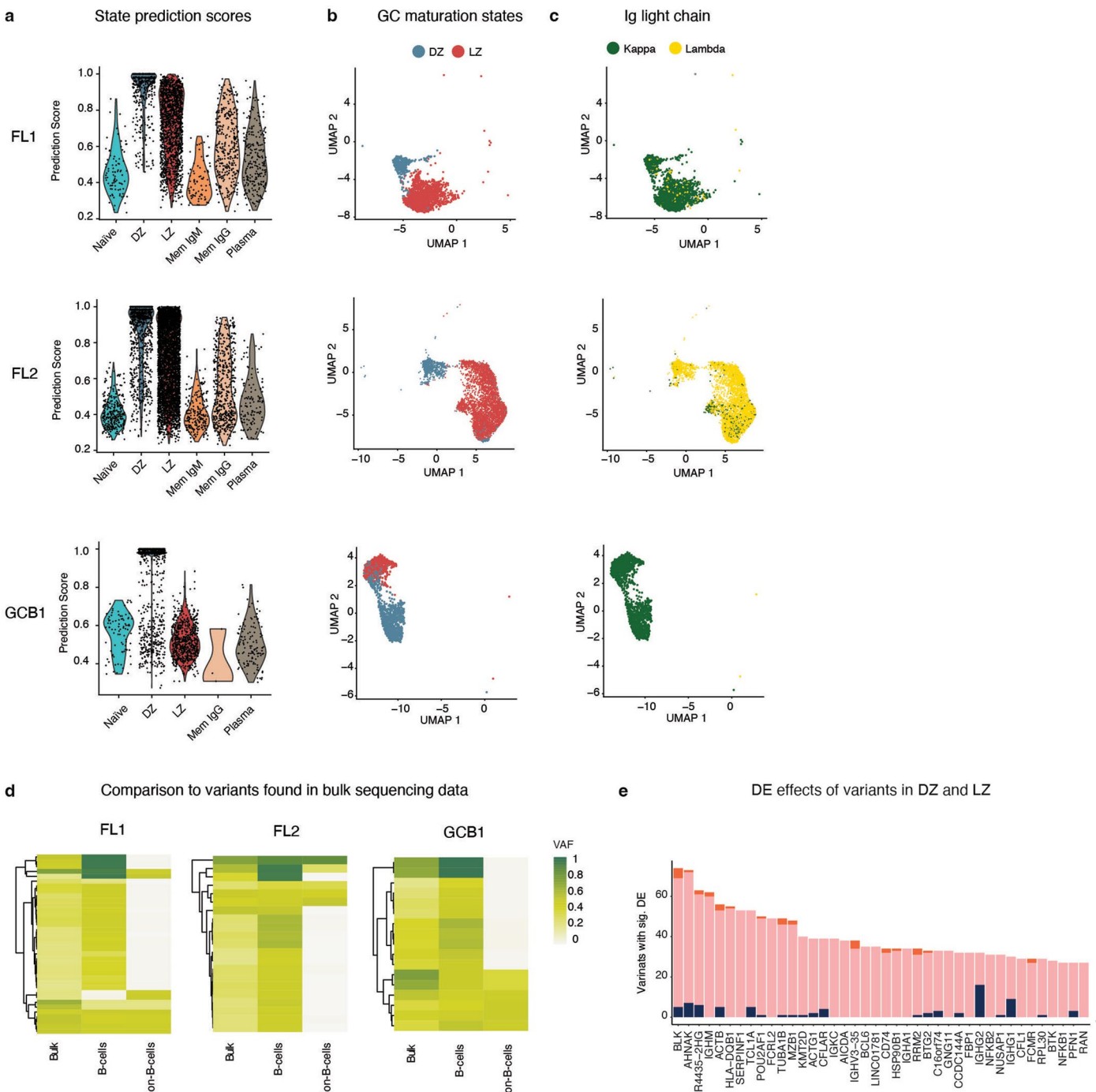

**Extended Data Fig. 9 | Assignment of maturation states in B-cell lymphoma patient samples, comparison of variant abundance and analysis of differentially expressed genes in DZ and LZ states in cells with and without variants. a**, Prediction scores of cell types assignment using a reference dataset[44,46]. Each cell is assigned a state, higher score indicates better assignment of this cell to the respective state. **b**, UMAP of cells belonging to the geminal center (GC) maturation states including DZ and LZ. Color indicates the state.

**c**, Ig light chain restriction. Color coded are either kappa or lamda Ig light chains. **d**, Comparison of variant allele frequencies (VAF) of variants found in bulk or in SDR-seq in either B-cells or non-B-cells. Color indicates VAF. **e**, Summed counts for differentially expressed (DE) genes between variant containing and non-containing cells within both DZ and LZ for most frequent variants for each patient.

# Reporting Summary

## Statistics

For all statistical analyses, confirm that the following items are present in the figure legend, table legend, main text, or Methods section.

| n/a | Confirmed | |
|---|---|---|
| ☐ | ☒ | The exact sample size (*n*) for each experimental group/condition, given as a discrete number and unit of measurement |
| ☒ | ☐ | A statement on whether measurements were taken from distinct samples or whether the same sample was measured repeatedly |
| ☐ | ☒ | The statistical test(s) used AND whether they are one- or two-sided<br>*Only common tests should be described solely by name; describe more complex techniques in the Methods section.* |
| ☒ | ☐ | A description of all covariates tested |
| ☐ | ☒ | A description of any assumptions or corrections, such as tests of normality and adjustment for multiple comparisons |
| ☐ | ☒ | A full description of the statistical parameters including central tendency (e.g. means) or other basic estimates (e.g. regression coefficient) AND variation (e.g. standard deviation) or associated estimates of uncertainty (e.g. confidence intervals) |
| ☐ | ☒ | For null hypothesis testing, the test statistic (e.g. *F*, *t*, *r*) with confidence intervals, effect sizes, degrees of freedom and *P* value noted<br>*Give P values as exact values whenever suitable.* |
| ☒ | ☐ | For Bayesian analysis, information on the choice of priors and Markov chain Monte Carlo settings |
| ☒ | ☐ | For hierarchical and complex designs, identification of the appropriate level for tests and full reporting of outcomes |
| ☐ | ☒ | Estimates of effect sizes (e.g. Cohen's *d*, Pearson's *r*), indicating how they were calculated |

*Our web collection on statistics for biologists contains articles on many of the points above.*

## Software and code

Policy information about availability of computer code

| Data collection | SDR-seq was performed using the Tapestri microfluidic device from Mission Bio. Flow cytometry was performed using BD Fortessa instruments running Diva (V9.0.1) software. Illumina sequencers (HiSeq and NextSeq) were used for NGS experiments. |
|---|---|
| Data analysis | Data analysis was performed using SDRranger (v1.0) to generate count/read matrices from RNA or gDNA NGS data (https://github.com/hawkjo/SDRranger). Code for TAP-seq prediction, generation of custom STAR references and processing of the data is available under https://github.com/DLindenhofer/SDR-seq. Packages used were AnnotationDbi (1.64.1), BiocManager (1.30.25), BiocParallel (1.36.0), biomaRt (2.58.2), Biostrings (2.70.3), BSgenome (1.70.2), BSgenome.Mmusculus.UCSC.mm10 (1.4.3), cardelino (1.4.0), circlize (0.4.16), data.table (1.17.0), dplyr (1.1.4), future.apply (1.11.3), GenomicRanges (1.54.1), ggpattern (1.1.4), ggplot2 (3.5.2), here (1.0.1), knitr (1.50), Matrix (1.6.5), org.Hs.eg.db (3.18.0), patchwork (1.3.0), pcaMethods (1.94.0), pheatmap (1.0.12), purrr (1.0.4), RColorBrewer (1.1.3), readr (2.1.5), readxl (1.4.5), reshape2 (1.4.4), Rmisc (1.5.1), rtracklayer (1.62.0), scales (1.4.0), Seurat (5.3.0), stringr (1.5.1), TAPseq (1.14.1), tibble (3.2.1), tidyr (1.3.1), tidyverse (2.0.0), topGO (2.54.0), STAR (2.7.11a), Python (3.8.0), GATK HaplotypeCaller (4.2.3.0), sinto (0.10.0) and samtools (1.17). |

For manuscripts utilizing custom algorithms or software that are central to the research but not yet described in published literature, software must be made available to editors and reviewers. We strongly encourage code deposition in a community repository (e.g. GitHub). See the Nature Portfolio guidelines for submitting code & software for further information.

## Data

Policy information about availability of data

All manuscripts must include a data availability statement. This statement should provide the following information, where applicable:
- Accession codes, unique identifiers, or web links for publicly available datasets
- A description of any restrictions on data availability
- For clinical datasets or third party data, please ensure that the statement adheres to our policy

Sequencing data and processed data for non-primary human data is available on GEO under accession number GSE268646. Sequencing data and processed data for primary human data is available on EGA under study number EGAS50000000374 and dataset ID EGAD50000000551. The dataset on EGA is read-only under ega-archive.org/datasets/EGAD50000000551. Access to the data will be granted for appropriate use in research and will be governed by the provisions laid out in the terms contained in the Data Access Agreement. Variant information for WTC-11 human iPSCs was downloaded from UCSC (https://s3-us-west-2.amazonaws.com/downloads.allencell.org/genome-sequence/AH77TTBBXX_DS-229105_GCCAAT_recalibrated.vcf.gz). Candidate cis-regulatory elements (cCRE) for five human iPSC lines (H1, H7, H9, iPS DF 6.9, iPS DF 19.11) were obtained from SCREEN (https://screen.encodeproject.org). iPSCs data for ParseBio data was obtained from https://www.parsebiosciences.com/customer-datasets/multi-omics-approach-for-near-full-length-human-ipsc-transcriptomes-in-cardiomyocyte-models/#download. NIH-3T3 data for TAP-seq primer prediction was obtained from https://www.10xgenomics.com/datasets/500-1-1-mixture-of-human-hek-293-t-and-mouse-nih-3-t-3-cells-3-lt-v-3-1-chromium-x-3-1-low-6-1-0. WTC-11 data for TAP-seq primer prediction and comparison of gene expression variance was obtained from https://www.ebi.ac.uk/biostudies/arrayexpress using accession number E-MTAB-6687. DNA sequences for custom gDNA and RNA references used for alignment were obtained in R using the BSgenome (1.70.2) package utilizing the "BSgenome.Hsapiens.UCSC.hg38" genome" (genome hg38, based on assembly GRCh38.p14 since 2023/01/31).

## Research involving human participants, their data, or biological material

Policy information about studies with human participants or human data. See also policy information about sex, gender (identity/presentation), and sexual orientation and race, ethnicity and racism.

| | |
|---|---|
| Reporting on sex and gender | No sex or gender-based analysis were performed as they were not relevant in this study. Informed consent from every patient was gathered beforehand to collect gender data and was determined based on self-reporting. Gender is reported in the EGA repository in the metadata. |
| Reporting on race, ethnicity, or other socially relevant groupings | This information was not collected. Therefore not applicable. |
| Population characteristics | Informed consent from every patient was gathered beforehand regarding age, diagnosis and treatment. None of this metadata is taken in consideration in the data analysis in this manuscript. Diagnosis was either follicular lymphoma (FL1 and FL2) or germinal center subtype diffuse large B-cell lymphoma (GCB1). Age at time of sampling was 59 (Fl1), 74 (Fl2) and 45 (GCB1). Sex was female for FL1 and male for FL2 and GCB1. Ann-Arbor clinical stage at time of sampling was IIIA (Fl1), IVA (Fl2) and IVB (GCB1). Relapse status at time of sampling was diagnosis (FL1 and GCB1) and relapse (Fl2). |
| Recruitment | Recruitment for this retrospective study was done from suitable biobanked material at University Hospital Heidelberg. |
| Ethics oversight | The study (S-254/2016) was approved by University of Heidelberg's Ethics Committee. We obtained informed consent from every patient beforehand. |

Note that full information on the approval of the study protocol must also be provided in the manuscript.

# Field-specific reporting

Please select the one below that is the best fit for your research. If you are not sure, read the appropriate sections before making your selection.

☒ Life sciences ☐ Behavioural & social sciences ☐ Ecological, evolutionary & environmental sciences

For a reference copy of the document with all sections, see nature.com/documents/nr-reporting-summary-flat.pdf

# Life sciences study design

All studies must disclose on these points even when the disclosure is negative.

| | |
|---|---|
| Sample size | No statistical methods were used to predetermine sample sizes. Instead, sample sizes were guided by the technical limitations of the Tapestri microfluidic device, which yields approximately 9,000 cells per run. This throughput is consistent with other widely used single-cell microfluidic platforms, such as 10x Genomics. For each experiment, the number of cells analyzed was chosen to be sufficient for qualitative and comparative assessment of assay performance and biological signal detection in the context o method development. <br><br> The manuscript primarily describes SDR-seq, and the experimental design focuses on demonstrating technical feasibility, robustness, and versatility. Robustness was shown by reproducing the assay across multiple independent runs, yielding reproducible combined single-cell readouts of gDNA and RNA in the same cell. Fixation condition effects were assessed in a multiplexed design within a single run, allowing direct cell-to-cell comparison under identical processing conditions. Primer panel size effects were evaluated in separate SDR-seq runs due to |

the need for distinct PCR panels, and sufficient cells were obtained in each run to assess performance and coverage.

In perturbation experiments, the observed low editing efficiency limited the ability to interpret a large number of eQTLs, but the data generated were sufficient to highlight this challenge and inform future optimization. Finally, samples from three B-cell lymphoma patients were processed across two independent SDR-seq runs. The sample size was adequate to demonstrate the applicability of the method , performing differential abundance testing of variants and differential gene expression analysis comparing distinct subclasses of cells within each patient.

Overall, sample sizes were selected to balance the throughput limits of the technology with the goal of establishing method feasibility, reproducibility, and practical use cases. The number of cells and samples per experiment was sufficient to achieve these aims.

| | |
|---|---|
| Data exclusions | Low quality cells were removed for downstream processing. Detailed thresholds set for each experiment can be found in https://github.com/DLindenhofer/SDR-seq. A second GCB sample was intended to be analyzed in this study. Low viability after thawing and dead cell removal prohibited the inclusion of this sample. |
| Replication | SDR-seq was shown to work in two different fixation conditions and across different panel sizes, perturbation assays and sample types. All attempts of performing SDR-seq as described in the manuscript were successful. Overall these were 10 independent SDR-seq runs. |
| Randomization | Not applicable in this study as experiments doing comparative analysis were assayed in a pooled setting. |
| Blinding | The experimenters were not blinded. Experimental procedures were automated and standardized, and data analyses were carried out predominantly using computational pipelines without manual intervention. The reported results are primarily descriptive and based on objective readouts such as sequencing metrics, read counts, and computationally derived molecular profiles. As such, the experimenter had no opportunity to influence the outcomes. |

# Reporting for specific materials, systems and methods

We require information from authors about some types of materials, experimental systems and methods used in many studies. Here, indicate whether each material, system or method listed is relevant to your study. If you are not sure if a list item applies to your research, read the appropriate section before selecting a response.

## Materials & experimental systems

| n/a | Involved in the study |
|---|---|
| ☒ | ☐ Antibodies |
| ☐ | ☒ Eukaryotic cell lines |
| ☒ | ☐ Palaeontology and archaeology |
| ☒ | ☐ Animals and other organisms |
| ☒ | ☐ Clinical data |
| ☒ | ☐ Dual use research of concern |
| ☒ | ☐ Plants |

## Methods

| n/a | Involved in the study |
|---|---|
| ☒ | ☐ ChIP-seq |
| ☐ | ☒ Flow cytometry |
| ☒ | ☐ MRI-based neuroimaging |

## Eukaryotic cell lines

Policy information about cell lines and Sex and Gender in Research

| | |
|---|---|
| Cell line source(s) | The HEK293 line was purchased from ATCC (CRL-3216). The WTC-11 iPSCs (GM25256)were purchased from the Coriell Institute for Medical Research. The NIH-3T3 cell line was purchased from DSMZ (ACC 59) |
| Authentication | None of the cell lines were independently authenticated. |
| Mycoplasma contamination | Cell cultures were routinely (every three months) tested and confirmed negative for mycoplasma. |
| Commonly misidentified lines (See ICLAC register) | No commonly misidentified lines have been used in this study. |

# Plants

Seed stocks | NA

Novel plant genotypes | NA

Authentication | NA

# Flow Cytometry

## Plots

Confirm that:

☒ The axis labels state the marker and fluorochrome used (e.g. CD4-FITC).

☒ The axis scales are clearly visible. Include numbers along axes only for bottom left plot of group (a 'group' is an analysis of identical markers).

☒ All plots are contour plots with outliers or pseudocolor plots.

☒ A numerical value for number of cells or percentage (with statistics) is provided.

## Methodology

Sample preparation | For flow cytometry analysis iPSCs were prepared in a single cell suspension using Accutase (StemCell Technologies - #07922). This was followed by filtering through a 35 μm cell strainer.

Instrument | BD Fortessa

Software | FACS Diva

Cell population abundance | A minimum of 20000 single cells was analyzed for each condition at each timepoint.

Gating strategy | Single cells were gated using forward and side scatters. Amplifier settiings were chosen to clearly display negative and positive populations. Gating strategies are provided in the Extended Data Fig. 6e directly next to each flow cytometry plot.

☒ Tick this box to confirm that a figure exemplifying the gating strategy is provided in the Supplementary Information.

