## [Peer Review File · Nature Methods]

Functional phenotyping of genomic variants using multiomic scDNA-scRNA-seq

Corresponding Author: Professor Lars Steinmetz

Version 0:

Decision Letter:

1st Sep 2024

Dear Dr Steinmetz,

Your Article, "Functional phenotyping of genomic variants using multiomic scDNA-scRNA-seq", has now been seen by 3 reviewers. As you will see from their comments below, although the reviewers find your work of considerable potential interest, they have raised a number of concerns. We are interested in the possibility of publishing your paper in Nature Methods, but would like to consider your response to these concerns before we reach a final decision on publication.

We therefore invite you to revise your manuscript to address these concerns, including the technical aspects of validations and data quality measurements.

Link Redacted

We hope to receive your revised paper within 12 weeks. If you cannot send it within this time, please let us know. In this event, we will still be happy to reconsider your paper at a later date so long as nothing similar has been accepted for publication at Nature Methods or published elsewhere.

OPEN SCIENCE REQUIREMENTS

REPORTING SUMMARY AND EDITORIAL POLICY CHECKLISTS

Reporting summary: <https://www.nature.com/documents/nr-reporting-summary.zip>
Editorial policy checklist: <https://www.nature.com/documents/nr-editorial-policy-checklist.zip>

IMAGE INTEGRITY

DATA AVAILABILITY

All novel DNA and RNA sequencing data, protein sequences, genetic polymorphisms, linked genotype and phenotype data, gene expression data, macromolecular structures, and proteomics data must be deposited in a publicly accessible database, and accession codes and associated hyperlinks must be provided in the "Data Availability" section.

CODE AVAILABILITY

Please include a "Code Availability" subsection in the Online Methods which details how your custom code is made available. Only in rare cases (where code is not central to the main conclusions of the paper) is the statement "available upon request" allowed (and reasons should be specified).

For more information on our code sharing policy and requirements, please see: <https://www.nature.com/nature-research/editorial-policies/reporting-standards#availability-of-computer-code>

SUPPLEMENTARY PROTOCOL

To help facilitate reproducibility and uptake of your method, we ask you to prepare a step-by-step Supplementary Protocol for the method described in this paper. We [encourage authors to share their step-by-step experimental protocols](https://www.nature.com/nature-research/editorial-policies/reporting-standards#protocols) on a protocol sharing platform of their choice and report the protocol DOI in the reference list. Nature Portfolio's protocols.io is a free-to-use and open resource for protocols; protocols deposited onto protocols.io are citable and can be linked from the published article. More details can found at [protocols.io](https://www.protocols.io/help/publish-articles).

ORCID

Sincerely,
Lei

Lei Tang, Ph.D.
Senior Editor
Nature Methods

Reviewers' Comments:

Reviewer #1 (Remarks to the Author):

In the manuscript entitled "Functional phenotyping of genomic variants using multiomic scDNA-scRNA-seq" by Lindenhofer et al., the authors reported their new method, scDR-seq, to target sequence up to 480 DNA and/or RNA (cDNA) sequences from the same single cells from a large number (over thousands) of single cells per sample (per analysis). I think the method is novel and importantly could be very useful for investigators in the field of understanding tumor heterogeneity, especially the heterogeneity in gene expression regulation. However, I have some comments and suggestions as follows:

1. Regarding mutations in gDNA, what is the minimum number of cells required to reliably detect a de novo mutation? My concern is that PCR-based amplification is relatively error-prone, and scDR-seq may not be able to detect de novo mutations occurring uniquely in one cell, which is acceptable. However, the minimum number of cells needs to be specified.
2. For gDNA, what is the maximum number of base pairs per cell that can be covered? In the context of a typical cancer driver gene panel, what fraction of base pairs can be covered by scDR-seq?
3. For RNA, what fraction of genes, which are typically considered important to a tumor cell, can be covered by scDR-seq?
4. Finally, for RNA, what is the pair-wise correlation in expression levels observed by scDR-seq between any two cells of the same genotype? How does this compare to other droplet-based or non-droplet-based scRNA-seq methods?

--
Xiao Dong

Reviewer #2 (Remarks to the Author):

The study by Lindenhofer et al., reports a significant technology advancement for single cell targeted DNA and RNA co-assay with moderate throughput and improved coverage.

Technology-wise, the new method allows recovery of 80% of all gDNA targets in more than 80% of cells across all panels, which hold high promise as a technical solution to link genotype and gene expression in pooled screens. This opens doors to a wide range of applications in certain cell lines, although generalizability to other systems may need substantial further work. A different fixative glyoxal, is tested with the commonly used PFA and an improved sensitivity observed for RNA recovery is informative and helpful to the scRNA-seq field.

My major concern of the current manuscript is with the design for one of the proof-of-principle application linking mutational burden with gene expression. Single-cell DNA sequencing in general is subject to very high false positive SNV calls, and one of the major reasons is spontaneous deamination in the amplification process. I suggest that the authors validate the results by adding a USER (NEB) enzyme step (a mixture of UDG and endonuclease VIII) immediately prior to amplification.

Otherwise, the study is carefully conducted and results are clearly presented.

Reviewer #2 (Remarks on code availability):

The code works for the example file.

Reviewer #3 (Remarks to the Author):

The authors describe SDR-Seq as a novel droplet-based single-cell genomics technique to investigate eQTL in functional genomic screens and tumor heterogeneity at high throughput. Using the established Tapestry platform, by carrying out glyoxal-based cell fixation and in-situ reverse transcription, it enables targeted genotyping and targeted gene expression analyses, including linking of different variants to distinct gene expression changes. Preliminary validation experiments are carried out in cell lines and primary samples within the context of B-cell lymphomas. This technique is claimed to allow interrogation of several hundred targets at both gDNA and RNA level in parallel.

There is clearly an unmet need for techniques that allow high throughput single cell genotyping and gene expression analysis. Overall the data presented convincingly establishes that it is possible to generate targeted gene expression data and gDNA sequencing in individual cells. However, the validation of the technique is incomplete and application to uncover new biology very preliminary and unconvincing.

With regards to additional QC and validation data it would be helpful to understand the cell yield and doublet rate by loading variable numbers of cells and with cell mixing experiments to definitively establish doublet rates and cross-contaminating reads which seems quite high; is it correct that 5% of reads mapped to the incorrect sample barcode (this is high)?

What is the minimum number of cells required for this technique? This is important as cell number is often limiting from primary patient samples.

Can this technique be used to amplify the mitochondrial genome, which would be helpful for clonal tracing.

Have the authors tested whole transcriptome amplification using smart-seq template switch primers. This should be tested and presented even if it does not work.

The gDNA coverage seems quite low even when using a very low bar of only 5 reads per cell to define detection (which is inadequate for genotyping). What is the overall allelic dropout rate and how does this benchmark for the same gDNA loci for the standard Tapestry method and plate based techniques for parallel gDNA and GE analysis? This can be assessed by studying known heterozygous SNPs. What is driving the 'failure' for detection of approx. 20% of gDNA loci? More information on panel design for RNA and gDNA and how this influences detection would be helpful.

The prime editing is extremely inefficient making it impossible for validation of the technique for the primary purpose outlined in the abstract. The authors need to demonstrate that their technique can be used to reliably detect altered expression of a gene following PE of a range of eQTLs with known effect on gene expression (or not). Demonstrating that their technique can identify alterations in gene expression following CRISPRi or introduction of a STOP codon is a very low bar and insufficient for validation purposes.

The lymphoma data is very preliminary and is presented in a way that is unclear. The authors should sequence a larger number of tumors in bulk, determine the key mutations associated with the tumors and then apply their technique to show that variants can be reliably detected in B-cells (and absent from other cell lineages that are not part of the tumor). The impact of subclonal mutations on gene expression then be presented.

Version 1:

Decision Letter:

22nd Jan 2025

Dear Dr Steinmetz,

Thank you for your letter detailing how you would respond to the reviewer concerns regarding your Article, "Functional phenotyping of genomic variants using multiomic scDNA-scRNA-seq". We have decided to invite you to revise your manuscript as you have outlined, before we reach a final decision on publication.

Kindly take note of the editorial requirements mentioned in the [parentheses] alongside reviewer #3's comments. Please do not

hesitate to contact me if you have any questions or would like to discuss these revisions further.

Link Redacted

We hope to receive your revised paper within 4 weeks. If you cannot send it within this time, please let us know. In this event, we will still be happy to reconsider your paper at a later date so long as nothing similar has been accepted for publication at Nature Methods or published elsewhere.

OPEN SCIENCE REQUIREMENTS

REPORTING SUMMARY AND EDITORIAL POLICY CHECKLISTS

IMAGE INTEGRITY

EXTENDED DATA FIGURES

DATA AVAILABILITY

Please include a “Data availability” subsection in the Online Methods. This section should inform readers about the availability of the data used to support the conclusions of your study, including accession codes to public repositories, references to source data that may be published alongside the paper, unique identifiers such as URLs to data repository entries, or data set DOIs, and any other statement about data availability. At a minimum, you should include the following statement: “The data that support the findings of this study are available from the corresponding author upon request”, describing which data is available upon request and mentioning any restrictions on availability. If DOIs are provided, please include these in the Reference list (authors, title, publisher (repository name), identifier, year). For more guidance on how to write this section please see: <http://www.nature.com/authors/policies/data/data-availability-statements-data-citations.pdf>

CODE AVAILABILITY

Please include a “Code Availability” subsection in the Online Methods which details how your custom code is made available. Only in rare cases (where code is not central to the main conclusions of the paper) is the statement “available upon request” allowed (and reasons should be specified).

MATERIALS AVAILABILITY

SUPPLEMENTARY PROTOCOL

To help facilitate reproducibility and uptake of your method, we ask you to prepare a step-by-step Supplementary Protocol for the method described in this paper. We [encourage authors to share their step-by-step experimental protocols](https://www.nature.com/nature-research/editorial-policies/reporting-standards#protocols) on a protocol sharing platform of their choice and report the protocol DOI in the reference list. Nature Portfolio's protocols.io is a free-to-use and open resource for protocols; protocols deposited onto protocols.io are citable and can be linked from the published article. More details can found at [protocols.io](https://www.protocols.io/help/publish-articles).

ORCID

Nature Methods is committed to improving transparency in authorship. As part of our efforts in this direction, we are now requesting that all authors identified as ‘corresponding author’ on published papers create and link their Open Researcher and Contributor Identifier (ORCID) with their account on the Manuscript Tracking System (MTS), prior to acceptance. This applies to primary research papers only. ORCID helps the scientific community achieve unambiguous attribution of all scholarly contributions. You can create and link your ORCID from the home page of the MTS by clicking on ‘Modify my Springer Nature account’. For more information please visit [please visit](http://www.springernature.com/orcid) www.springernature.com/orcid.

Sincerely,
Lei

Lei Tang, Ph.D.
Senior Editor
Nature Methods

Reviewers' Comments:

Reviewer #1 (Remarks to the Author):

My question #4 was not directly addressed.

Line 343: "These methods enable a whole genome ..." MDA and PTA are primarily used for de novo SNV detection. I do not think that PCR-based methods, including SDR-seq and the MissionBio system, can effectively serve this purpose. This is due to the significantly higher error rate of PCR—several orders of magnitude greater than the rate at which de novo mutations occur at the single-cell level. In fact, the noise variants, typically centered around a VAF of ~0.1%, pose a detection limit for these methods (as the authors showed in the revision). In context, high-depth bulk WES (2000x) can reliably detect true de novo SNVs (rather than artifacts) around this same VAF threshold. For this reason, I don't think SDR-seq is well-suited for de novo SNV detection, but SDR-seq would be important for other purposes as demonstrated by the authors. The above could be addressed by revising the discussions in the manuscript.

Line 347: "SDR-seq offers significantly improved throughput (~100-fold) compared to PTA-based ..." This statement is incorrect. SDR-seq covers only 40 kb of the genome, whereas PTA can cover almost the entire 3 Gb human genome per single cell. Even at a scale of 9,000 cells processed via SDR-seq, the total genome coverage is only ~10% of a single-cell genome, which PTA can achieve. Therefore, SDR-seq does not offer comparable throughput in terms of genome coverage. However, again, I don't think this was why SDR-seq was developed for, and could be addressed by revising the discussions.

Reviewer #2 (Remarks to the Author):

The authors have addressed all my comments. I agree with their reasons for not doing the suggested experiments with USER nicking at the deamination sites. The additional analyses are careful and thorough.

Reviewer #3 (Remarks to the Author):

Largely the authors have not addressed my previous comments which is disappointing as I don't think these suggestions were unreasonable

1. Cell mixing experiments were not done using the established conditions with glyoxal i.e. a mix of cells within the same starting well that can then be genetically demultiplexed to establish definitively the doublet rate using the glyoxal fixation conditions. This is important information for anyone wishing to use the technique. The issue with cross contaminating reads is not dealt with and again, a cell mixing experiment will establish cross-contamination between cells (for RNA and potentially DNA) using their technique. [the reviewer asks for the cross-contamination between cells, not the contamination from ambient RNA.]

2. I asked what the minimum number of cells is for their technique. The authors respond by describing the cell numbers they used in the experiments they had already carried out, which does not answer the question. This requires additional experimentation to titrate down cell numbers and establish a reasonable window of cell input using this approach. This is a major issue with primary cell material and important information for a method paper. [please provide information on the minimum number of cells required for at least one of the demonstrations presented in the paper.]

3. Perhaps I should have been more specific here in the way this comment was phrased. Of course, I appreciate that in principle the technique could be used to amplify mitochondrial genomes, I was rather expecting/requesting that the authors carry out some additional experimentation to demonstrate this approach using their method (which would also in parallel provide some validation of the veracity of mutation calling). [we would not request an additional experiment on mitochondrial genomes.]

4. Whole transcriptome analysis would increase my enthusiasm for this method a great deal. It is a shame that this did not work. I think it is helpful to include this in the manuscript as it is something quite obvious that users will be interested in trying and may waste a lot of time repeating experiments already conducted. [please discuss this limitation in the paper.]

5. The ADO seems high. The authors are biasing their analysis by only selecting amplicons with a high coverage (defined as detected in >80% of the cells), which if I understand correctly is only 80% of the amplicons. To directly compare with Tapestry ADO it should be easy to simply include gDNA primers for a run to account for the issue with panel size the authors are concerned about. As things stand, it is very possible that their method increases ADO rate in comparison with a standard Tapestry method, and that this could be optimised. There are many reasons why their method which permeabilizes the cells might increase ADO and this should be tested.

6. The manuscript sets out that the primary aim is to establish a method to systemically study impact of genome variants on gene expression. What I am suggesting is that they address this with some (a few e.g. 5) well-established eQTLs occurring in functional genome regulatory elements that are already proven to be functional, of which there are many examples e.g. rs2836882 risk allele has a clear effect on ETS2 expression in bulk analysis as recently reported in Nature. The authors should select a small number of such well validated function SNPs found at risk alleles and validate that upon editing they can detect the expected change in gene expression in the relevant cell type. This will provide robust support to their overarching aim that their technique can be used to systematically study genome variation. Identifying a change in gene expression by knocking down a gene with CRISPRi (or STOP codon) is a very low bar and does not really test what the authors themselves set out to address in this manuscript. I do not think the data in 3K address this crucial point. [If it's not too time-consuming, it would be helpful to include an example of validated SNPs.]

7. I suggested analysis of a larger patient cohort which was not done. The primary patient analysis remains preliminary with only three patients analysed. What are the specific driver mutations they analysed (CREBBP, KMT2D, EZH2? Etc). Was detection of any specific individual driver mutation associated with distinct gene expression changes. Do the identified clones allow a clonal structure to be inferred and in the non-B-cells can a pre-lymphoma clone be identified? How does the ADO impact this? Integration with mitochondrial genome analysis would provide an extra layer of validation here (see point 3). Are

there any non-mutant B-cells. Are these different from normal healthy donor B-cells? etc. Overall, in my view, the data as presented does not convincingly validate or illustrate the utility of this technique. The power of combined gene expression and mutation analysis in individual cells to analyse primary patient material is that it allows an intra-patient comparison of WT versus mutant cells within the same cell type; it would be helpful if some illustrative examples could be presented in a larger patient cohort. [we would not require a larger patient cohort analysis.]

I think the authors have developed a really interesting technique but there are obvious and important outstanding QC and validation experiments that in my view should be carried out before this is published as a top tier methods paper.

Version 2:

Decision Letter:

Our ref: NMETH-A56850B

19th May 2025

Dear Dr. Steinmetz,

I sincerely apologize for the delayed review process. Thank you for submitting your revised manuscript "Functional phenotyping of genomic variants using multiomic scDNA-scRNA-seq" (NMETH-A56850B). We have now consulted with all original referees about the concerns raised by reviewer #3. The reviewers find that the paper has improved in revision, and therefore we'll be happy in principle to publish it in Nature Methods, pending minor revisions to satisfy the referees' final requests and to comply with our editorial and formatting guidelines. Please focus on clarifying the technical points raised by Reviewer#1.

TRANSPARENT PEER REVIEW

ORCID

Sincerely,
Lei

Lei Tang, Ph.D.
Senior Editor
Nature Methods

Reviewer #1 (Remarks to the Author):

The authors have addressed all of my concerns as Reviewer #1.

Regarding the concerns raised by Reviewer #3:

1. The response is acceptable.

2. Although the two links provided in the response do not work for me, the number of input cells used in each experiment is specified in the revised manuscript. The reported input cell numbers are acceptable.
3. I agree with the authors' response.
4. I agree with the authors' response, although I note that a whole-transcriptome assay would indeed be more innovative—albeit significantly more costly.
5. The response is acceptable.
6. In this case, I share the concerns raised by Reviewer #3. The evidence presented in Figure 3g may not be sufficiently convincing. Some key information appears to be missing:
 - Are the 56 high-confidence eQTLs known causal variants?
 - For the known causal variants, are the expression changes reported in Figure 3g consistent with the known direction of eQTL effects in the literature?In the authors' new experiment, the results—as they themselves acknowledge—do not align with previously published findings. This discrepancy, in combination with the above uncertainties, remains a concern.
7. I agree with the authors' response.

Finally, The reporting summary is acceptable.

Reviewer #2 (Remarks to the Author):

The authors have addressed all my comments.

Reviewer #2 (Remarks on code availability):

works for the example, didn't try beyond that.

Reviewer #3 (Remarks to the Author):

Sincere apologies for the slow response.

With regards to my previous comments:

1. The authors have carried out a cell mixing experiment. The doublet rate and cross-contamination is within acceptable levels. This fully addresses my question.
2. The range of acceptable cell inputs is crucial information. The authors simply summarise what they did and not the range of acceptable inputs with their method. 1.5 million cells was the minimum number they used which would preclude many primary cell experiments. They suggest users of their method should work this out for themselves.
3. The authors did not address this question about mitochondrial genome mutations
4. The authors did not address this question and suggestion to include negative data on their attempts to establish RNA-seq with this method; something that many groups will no doubt attempt. This will slow down progress as other may repeat the same experiments that the authors already know do not work. Why not share this with the community now?
5. They are comparing apples and oranges with the ADO comparison. My sense is the ADO is somewhat higher than standard Tapestry with their method but hard to know with certainty without a direct comparison which they do not wish to do.
6. I remain unconvinced their technique is a major step forward with regards to ability to systemically study genome variation at scale, which they claim to be the main aim of their study. They acknowledge that the new data reported in rebuttal document were discordant with published data. A large number of the edited variants show no impact on expression which if they carefully selected variants with a known impact on gene expression is a concern.
7. I asked the authors to carry out more extensive pt analysis (not done) and link presence (or absence) or a known driver mutation in lymphoma to relevant alterations in gene expression (not done). The data shown are very preliminary with regards to convincing demonstration of the utility of this technique to study somatic variation in cancer or mosaicism.

Technically what is presented is sound and the mixing experiment is a useful addition. We disagree on other points, broadly about:

1. Providing maximum information for users of the technique, which in my view is important for a methods paper.
2. Definitive demonstration of utility of the technique i.e. how has this enabled insights into biology and/or disease that would otherwise not be possible. The data is currently preliminary.

I do not wish to delay publication and suggest this should now be an editorial decision.

Version 3:

Decision Letter:

29th Jul 2025

Dear Dr Steinmetz,

I am pleased to inform you that your Article, "Functional phenotyping of genomic variants using multiomic scDNA-scRNA-seq", has now been accepted for publication in Nature Methods. The received and accepted dates will be 18th Jun 2024 and 29th Jul 2025. This note is intended to let you know what to expect from us over the next month or so, and to let you know where to address any further questions.

Over the next few weeks, your paper will be copyedited to ensure that it conforms to Nature Methods style. Once your paper is typeset, you will receive an email with a link to choose the appropriate publishing options for your paper and our Author Services team will be in touch regarding any additional information that may be required. It is extremely important that you let us know now whether you will be difficult to contact over the next month. If this is the case, we ask that you send us the contact information (email, phone and fax) of someone who will be able to check the proofs and deal with any last-minute problems.

Authors may need to take specific actions to achieve compliance with funder and institutional open access mandates.

If your research is supported by a funder that requires immediate open access (e.g. according to [Plan S principles](https://www.springernature.com/gp/open-science/plan-s-compliance) or the [NIH public access policy](https://www.springernature.com/gp/open-science/us-federal-agency-compliance)) then you should select the gold OA route, and we will direct you to the compliant route where possible. Because authors warrant under our subscription licensing terms that they haven't committed to licensing any version of their article under a licence inconsistent with the terms of our agreement – including the applicable embargo period – publication under the subscription model isn't suitable for authors whose funders require no embargo.

If you are active on Twitter/X or Bluesky, please e-mail me your and your coauthors' handles so that we may tag you when the paper is published.

Best regards,
Lei

Lei Tang, Ph.D.
Senior Editor
Nature Methods

** Visit the Springer Nature Editorial and Publishing website at http://editorial-jobs.springernature.com?utm_source=ejP_NMeth_email&utm_medium=ejP_NMeth_email&utm_campaign=ejp_Nmeth for more information about our career opportunities. If you have any questions please click http://editorial-jobs.springernature.com?utm_source=ejP_NMeth_email&utm_medium=ejP_NMeth_email&utm_campaign=ejp_Nmeth

href="mailto:editorial.publishing.jobs@springernature.com">here.**

We thank the reviewers for their thorough evaluation of our manuscript. In response to their comments we included extensive revisions to our manuscript. These are the key modifications:

- **Data reanalysis for more stringent variant detection:** We reanalyzed all the data presented in the manuscript, setting a coverage threshold of 10 reads to ensure more confident gDNA allele detection. This threshold aligns with recommendations from the MissionBio scDNA-seq platform.
- **Allelic dropout analysis:** We conducted analysis on allelic dropout frequency using heterozygous variants in SDR-seq and put those results into context to other droplet, split-pooling, and plate-based technologies. This comparison demonstrates either higher specificity or throughput of SDR-seq relative to alternative methods.
- **Noise level assessment:** We evaluated noise levels in variant calling to determine the detection limit for low-abundance variants in SDR-seq, providing critical insights for future SDR-seq applications.
- **Doublet detection using sample barcodes:** We included data on doublet detection via sample barcodes introduced during in-situ reverse transcription, an approach analogous to species-mixing studies. This analysis illustrates our capacity to effectively remove doublets and contaminating reads on a per-cell basis.
- **Extended analysis on B-cell lymphoma samples:** We expanded our analysis of B-cell lymphoma patient samples with SDR-seq, incorporating variant comparisons to bulk gDNA sequencing, clustering on variant information, clonal analysis and more stringent variant filtering.
- **Comparison of gene expression variance:** We performed a comparative analysis of gene expression variance across SDR-seq, 10x Genomics, and ParseBio platforms, highlighting the greater sensitivity of SDR-seq's targeted readout in detecting specific genes.
- **Additional reviewer-suggested revisions:** We implemented several minor revisions, including clarifications within the text, additional technical details about SDR-seq such as primer panel design, cell input requirements, sequence length for variant calling and potential applications for mitochondrial variant lineage tracing.

Please find our comments as point-by point responses below in green. Reviewer comments are in black.

Passages that include significant text changes in the revised manuscript are highlighted in **green**.

Reviewer #1 (Remarks to the Author):

In the manuscript entitled “Functional phenotyping of genomic variants using multiomic scDNA-scRNA-seq” by Lindenhofer et al., the authors reported their new method, scDR-seq, to target sequence up to 480 DNA and/or RNA (cDNA) sequences from the same single cells from a large number (over thousands) of single cells per sample (per analysis). I think the method is novel and importantly could be very useful for investigators in the field of understanding tumor heterogeneity, especially the heterogeneity in gene expression regulation. However, I have some comments and suggestions as follows:

We thank the reviewer for their comments and acknowledging the significance of the presented scDNA-scRNA-seq technology.

1. Regarding mutations in gDNA, what is the minimum number of cells required to reliably detect a de novo mutation? My concern is that PCR-based amplification is relatively error-prone, and scDR-seq may not be able to detect de novo mutations occurring uniquely in one cell, which is acceptable. However, the minimum number of cells needs to be specified.

We thank the reviewer for this question as it is critical to correctly assess true variants and distinguish them from noise during the PCR and NGS sequencing. To address this, we added additional analysis of the SDR-seq panel size experiment (Extended Data Fig. 5c-i). gDNA targets in this experiment were designed to cover heterozygous variants based on publicly available bulk sequencing data. This SDR-seq data can be therefore reliably used to distinguish true heterozygous variants from background noise found in the experiment. Noise variants (falsely called transversions or insertions/deletions) were present in <0.15% of cells (Extended Data Fig. 5f, g), while heterozygous variants were correctly called in an average of 87-94% of cells in highly covered amplicons (detected in > 80 % of cells with more than 10 reads) (Extended Data Fig. 5c, d). Variant allele frequency (VAF) was a good indicator to

distinguish true variants from noise (Extended Data Fig. 5i). For true variants the VAF was centred around 0.5, while for noise variants the VAF was clearly different and centred around 0.1.

The scDNA-seq method from MissionBio using their Tapestry technology reports on a limit of detection of SNVs of 0.1 % depending on the type and length of the variant. This is in line with our observations (Extended Data Fig. 5f).

As for other methods, the de-novo detection of variants also depends on the total number of cells assayed together with other quality metrics. SDR-seq can assay around 9,000 cells in an individual experiment and assuming noise variants of <0.15 % the minimum number of cells needed for a specific variant is around 14 cells. However, the limit of detection also depends on the type of variant that needs to be detected as noise conversion rates differ depending on the base that is converted (Extended Data Fig. 5f). Depending on experimental needs and the data obtained different thresholds for variant detection may be used. Contrasting well-based methods, SDR-seq can measure thousands of cells simultaneously in a single experiment with high sensitivity on given loci of interest therefore also contributing to a better sensitivity to detect de-novo variants. We included this information now in the manuscript both in the results and discussion section.

2. For gDNA, what is the maximum number of base pairs per cell that can be covered? In the context of a typical cancer driver gene panel, what fraction of base pairs can be covered by scDR-seq?

The total amount of base pairs covered depends on the gDNA panel size in the experiment. On average each amplicon can cover around 180 bp excluding the primer regions (Reviewer Fig. 1a). In our panel size testing experiment we evaluated different panel sizes with up to 480 primer pairs in total, equally divided between gDNA and RNA targets. Therefore, the largest gDNA panel tested included 240 amplicons enabling highly accurate variant calling on 42.8 kb of sequence per cell (Reviewer Fig. 1b). We observed strong correlation between differently sized panels and did not reach saturation yet (Fig.

2a-e), so bigger panels or different proportions of gDNA and RNA targets could increase the number of base pairs for variant calling in a SDR-seq experiment.

Reviewer Figure 1

Reviewer Fig. 1. Overview of base pairs covered in SDR-seq experiments. a, Average base pairs covered per amplicons for variant calling. Sequences covered by primers were removed as they don't allow for accurate variant calling in these regions. **b,** Total base pairs covered for different sized gDNA panels in SDR-seq.

MissionBio offers commercially available gDNA panels to profile various cancer types with gDNA panel sizes ranging from 70 to 339 amplicons. These panels have been used in numerous studies to get insights about cancer heterogeneity and clonality within tumours (Matthew Schwede et al., 2024; Guangrong Qin et al., 2024; Monika M. Toma et al. 2024; Nadeu F et al. 2022; Blombery et al., 2022). The biggest panel sizes of SDR-seq are comparable to these commercially available ones, suggesting that they can be used in a similar fashion for gDNA readouts on a single cell level.

A commercial solution for cancer gene panels in bulk is the Illumina TruSeq® Amplicon Cancer Panel that covers around 35 kb with frequent mutational hot spots. This size is also comparable with the size that can be covered in SDR-seq. Other more unbiased targeted cancer gene panels for bulk analysis can cover more sequence but lack the single cell resolution for assessing variants. We added the information about the maximum number of base pairs covered using SDR-seq now in the discussion section of the manuscript.

3. For RNA, what fraction of genes, which are typically considered important to a tumor cell, can be covered by scDR-seq?

The selection of genes to assay depends on the tumour type and specific biological questions. To profile the B-cell lymphoma samples we chose genes based on a mixture of literature and data-driven genes informed by prior scRNA-seq profiling of these samples using 10x Genomics (Fitzgerald et al., 2023). Our gene panel included genes for maturation markers found in literature, variable features, differentially expressed genes and housekeeping genes. A detailed description of the gene selection criteria is provided in the methods section, and a full list of these genes is available in Supplementary Table 4.

Although targeted gene readouts are biased through prior selection, they allow for increased sensitivity for the targeted genes (Schraivogel et al., 2020). This is especially advantageous for examining relationships between specific variants and gene expression in a regulatory context as it enables detection of subtler expression changes driven by variants.

The same B-cell lymphoma samples used in this study were also profiled by scRNA-seq using 10x Genomics (Fitzgerald et al., 2023). This allows us to compare the clustering of malignant cells either with all genes measured or with the SDR-seq panel genes to profile these samples (Reviewer Fig. 2). Overall, cell clustering is very comparable in both conditions, indicating that cell type assignment is highly similar. This suggests that the SDR-seq panel genes included sufficient genes for cell type assignment.

For different tumour types the number and type of genes that are considered important can vary. Custom targeted RNA panels can be designed to answer specific questions also beyond tumour biology. We tested a total of 480 amplicons in a single experiment, while MissionBio's scDNA-seq platform was also successfully used with around 1000 amplicons. Therefore, depending on experimental needs

both the type of genes and their number can be adjusted. We included a better description on RNA target selection now in the discussion section of the manuscript.

Reviewer Figure 2

Reviewer Fig. 2. Clustering with all genes or panel genes in B-cell lymphoma samples. scRNA-seq B-cell lymphoma samples from Fitzgerald et al., 2023 that were using in this manuscript are either clustered with all genes found (top) or with genes that were used in SDR-seq experiment (bottom).

4. Finally, for RNA, what is the pair-wise correlation in expression levels observed by scDR-seq between any two cells of the same genotype? How does this compare to other droplet-based or non-droplet-based scRNA-seq methods?

We thank the reviewer for this comment as it's important to assess variability in gene expression in SDR-seq. The majority of our experiments were conducted using human iPSCs. Although these cells theoretically have the same genotype, there is some inherent variability in gene expression (Nguyen et al., 2018; Carcamo-Orive 2017). Therefore a measure of variability has to be taken with caution even when cells have the same genotype. Additionally, cells accumulate variants during cultivation in cell culture, which we also observe for the POU5F1 locus (Fig. 3k) which can contribute to gene expression variability.

To evaluate SDR-seq's gene expression performance relative to other scRNA-seq methods (10x Genomics and ParseBio) we analysed the relationship between average gene expression and variance

(Fig. 1o). Highly expressed genes show less variability than mid/lowly expressed genes, while SDR-seq shows less variability than other single cell methods that read-out gene expression either via droplets (10x Genomics) or split-pooling (ParseBio). This is due to the targeted approach of the gene expression readout in SDR-seq, enriching for transcripts of interest yielding better sensitivity for those. We included these analyses now in the revised manuscript.

Reviewer #2 (Remarks to the Author):

The study by Lindenhofer et al., reports a significant technology advancement for single cell targeted DNA and RNA co-assay with moderate throughput and improved coverage.

Technology-wise, the new method allows recovery of 80% of all gDNA targets in more than 80% of cells across all panels, which hold high promise as a technical solution to link genotype and gene expression in pooled screens. This opens doors to a wide range of applications in certain cell lines, although generalizability to other systems may need substantial further work. A different fixative glyoxal, is tested with the commonly used PFA and an improved sensitivity observed for RNA recovery is informative and helpful to the scRNA-seq field.

We thank the reviewer for commenting on the significance of our manuscript and its potential impact.

My major concern of the current manuscript is with the design for one of the proof-of-principle application linking mutational burden with gene expression. Single-cell DNA sequencing in general is subject to very high false positive SNV calls, and one of the major reason is spontaneous deamination in the amplification process. I suggest that the authors validate the results by adding a USER (NEB) enzyme step (a mixture of UDG and endonuclease VIII) immediately prior to amplification.

We thank the reviewer for this comment. We have now quantified frequency of noise during variant calling and added this data to the manuscript (Extended Data Fig. 5c-i). In the experiment to test the scalability of SDR-seq by using differentially sized panels, gDNA target sites were chosen based on the presence of heterozygous SNPs measured by bulk gDNA sequencing. This allows us to discriminate between true heterozygous variants and noise levels (Extended Data Fig. 5h). Overall the frequency of noise is very low for miscalled variants causing a base conversion using SDR-seq, ranging from 0.004-0.17 % depending on the base that is converted (Extended Data Fig. 5f). Indeed conversion mutations

caused by deamination (A->G/T->C or (C->T/G->A) are the most frequent miscalled variants with 0.12-0.17 % and 0.05-0.06 % with conversions from A->G/T->C being the most frequent. The enzyme suggested by the reviewer would only address conversion caused from C->T via an intermediate Uracil product caused by deamination of Cytosine. Additionally, cleavage of Uracils via the suggested USER enzyme might result in the production of chimeric PCR products during library generation and thereby cross contamination of gDNA reads into the remaining cells. Since noise levels are low, C->T are not the most frequent base conversion and the potential risk of contaminating gDNA reads due to chimeric PCR by products, we think that the potential benefit of including a USER enzyme step is not warranted. Nonetheless this was a worthwhile suggestion for us to consider noise and conversion rates and included this data now in the revised manuscript.

Additionally, we included updated analysis to link mutational burden in the B-cell lymphoma samples to elevated levels of B-cell receptor signalling. We subset cells for high and low variant burden in DZ and LZ maturation states and extended our analysis from BLK to all genes that are associated with B-cell receptor signalling in our targeted gene panel (Fig. 4i). This shows that cells with higher mutational burden have generally elevated levels of B-cell receptor signalling. We added this data now to the revised manuscript and updated the text accordingly.

Otherwise, the study is carefully conducted and results are clearly presented.

Reviewer #2 (Remarks on code availability):

The code works for the example file.

Reviewer #3 (Remarks to the Author):

The authors describe SDR-Seq as a novel droplet-based single-cell genomics technique to investigate eQTL in functional genomic screens and tumor heterogeneity at high throughput. Using the established Tapestri platform, by carrying out glyoxal-based cell fixation and in-situ reverse transcription, it enables targeted genotyping and targeted gene expression analyses, including linking of different variants to distinct gene expression changes. Preliminary validation experiments are carried out in cell lines and primary samples within the context of B-cell lymphomas. This technique is claimed to allow interrogation of several hundred targets at both gDNA and RNA level in parallel.

There is clearly an unmet need for techniques that allow high throughput single cell genotyping and gene expression analysis. Overall the data presented convincingly establishes that it is possible to generate targeted gene expression data and gDNA sequencing in individual cells. However, the validation of the technique is incomplete and application to uncover new biology very preliminary and unconvincing.

1. With regards to additional QC and validation data it would be helpful to understand the cell yield and doublet rate by loading variable numbers of cells and with cell mixing experiments to definitively establish doublet rates and cross-contaminating reads which seems quite high; is it correct that 5% of reads mapped to the incorrect sample barcode (this is high)?

We thank the reviewer for raising these important considerations. SDR-seq leverages the microfluidic system of MissionBio's Tapestri technology, which determines the doublet rate in this system. Doublet rates have been assessed by MissionBio previously (P028, MissionBio support) and studies using scDNA-seq using Tapestri technology (Zhang et al., 2023; Yuan et al., 2024). These report doublet rates of around 8% at the recommended cell loading concentration, typically yielding around 3000 cells.

SDR-seq is designed to introduce sample barcodes during the in-situ reverse transcription step, uniquely tagging transcripts from cells in different wells before loading onto the Tapestry microfluidic system. This barcoding approach allows for sample multiplexing within a single Tapestry run. For our initial proof-of-principle experiment, we used two distinct sample barcode sets to distinguish between PFA and glyoxal fixation conditions, thereby creating a mixing experiment analogous to species-mixing studies. We assessed doublets prior to the doublet removal step in our analysis (Extended Data Fig. 1b), finding approximately 15% of cells identified as doublets before removal. We reasoned that due to the use of our sample barcoding we can easily remove doublets during our analysis and therefore loaded more cells, yielding around 9,000 cells in this experiment. For subsequent experiments, we expanded from two to eight sample barcodes to enable more stringent doublet removal.

Ambient RNA is a well-known issue for each droplet-based single cell technology, typically ranging from 1-3% in 10x Genomics data, while computational tools have been developed to tackle this issue (Yang et al., 2020; Young et al., 2020). Companies that commercialised Split-seq like ParseBio or ScaleBio advertise their single cell solutions in part as having removed ambient RNA significantly compared to droplet-based technologies like 10x Genomics, enabling better resolved clustering and separation of cell types. They reason that this is due to the extensive washing during each split-pooling step. Due to sample barcoding before the droplet stage during in-situ reverse transcription, SDR-seq allows to remove contaminating reads that contain sample barcodes not belonging to this cell without the need of such computational tools. This enables us to clean up the data before further downstream analysis and is an advantage compared to other droplet based methods like 10x Genomics.

We have added an analysis of doublet detection and removal in the revised manuscript, along with additional details on doublet and contaminating read detection in the methods section.

2. What is the minimum number of cells required for this technique? This is important as cell number is often limiting from primary patient samples.

In the majority of our experiments we used a total of 480,000 cells during the in-situ reverse transcription step with 10,000 cells per well. This count ensures that, even after some cell loss during the pooling spin step, there remains a sufficient number of cells as input for droplet generation on the Tapestri machine.

If cell numbers are limited per sample, multiple samples can be fixed and processed in separate wells during in-situ reverse transcription, using distinct sample barcodes to enable multiplexing and achieve the necessary cell count to fully utilise an entire Tapestri lane. Alternatively, fewer cells can be loaded for the Tapestri droplet generation step although this would result in overall fewer cells that are recovered per run. During fixation we observe a cell loss of around 10-30% depending on the sample which needs to be taken into account. So overall the minimum number of cells needed depends on experimental needs and the number of samples that should be multiplexed on the same Tapestri run.

We also processed primary patient samples for this study (Fig. 4). For one of these runs we used 350,000 cells as an input for the reverse transcription which was the smallest input we used. This yielded a total of 8,400 cells for this particular SDR-seq run, demonstrating that even with lower cell numbers as an input we can yield good recovery of cells.

We have included a more detailed explanation of input cell numbers and cell loss in the methods section of the manuscript. We thank the reviewer for highlighting this crucial point, as it is valuable for readers assessing the suitability of SDR-seq for their applications.

3. Can this technique be used to amplify the mitochondrial genome, which would be helpful for clonal tracing.

Yes the in-situ reversed cells contain mitochondria allowing for the use of specific primer pairs to target and amplify the mitochondrial genome for variant calling. This approach indeed holds promise for

clonal tracing within cell populations making it a highly compelling application of SDR-seq. The scDNA-seq product of MissionBio has been used for such applications already (Penter et al., 2023). We thank the reviewer for this suggestion and have now included this potential application in the discussion section of the manuscript.

4. Have the authors tested whole transcriptome amplification using smart-seq template switch primers. This should be tested and presented even if it does not work.

We performed experiments to obtain whole transcriptome readouts in SDR-seq by using template switch oligos (TSO) in combination with targeted gDNA readout (Reviewer Fig. 3a). Specifically, we used a TSO during the in-situ RT reaction followed by corresponding amplification primers in the droplets to capture full-length transcripts. While we successfully recovered gDNA targets in this experiment, RNA molecule recovery was minimal (Reviewer Fig. 3b-d). This means our initial attempts for a combined whole transcriptome readout in SDR-seq were not yet successful but we have ongoing efforts to test other conditions. Given that these efforts are ongoing and very preliminary, we feel it may not be valuable to include them in the manuscript at this stage.

Reviewer Figure 3

Reviewer Fig. 3. Attempted whole transcriptome capture using SDR-seq. **a**, Overview of the SDR-seq method for whole transcriptome capture using a TSO during reverse transcription. **b**, Number of RNA UMIs and gDNA reads detected per cell using the TSO method. **c**, Violin plot of the distribution of the unique genes (left) and RNA UMIs (right) detected per cell. **d**, Violin plot of the distribution of the unique gDNA targets (left) and gDNA reads (right) detected per cell.

5. The gDNA coverage seems quite low even when using a very low bar of only 5 reads per cell to define detection (which is inadequate for genotyping). What is the overall allelic dropout rate and how does this benchmark for the same gDNA loci for the standard Tapestry method and plate based techniques for parallel gDNA and GE analysis? This can be assessed by studying known heterozygous SNPs. What is driving the 'failure' for detection of approx. 20% of gDNA loci? More information on panel design for RNA and gDNA and how this influences detection would be helpful.

We thank the reviewer for these valuable comments. Allelic dropout (ADO) occurs when a sample is genotyped, but one or both alleles are not detected. In a single diploid cell there are only two gDNA molecules present that need to be amplified simultaneously for proper variant calling on a single cell level. This contrasts to techniques that use RNA molecules for variant calling, which are predominately more abundant and easier to detect. Amplification from a single allele or insufficient coverage would lead to an ADO.

MissionBio reports an ADO rate below 10 % for well-covered amplicons in their commercial scDNA-seq technology. To determine ADO in SDR-seq we added additional analysis of the panel size experiment (Extended Data Fig. 5c-i) to address this question. In this experiment we designed gDNA targets to cover heterozygous variants identified through public bulk sequencing data which allowed us to assess ADO in SDR-seq. We selected amplicons with high coverage defined as detected in more than 80 % of cells (Extended Data Fig. 5c) and selected for heterozygous SNPs. Across the panels tested (120, 240 and 480 total size - half of them gDNA target sizes) we could correctly call on average 87-94 % of heterozygous SNPs (Extended Data Fig. 5d). Higher dropout rates in larger panels correlated with reduced detection of a given amplicon/cell (Extended Data Fig. 5e). These numbers are overall comparable to ADO rates provided by the MissionBio support. As described above, one contributor to ADO could be the amplification from only one gDNA allele during the multiplexed PCR on a single cell level. Indeed we find cells with high coverage of either REF or ALT alleles, whereas the majority of them are correctly called as HET (Reviewer Fig. 4, Extended Data. Fig. 5d). However, selecting cells

based on overall coverage per cell per target site would not help remove these cells (Reviewer Fig. 4). Instead, another selection criteria was to only include variants with a GQ value bigger than 30, which removes variants that are highly covered, but where the variant allele frequency (VAF) is on the edges of the HET distribution. However, to remove poorly covered cells in terms of read depth we reanalyzed all our data in the manuscript with 10 reads per cell per target site as a threshold to classify detection for allele and variant calling. This is also the threshold used and recommended by MissionBio support.

Reviewer Figure 4

Reviewer Fig. 4. Heterozygous variants in panel size experiment. Heterozygous variants found in the 120 panel of the panel size experiment at three different gDNA target sites. Variant allele frequency and read depth are shown for each cell. Top panels show the assigned genotype, bottom the genotype quality as colour. Chromosomal positions and variants are indicated.

Methods that have a similar throughput to SDR-seq and enable a simultaneous readout of RNA and gDNA in the same cell are DEFND-seq and HIPSD&R-seq (Olsen et. al., 2023; Lazareva et al., 2023). Both of them use nucleosome depletion followed by gDNA tagmentation and loading on a 10x Chromium multiome platform that was designed to both capture chromatin accessibility and a whole transcriptome readout. ADO for calling variants at defined loci is very high and these methods are predominantly used to determine copy number variants (CNVs). DEFND-seq reports that either one REF/ALT or both alleles were detected in 76 or 56 cells out of 1,821 nuclei for two distinct variants, respectively. This contrasts the high allelic detection rate of 87-94% using SDR-seq. Another technology with high throughput is sci-L3-RNA/DNA which uses a split-pooling indexing approach

(Yin et al., 2020). gDNA readout for this technology is also based on tagmentation, and they report similar detection results for variants at distinct loci as DEFND-seq and HIPSD&R-seq.

There are numerous plate based methods for a combined single-cell readout of RNA and gDNA which inherently have limited throughput (Yu et al., 2023; Rodriguez-Meira et al., 2019; Macaulay et al., 2015; Han et al., 2017; Hou et al., 2016; Dey et al., 2015; Chung et al., 2024; Zachariadis et al., 2020; Li et al., 2015; van Strijp et al., 2017). These either separate gDNA/RNA before library generation or amplify them in the same reaction. For gDNA readouts they rely on tagmentation, multiple displacement (MDA) or primary template-directed amplification (PTA). Tagmentation and MDA-based technologies also suffer from high allelic dropout rates while PTA-based technologies enable a more uniform coverage of the genome (Luquette et al., 2022). The commercial solution for PTA-based gDNA readout with a combined transcriptome readout (ResolveOME) enables a high recovery of correctly called alleles (> 90 %) when libraries are sequenced at saturation. To reach saturation they recommend sequencing each gDNA library of a single cell at a depth of 250 Million reads per cell. Due to the targeted nature of SDR-seq, accurate variant calling for loci of interest could be achieved at a much lower sequencing depth (10,000-20,000 reads/cell) while the throughput was drastically increased (~100 fold). We have added a discussion of allelic dropout of SDR-seq compared to other methods to the discussion section of the revised manuscript.

MissionBio offers gDNA panels with pre-tested primers to cover variants in various cancer samples that range from 70 to 339 amplicons. As gDNA targets sites for our experiments did not overlap with those from these commercial panels we used custom gDNA panels for all our experiments that were designed with the Tapestry Designer tool from Mission Bio. As these primers were not experimentally validated beforehand a certain drop-out is to be expected as not all of them will perform well in a multiplexed setting. Validating primers in an experimental setting would also entail performing a scDNA-seq, SDR-seq experiment or bulk multiplexed PCR followed by NGS. We directly tested primers in a SDR-seq setting for all our experiments. Failing primer pairs can be replaced in follow-up

experiments for target sites of interest. We now included a more detailed description of the primer design in the methods section of the manuscript to clarify this issue.

A direct comparison of SDR-seq and standard Tapestry gDNA sequencing may yield only limited insights, as SDR-seq will include RNA primers and thus a larger total number of amplicons. Higher number of amplicons in the multiplexed PCR and changes in the overall compositions of panels can decrease the overall coverage of some gDNA targets and change the overall coverage pattern (Figure 2b, c; Extended Data Fig. 4d, e) which will affect ADO rates. It would thus be difficult to attribute any decreases in performance to the SDR-seq method vs panel size. Instead we checked the variance of read coverages between gDNA panels using SDR-seq and public Mission Bio datasets (Reviewer Fig. 5). MissionBio data sets showed on average lower variance compared to SDR-seq. However, they used validated gDNA primer panels, while SDR-seq gDNA primer panels were not validated before usage.

Reviewer Figure 5

Reviewer Fig. 5. gDNA panel uniformity across SDR-seq and public MissionBio datasets. Average gDNA read variance per target is shown for SDR-seq and public MissionBio datasets.

6. The prime editing is extremely inefficient making it impossible for validation of the technique for the primary purpose outlined in the abstract. The authors need to demonstrate that their technique can be used to reliably detect altered expression of a gene following PE of a range of eQTLs with known effect on gene expression (or not). Demonstrating that their technique can identify alterations in gene expression following CRISPRi or introduction of a STOP codon is a very low bar and insufficient for validation purposes.

In the abstract we highlight that inefficient editing makes it necessary to directly read out editing at distinct gDNA loci. We acknowledge that prime editing was inefficient in iPSCs, while base editors showed improved results. Nonetheless, SDR-seq enables us to associate both coding (STOP codons) and non-coding (eQTL) variants with gene expression changes, even at low editing efficiencies, as we can detect minimal editing outcomes and link them with gene expression levels. Additionally, we showcase another application of SDR-seq in primary B-cell lymphoma samples, where cells with higher mutational burden exhibit elevated levels of B-cell receptor signalling. These and other potential applications underscore the value of combined variant and gene expression readouts offered by SDR-seq, which is a key feature emphasised in our abstract. We adjusted the text in the abstract to highlight the variable editing efficiencies in the editing screens we observe.

We respectfully disagree with the reviewer's comment that experiments using CRISPRi or STOP codons cannot test the overall sensitivity of SDR-seq to detect gene expression changes. CRISPRi allows us to assess the effect of strong perturbations while STOP codons enable us to detect more variable gene expression effects elicited by nonsense-mediated decay. Our results show that the vast majority of CRISPRi gRNAs that target transcription start sites show strong and significant perturbation effects on their respective target genes (Fig. 3a-c, Extended Data Fig. 6a-c). Depending on the length of the transcript and their position STOP codons are known to have effects on expression levels of the respective gene. We show that we can associate STOP variants within transcripts to expression levels in an allele-specific manner (Fig. 3d-g; Extended Data Fig. 6d, e; Extended Data Fig. 7). Together, these experiments demonstrate that SDR-seq is in principle sensitive to detect strong and known gene expression changes, important information for any potential application of SDR-seq.

Additionally, we performed a base editing screen to target eQTLs (Fig. 3h-k, Extended Data Fig. 8). eQTLs are commonly identified by correlating genetic variation to gene expression using either bulk or single cell data. These eQTL variants then explain a part of the gene expression variation that is found

in this particular cell type or model system. While usually one particular lead eQTL is reported, many variants are found in high linkage disequilibrium which could be equally causative for the observed gene expression changes. Many of those variants are specific to a given cell type or model system and not easily transferable to others while they are predominantly not experimentally validated and poorly characterised. Therefore picking highly confident eQTLs is challenging. We chose eQTLs in iPSCs based on multiple studies, prioritising variants reported more than once (DeBoever et al., 2017; Cuomo et al., 2020). Furthermore, they were selected for predicted effects on highly expressed target genes, overlap to ATAC-seq peaks and compatibility to be installed via either adenine or cytosine base editors (see methods). This resulted in 56 high confidence targets that were tested using SDR-seq. The most interesting results were observed for an eQTL variant located at the POU5F1 locus (Fig. 3k). We also observed inherent genetic variability at this locus, with additional effects on POU5F1 gene expression. This highlights the power of SDR-seq to associate variants to gene expression profiles in iPSCs. Therefore we already included data of eQTLs that can have an effect on gene expression in the manuscript.

Apart from showcasing that eQTLs can elicit gene expression changes in the manuscript, we think that a more extensive study of eQTL loci is beyond the scope of this study but would potentially be an interesting mechanistic follow-up study using SDR-seq. We included textual modifications in the revised manuscript to better highlight the intent of the CRISPRi and editing experiment and discuss the limited editing efficiencies we observe in the discussion section.

7. The lymphoma data is very preliminary and is presented in a way that is unclear. The authors should sequence a larger number of tumors in bulk, determine the key mutations associated with the tumors and then apply their technique to show that variants can be reliably detected in B-cells (and absent from other cell lineages that are not part of the tumor). The impact of subclonal mutations on gene expression then be presented.

We thank the reviewer for these points and apologise for being not clear in how variants were selected for the B-cell lymphoma SDR-seq experiment. Targeted gDNA panels were chosen based on previous bulk gDNA sequencing data from these samples (Fitzgerald et. al., 2023). gDNA panels were designed for regions with >20 % variant allele frequency, allowing for targeted detection of relevant variants. A comparison of SDR-seq to this bulk gDNA sequencing is now included in the manuscript (Extended Data Fig. 9d). For the majority of variants we find them exclusively in B-cells, while some other somatic variants are also found in non-B-cells. This shows that SDR-seq can reliably detect variants in cell types of interest that are also detected with orthogonal technologies. This data and detailed information on variant selection for the B-cell lymphoma SDR-seq experiment is now added to the main text and methods section of the manuscript.

Additionally, we included a more detailed analysis of the primary B-cell lymphoma patient samples (Fig. 4, Extended Data Fig. 9 and 10). This included a more stringent read cutoff of 10 for variant calling and allele detection, along with more stringent variant filtering for downstream analysis. We added clustering analysis based on variants and subsequent clone identification which shows that the composition of maturation states can vary within different clones (Fig. 4e). Additionally, we extended our analysis to link mutational burden to elevated levels of B-cell receptor signalling. Subsetting cells into high and low variant burden within DZ and LZ maturation states, we expanded our analysis from BLK to all genes that are associated with B-cell receptor signalling in our targeted gene panel. We can show that cells with higher variant burden display generally higher average expression of genes that are involved in B-cell receptor signalling (Fig. 4i). We believe this demonstrates the analytical potential SDR-seq provides to the scientific community.

We thank the reviewers for their thorough evaluation of our manuscript. In response to their comments we performed several revisions. These include:

- **Species cell mixing SDR-seq experiment:** We performed a species cell mixing experiment using human and mouse cells to check for cross contamination of nucleic acids during SDR-seq. This demonstrates little cross contamination of both gDNA (< 0.16 %) and RNA (< 0.8-1.6 %).
- **Variant editing in cardioids:** We tested non-coding variants for their impact on gene expression on nearby genes. This identified several variants that had a significant impact on gene expression of nearby genes.
- **Comparison of gene expression variance:** We cross-correlated gene expression levels of individual cells across SDR-seq, 10x Genomics, and ParseBio platforms. This confirms greater sensitivity due to SDR-seq's targeted readout in detecting specific genes.
- **Detailed protocol for SDR-seq:** To improve accessibility for potential users of SDR-seq we provide a detailed protocol on protocols.io.
- **Additional reviewer-suggested revisions:** We implemented several minor textual revisions and clarifications.

Please find our comments as point-by point responses below in green. Reviewer comments are in black. Passages that include significant text changes in the revised manuscript are highlighted in .

Reviewer #1:

Remarks to the Author:

My question #4 was not directly addressed.

We apologize for not directly addressing this point in the previous response as we thought the question aimed overall at the variance that is measured in gene expression using SDR-seq. We previously put this in context to other single cell methods that read out gene expression (Fig. 1o). This shows that SDR-seq has reduced gene expression variance compared to 10x Genomics and ParseBio.

We now correlated cell-to-cell gene expression levels of 100 subsampled cells in our proof-of-concept SDR-seq experiment, 10x Genomics and ParseBio data acquired in human iPSCs. This data is in line with the above described analysis using gene expression variance as a standard for measurement stability. We included this analysis in our revised manuscript (Extended Data Fig. 2e).

Line 343: "These methods enable a whole genome ..." MDA and PTA are primarily used for de novo SNV detection. I do not think that PCR-based methods, including SDR-seq and the MissionBio system, can effectively serve this purpose. This is due to the significantly higher error rate of PCR—several orders of magnitude greater than the rate at which de novo mutations occur at the single-cell level. In fact, the noise variants, typically centered around a VAF of ~0.1%, pose a detection limit for these methods (as the authors showed in the revision). In context, high-depth bulk WES (2000x) can reliably detect true de novo SNVs (rather than artifacts) around this same VAF threshold. For this reason, I don't think SDR-seq is well-suited for de novo SNV detection, but SDR-seq would be important for other purposes as demonstrated by the authors. The above could be addressed by revising the discussions in the manuscript.

We thank the reviewer for this comment and revised the corresponding section in the manuscript.

Line 347: "SDR-seq offers significantly improved throughput (~100-fold) compared to PTA-based ..." This statement is incorrect. SDR-seq covers only 40 kb of the genome, whereas PTA can cover almost the entire 3 Gb human genome per single cell. Even at a scale of 9,000 cells processed via SDR-seq, the total genome coverage is only ~10% of a single-cell genome, which PTA can achieve. Therefore, SDR-seq does not offer comparable throughput in terms of genome coverage. However, again, I don't think this was why SDR-seq was developed for, and could be addressed by revising the discussions.

We clarified this statement accordingly in the corresponding section of the discussion, now specifying throughput in terms of cell numbers and highlighting the limitation of SDR-seq in terms of genome coverage due to its targeted approach.

Reviewer #2:

Remarks to the Author:

The authors have addressed all my comments. I agree with their reasons for not doing the suggested experiments with USER nicking at the deamination sites. The additional analyses are careful and thorough.

We thank the reviewer for their comments.

Reviewer #3:

Remarks to the Author:

Largely the authors have not addressed my previous comments which is disappointing as I don't think these suggestions were unreasonable

1. Cell mixing experiments were not done using the established conditions with glyoxal i.e. a mix of cells within the same starting well that can then be genetically demultiplexed to establish definitively the doublet rate using the glyoxal fixation conditions. This is important information for anyone wishing to use the technique. The issue with cross contaminating reads is not dealt with and again, a cell mixing experiment will establish cross-contamination between cells (for RNA and potentially DNA) using their technique.

To address the reviewers' concern we now included a cell species mixing experiment in the revised manuscript using human iPSCs and mouse NIH-3T3 cells (Extended Data. Fig 2f-k). Using cells either individually or mixed during the *in-situ* RT, we can compare contamination of nucleic acids during *in-situ* RT or generally from ambient RNA during droplet generation and multiplexed PCR. This experiment shows that little contamination of gDNA (< 0.16 %) or RNA (<1.6 %) occurs between species taking the sample BC information into account that is introduced during *in-situ* RT. Additionally, the sample BC information can be used to effectively remove the vast majority of doublets and contaminating ambient RNA. We thank the reviewer for suggesting this experiment as its also of high interest for any potential user of SDR-seq. These data are included in the revised manuscript.

2. I asked what the minimum number of cells is for their technique. The authors respond by describing the cell numbers they used in the experiments they had already carried out, which does not answer the question. This requires additional experimentation to titrate down cell

numbers and establish a reasonable window of cell input using this approach. This is a major issue with primary cell material and important information for a method paper.

It is difficult to provide a specific minimum input cell number because yields from the fixation and RT step are variable across users and cell types, so it would be recommended for users to test yields for themselves in their application. For the SDR-seq experiments presented in the manuscript, cell counts during the *in-situ* RT reaction were carefully calculated so that they remained within the optimal input range for the Tapestri device. Varying cell numbers as input for the Tapestri device has been carried out by MissionBio and is inherent to the microfluidic device. We therefore believe that the proposed experiment of varying cell numbers and input for SDR-seq would only yield limited additional insights.

To address the reviewer's concern, we added a clearer description of the minimum cell numbers that were used in this study in the discussion section of the manuscript. Additionally, we provide a detailed protocol on protocols.io under DOI: [dx.doi.org/10.17504/protocols.io.6qpvr9q43vmk/v1](https://doi.org/10.17504/protocols.io.6qpvr9q43vmk/v1). This protocol will be published alongside the manuscript in case of acceptance. Currently it is available with this private link for reviewers.

3. Perhaps I should have been more specific here in the way this comment was phrased. Of course, I appreciate that in principle the technique could be used to amplify mitochondrial genomes, I was rather expecting/requesting that the authors carry out some additional experimentation to demonstrate this approach using their method (which would also in parallel provide some validation of the veracity of mutation calling).

We think this is beyond the scope of this paper. It's a very specialized application and recent studies question the reliability of mitochondrial variants for clonal tracing (Lareau et al., 2024; Wang et al., 2024).

4. Whole transcriptome analysis would increase my enthusiasm for this method a great deal. It is a shame that this did not work. I think it is helpful to include this in the manuscript as it is something quite obvious that users will be interested in trying and may waste a lot of time repeating experiments already conducted.

We disagree with the reviewer on this point and highlighted this also in the previous response to reviewers. As this is a topic of ongoing efforts and we do not have publishable data on this yet, we do not think it's valuable to include them. In addition, we see particular value in having a targeted readout as it enables higher sensitivity to detect gene expression changes for selected genes. In the revised manuscript we now discussed that our previous attempts for a whole transcriptome method were unsuccessful, while other experimental approaches in the future might work.

5. The ADO seems high. The authors are biasing their analysis by only selecting amplicons with a high coverage (defined as detected in >80% of the cells), which if I understand correctly is only 80% of the amplicons. To directly compare with Tapestry ADO it should be easy to simply include gDNA primers for a run to account for the issue with panel size the authors are concerned about. As things stand, it is very possible that their method increases ADO rate in comparison with a standard Tapestry method, and that this could be optimised. There are many reasons why their method which permeabilizes the cells might increase ADO and this should be tested.

Our revised manuscript now contains a detailed description of ADO rates in SDR-seq, which we think was a valuable suggestion from the reviewer. However, further comparison of our dropout rates to those of the standard Tapestry method is of limited value for users of SDR-seq.

MissionBio reports allelic dropout (ADO) rates of <10 % in their scDNA-seq panels with validated primers that generate highly covered amplicons. Our data demonstrate that SDR-seq

correctly calls heterozygous SNPs at rates of 87-94 % for highly covered amplicons. These results indicate that the ADO rates for both methods are highly comparable and strongly suggest that cell permeabilization in SDR-seq has minimal, if any, impact on ADO rates.

The high similarity of ADO rates suggests that the proposed experiment to directly compare the performance of gDNA primers in SDR-seq versus standard scDNA-seq would very likely detect minor differences - if any. Therefore we believe that this does not warrant additional experiments.

To address this point, we included a statement about ADO rates of standard Tapestry scDNA-seq and SDR-seq in the revised manuscript.

6. The manuscript sets out that the primary aim is to establish a method to systemically study impact of genome variants on gene expression. What I am suggesting is that they address this with some (a few e.g. 5) well-established eQTLs occurring in functional genome regulatory elements that are already proven to be functional, of which there are many examples e.g. rs2836882 risk allele has a clear effect on ETS2 expression in bulk analysis as recently reported in Nature. The authors should select a small number of such well validated function SNPs found at risk alleles and validate that upon editing they can detect the expected change in gene expression in the relevant cell type. This will provide robust support to their overarching aim that their technique can be used to systematically study genome variation. Identifying a change in gene expression by knocking down a gene with CRISPRi (or STOP codon) is a very low bar and does not really test what the authors themselves set out to address in this manuscript. I do not think the data in 3K address this crucial point.

To address the reviewers' concern, we evaluated the effect of non-coding de novo variants (DNVs) previously shown to affect gene expression of nearby genes in iPSC-derived cardiomyocytes (Xiao et

al., 2024). To introduce these variants, we used both base editing (BE) and an HDR-based approach involving Cas9-induced cleavage combined with a single-stranded template oligonucleotide, similar to the previous study (Xiao et al., 2024) (Reviewer Fig. 1a). BE and HDR electroporations/lipofections were performed separately, and iPSCs were mixed before differentiation into cardioids to account for variability in differentiation (Hofbauer et al., 2021). On day 8 of differentiation, cardioids were subjected to SDR-seq using a gene panel that included marker genes, housekeeping genes, and target genes within a 700 kb window surrounding the DNV sites. We identified cardiomyocyte, cardiac fibroblast, and endocardial cell types, with each experimental condition represented at similar levels (Reviewer Fig. 1b, c). Editing efficiency was unfortunately generally low for the majority of intended edits, while also Cas9-mediated insertions and deletions could be observed (Reviewer Fig. 1d). Overall, editing was more efficient with the HDR-based approach. We tested both intended and bystander edits for their effects on gene expression of target genes within the 700 kb window (Reviewer Fig. 1e), identifying several variants with a significant impact on gene expression in cardiomyocytes and cardiac fibroblasts (Reviewer Fig. 1f).

The differentially expressed genes identified in SDR-seq partially overlapped with those reported previously using isogenic cell lines and qPCR (Xiao et al., 2024). Variants at two of four DNV sites affected the same genes reported by Xiao et al., while we found genes affected that were not reported to be changed in their study together with other genes that they did not assess. This discrepancy could arise from several factors, including low editing for the intended edits in our experiment impairing statistical power, the inclusion of insertions/deletions and bystander edits in our analysis, variability in cardiomyocyte differentiation using isogenic cell lines, differences in model systems (3D cardioids in SDR-seq vs. 2D differentiated CMs in the previous study) or differences in the time point of measurement (day 8 after differentiation in SDR-seq vs. day 17 in the previous study). Especially variability in differentiation using the isogenic cell lines in terms of CM purity and quality might contribute to differing results. SDR-seq allows to associate variants with gene expression changes in distinct cell types individually, whereas the bulk approach would be sensitive to such differentiation variations.

This data suggests that variants at non-coding regions can be associated with gene expression changes of nearby genes using SDR-seq. The discrepancy to the previous study analyzing these DNV-sites needs to be addressed in further experiments, which we believe is beyond the scope of this study.

Reviewer Figure 1

Reviewer Fig. 1. Editing at non-coding DNV sites in cardioids. **a**, Overview of the experimental outline to introduce variants at non-coding DNV sites via electroporation (HDR) or lipofection (BE). Human iPSCs were differentiated into cardioids before performing SDR-seq at day 8 of differentiation. **b**, UMAP highlighting the different cell types clustered by gene expression. Number of cells for each cell type is indicated as a bar graph. **c**, UMAP highlighting the different experimental conditions clustered by gene expression. Number of cells within

an experimental condition is indicated as a bar graph as percentage of total. **d**, Editing efficiencies for each experimental condition as bar graph. HET (cyan) and ALT (green) alleles shown for each edit. Intended edit for each DNV-site is indicated by a star. **e**, Volcano plot for non-coding DNV experiment indicating foldchange and *P*-value. Significant hits (*P*-value < 0.05) are colored. Comparison between the different alleles is shown as shapes. REF, reference allele; HET, heterozygous allele; ALT, alternative allele. DE testing by Wilcoxon rank sum test. **f**, Impact of non-coding DNVs shown on cardiomyocytes and fibroblasts. Variant shown on bottom, significantly altered genes are shown on top. Color indicates expression (Z-score, data is scaled by column), genotypes and DNV-sites. Of note, variants on the X chromosome can only be of ALT genotype as the used human iPSC cell line (WTC-11) is of male origin.

SDR-seq enables the confident detection of gene expression changes in diverse experimental settings, including CRISPRi, STOP codons and eQTLs (Fig. 3, Extended Data Fig. 6-8). In Fig. 3f-h, we installed eQTL variants using base editing and measured the resulting gene expression. As multiple eQTLs are significant here (Fig. 3g), this demonstrates SDR-seq's ability to link genome variants with gene expression changes. Fig. 3h further shows that naturally occurring variation can simultaneously be detected and associated with expression differences, highlighting the method's sensitivity and versatility.

Our data demonstrate that non-coding variants can indeed affect gene expression (Reviewer Fig. 1), despite discrepancies with the previous study. Given that we have already shown links between installed SNVs and expression changes in the current manuscript, we believe the value of adding this additional DNV editing data to the manuscript is limited.

7. I suggested analysis of a larger patient cohort which was not done. The primary patient analysis remains preliminary with only three patients analysed. What are the specific driver mutations they analysed (CREBBP, KMT2D, EZH2? Etc). Was detection of any specific individual driver mutation associated with distinct gene expression changes. Do the identified clones allow a

clonal structure to be inferred and in the non-B-cells can a pre-lymphoma clone be identified? How does the ADO impact this? Integration with mitochondrial genome analysis would provide an extra layer of validation here (see point 3). Are there any non-mutant B-cells. Are these different from normal healthy donor B-cells? etc. Overall, in my view, the data as presented does not convincingly validate or illustrate the utility of this technique. The power of combined gene expression and mutation analysis in individual cells to analyse primary patient material is that it allows an intra-patient comparison of WT versus mutant cells within the same cell type; it would be helpful if some illustrative examples could be presented in a larger patient cohort.

We believe that extending this analysis to more patients is beyond the scope of the current manuscript as the current data showcases the power of using SDR-seq to profile such samples. We conducted a thorough analysis on these primary patient samples, showing that cells with a higher mutational burden display elevated B-cell receptor signaling. Patients were chosen based on previous bulk sequencing data (Fitzgerald et. al., 2023) as described in the manuscript. We find distinct variants enriched in B-cells compared to non-malignant cells. Using SDR-seq we performed analysis to detect differentially enriched variants between distinct maturation states, and compared transcription levels between cells with and without variants in a global manner rather than focussing on individual variants. The additional analysis requested by the reviewer would not aid in further illustrating our main finding, which is that cells with higher mutational burden show distinct gene expression profiles associated with malignancy, and would only marginally improve the demonstration of the utility of SDR-seq.

I think the authors have developed a really interesting technique but there are obvious and important outstanding QC and validation experiments that in my view should be carried out before this is published as a top tier methods paper.

Reviewer #1 (Remarks to the Author):

The authors have addressed all of my concerns as Reviewer #1.

Regarding the concerns raised by Reviewer #3:

1. The response is acceptable.

Thank you for your comment.

2. Although the two links provided in the response do not work for me, the number of input cells used in each experiment is specified in the revised manuscript. The reported input cell numbers are acceptable.

We apologize that the link did not work for the protocols.io. We updated the link now in the revised manuscript.

3. I agree with the authors' response.

Thank you for your comment.

4. I agree with the authors' response, although I note that a whole-transcriptome assay would indeed be more innovative—albeit significantly more costly.

Thank you for your comment.

5. The response is acceptable.

Thank you for your comment.

6. In this case, I share the concerns raised by Reviewer #3. The evidence presented in Figure 3g may not be sufficiently convincing. Some key information appears to be missing:

– Are the 56 high-confidence eQTLs known causal variants?

eQTLs are typically discovered through statistical associations between genetic variants and the expression levels of nearby or distant genes. However, an association does not necessitate that the genetic variant is the direct cause of the expression change. As described in the methods section of the manuscript, variants were selected based on two publications that identified eQTLs in iPSCs (DeBoever et al., 2017; Cuomo et al., 2020), which overlapped with ATAC-seq peaks (Ren et al., 2021) and were compatible to be installed with adenosine or cytosine base editors. To our knowledge, none of the eQTLs tested in this study has been experimentally tested for causality of gene expression changes. We modified the corresponding section in the manuscript to further detail selection criteria and indicate the associative nature of the selected and tested eQTLs.

“In addition to installing eQTLs with PE, we tested the use of base editors (BE) in human iPSCs. We selected 56 high-likelihood eQTLs with a potential association for gene expression changes based on multiple studies, including non-coding variants, that are located in open chromatin and editable with ABE8e or CBE base editors³⁶⁻³⁸ (Fig. 3f). None of these variants have previously been experimentally validated in an endogenous context as causative for transcriptional regulation to our knowledge.”

– For the known causal variants, are the expression changes reported in Figure 3g consistent with the known direction of eQTL effects in the literature?

As described above, none of the variants that were tested in this study had been experimentally validated as causal. As these variants were only predicted to have association with differential gene expression, we do not necessarily expect all of them to align with observed effects by SDR-seq.

For an eQTL variant located the POU5F1 locus, the strongest hit that we focus on in the manuscript, the beta directionality and the fold-change of the variant effect correspond (Cuomo et al., 2020, Beta: -1.38; SDR-seq log2FC: -0.19). However, not all of our hits are in agreement with the eQTL betas. This is not surprising: eQTL studies do not always nominate the correct lead SNP. It is possible that nearby variants in high linkage disequilibrium actually drive observed expression differences while the lead SNP has no, or a different effect. Adjacent variants can cause gene expression changes in different directions (Martyn et al., 2025), potentially also in a combinatorial manner, as showcased by the natural genetic variation at the POU5F1 locus we identified (Fig. 3h). A primary motivation for SDR-seq is to grant the ability to experimentally validate or disqualify SNP eQTLs, which we do here.

We modified the corresponding section in the manuscript to include the information about the corresponding beta values with our strongest hit.

“Human iPSCs accumulate somatic mutations during cell culture, while they undergo constant competitive selection for variants that are advantageous in culture conditions³⁹. We found a synonymous variant in the 3' end of POU5F1, a gene encoding for a critical pluripotency factor, which significantly altered gene expression in the same direction as observed in prior eQTL studies³⁶ (Extended Data Fig. 8a). However, after assessing variants that may have accumulated during culturing along the entire amplicon of POU5F1, we found that certain combinations of variants showed different effects on POU5F1 expression (Fig. 3h, Extended Data Fig. 8b).”

In the authors' new experiment, the results—as they themselves acknowledge—do not align with previously published findings. This discrepancy, in combination with the above uncertainties, remains a concern.

Indeed, we find discrepancies to the previous study that analyzed the variants we tested in cardioids using SDR-seq. As we pointed out in our previous response to reviewers, this could have several reasons. These include low statistical power due to low editing for the intended edits in the SDR-seq experiment, the inclusion of insertions/deletions and bystander edits in the differential gene expression testing using SDR-seq, variability in cardiomyocyte differentiation and purity impacting bulk qPCR measurements performed by Xiao et al. versus single-cell measurements and cell-type resolved differential gene expression testing using SDR-seq, differences in model systems or differences in the time point of measurement. Slight differences in differentiation efficiencies of both cardiomyocyte purity (ratio of cardiomyocytes to fibroblasts or other cell types) and quality (i.e., atrial vs. ventricular cardiomyocytes) might significantly confound interpretation of bulk measurement results in the previous study while SDR-seq enables to perform analysis in a cell-type specific manner.

We performed additional analysis to highlight the validity of measured effects in this variant SDR-seq cardioid experiment. This includes control statistical testing to discriminate if the measured effects could only be due to noise/heterogeneity from SDR-seq measurements. To do so we randomly shuffled genotypes during statistical testing instead of using our measured and assigned genotypes (Reviewer Fig. 1f). This shows that no significant gene expression changes are observed if we randomly assign genotypes during statistical testing, strongly suggesting that significant gene expression changes observed by our true assigned genotypes measure real biological changes. Additionally, we want to highlight that gene expression changes elicited by our measured variants are predominantly dosage dependent by allele, with HET variants showing an intermediate effect between REF and ALT, in line with expectations (Reviewer Fig. 1g).

Reviewer Figure 1

Reviewer Fig. 1. Editing at non-coding DNV sites in cardioids. **a**, Overview of the experimental outline to introduce non-coding de novo variants (DNV) for congenital heart disease (CHD) via electroporation (HDR) or lipofection (BE). Human iPSCs were differentiated into cardioids before performing SDR-seq at day 8 of differentiation. **b**, UMAP highlighting the different cell types clustered by gene expression. Number of cells for each cell type is indicated as a bar graph. **c**, UMAP highlighting the different experimental conditions clustered by gene expression. Number of cells within an experimental condition is indicated as a bar graph as percentage of total. **d**, Editing efficiencies for each experimental condition as bar graph. HET (cyan) and ALT (green) alleles shown for each edit. Intended edit for each DNV-site is indicated by a star. **e**, Volcano plot for non-coding DNV experiment using individual variants indicating foldchange and *P*-value. Significant hits (*P*-value < 0.05) are colored. Comparison between the different alleles is shown as shapes. REF, reference allele; HET, heterozygous allele; ALT, alternative allele. DE testing by Wilcoxon rank sum test. **f**, Volcano plot displaying control statistical testing results, in which genotypes were randomly assigned within cardiomyocyte and fibroblast clusters. The number of cells assigned to each random genotype matched the number observed for actual genotypes. Likewise,

the number of differential expression tests performed mirrored that of the real dataset. Comparison between the different alleles is shown as shapes. REF, reference allele; HET, heterozygous allele; ALT, alternative allele. DE testing by Wilcoxon rank sum test. **g**, Impact of non-coding DNVs shown on cardiomyocytes and fibroblasts. Variant shown on bottom, significantly altered genes are shown on top. Color indicates expression (Z-score, data is scaled by column), genotypes and DNV-sites. Of note, variants on the X chromosome can only be of ALT genotype as the used human iPSC cell line (WTC-11) is of male origin.

We further investigated gene expression changes elicited by combinatorial presence of variants (Reviewer Fig. 2a). We specifically chose to evaluate the opposing effects on ACVR1B by DNV-1 by two by variants in close proximity could be disentangled by this analysis (Reviewer Fig. 2b). Taking the combinatorial effects of variants into account, all variants lead to the downregulation of ACVR1B expression. Variant-centric analysis that focuses only one specific position ignores variants at other positions that could also impact gene expression (Reviewer Fig. 2b). This highlights the importance of considering the combinatorial effects of variants when performing this analysis, as we also showcase in the manuscript (Fig. 3h).

The data of manuscript (CRISPRi, prime editing of STOP codons and base editing for eQTLs) and the review experiment shown here (HDR/base editing for congenital heart disease variants) strongly suggest that SDR-seq is capable of detecting gene expression changes in general and caused by individual variants in both coding and non-coding regions.

Reviewer Figure 2

Reviewer Fig. 2. Editing at non-coding DNV sites in cardioids using combined genotype information. a, Volcano plot for non-coding DNV experiment using combinatorial genotypes indicating foldchange and P -value. Significant hits (P -value < 0.05) are colored. DE testing by Wilcoxon rank sum test. **b**, Gene expression for combined genotypes (left) and variant-centric analysis (right), exemplifying the importance of using combined genotype information for differential gene expression analysis.

7. I agree with the authors' response.

Thank you for your comment.

Finally, The reporting summary is acceptable.

Thank you for your comment.

Reviewer #2 (Remarks to the Author):

The authors have addressed all my comments.

Thank you for your comment.

Reviewer #2 (Remarks on code availability):

works for the example, didn't try beyond that.

Thank you for your comment.

Reviewer #3 (Remarks to the Author):

Sincere apologies for the slow response.

With regards to my previous comments:

1. The authors have carried out a cell mixing experiment. The doublet rate and cross-contamination is within acceptable levels. This fully addresses my question.

Thank you for your comment.

2. The range of acceptable cell inputs is crucial information. The authors simply summarise what they did and not the range of acceptable inputs with their method. 1.5 million cells was the minimum number they used which would preclude many primary cell experiments. They suggest users of their method should work this out for themselves.

We show in our manuscript that SDR-seq can be applied to human iPCS, a cell culture model system, and primary patient samples. Reporting the cell input numbers that worked for these applications is highly important information for anyone with similar cellular models or primary samples. Exhaustively testing cell input for all possible model systems is beyond the scope of this study. Due to the high heterogeneity of different samples that can be processed, some level of user specific testing will need to be done depending on experimental objectives.

3. The authors did not address this question about mitochondrial genome mutations

We think that assessing mitochondrial mutations in the context of SDR-seq is beyond the scope of this study.

4. The authors did not address this question and suggestion to include negative data on their attempts to establish RNA-seq with this method; something that many groups will no doubt attempt. This will slow down progress as other may repeat the same experiments that the authors already know do not work. Why not share this with the community now?

As we pointed out we are working on this currently. Adding this data is beyond the scope of this study.

5. They are comparing apples and oranges with the ADO comparison. My sense is the ADO is somewhat higher than standard Tapestri with their method but hard to know with certainty without a direct comparison which they do not wish to do.

As we highlighted in our previous response, we think this comparison is beyond the scope of this study.

6. I remain unconvinced their technique is a major step forward with regards to ability to systemically study genome variation at scale, which they claim to be the main aim of their study. They acknowledge that the new data reported in rebuttal document were discordant with published data. A large number of the edited variants show no impact on expression which if they carefully selected variants with a known impact on gene expression is a concern.

As detailed in our response to reviewer 1, we performed additional analysis on the variant experiment in heart organoids performing random assignment of genotypes as opposed to our measured one as a control for the differential gene expression testing (Reviewer Fig. 1f). This shows no significant changes if genotypes were shuffled and assigned randomly, strengthening the effects we observe for the variants tested. As pointed out in our previous response, we believe that the observed changes could be due to differences in the model system used, together with the measurements in bulk in the previous study from Xiao et al., and the single cell measurements performed with SDR-seq, alongside other differences in the experimental setups. Slight differences in differentiation efficiencies of both cardiomyocyte purity and quality (i.e., atrial vs. ventricular cardiomyocytes) might confound interpretation of results in the previous study.

7. I asked the authors to carry out more extensive pt analysis (not done) and link presence (or absence) or a known driver mutation in lymphoma to relevant alterations in gene expression (not done). The data shown are very preliminary with regards to convincing demonstration of the utility of this technique to study somatic variation in cancer or mosaicism.

As pointed out previously we believe this is beyond the scope of this study.

Technically what is presented is sound and the mixing experiment is a useful addition. We disagree on other points, broadly about:

1. Providing maximum information for users of the technique, which in my view is important for a methods paper.

We believe that we provided all the information that is useful for potential users of SDR-seq. This includes exact information about experimental design, cell input, primer design and data analysis. Testing of cell input for specific model systems might be necessary for many single cell applications, while we aim to include a WTX SDR-seq method in future publications if our attempts are successful.

2. Definitive demonstration of utility of the technique i.e. how has this enabled insights into biology and/or disease that would otherwise not be possible. The data is currently preliminary.

We show that SDR-seq is capable of amplifying hundreds of both gDNA targets and genes simultaneously in the same cell, with the ability to correctly determine variant zygosity in the > 87-95% of cells. By employing a series of experiments (CRISPRi, prime editing for STOP codons and base editing for eQTLs in iPSCs), we can show that SDR-seq is sensitive to detect gene expression changes

and confidently link them to variants. Furthermore, we show that SDR-seq is capable of profiling primary B-cell lymphoma patient samples

I do not wish to delay publication and suggest this should now be an editorial decision.